# Robust cytoplasmic partitioning by solving a cytoskeletal instability

Melissa Rinaldin[1,2✉], Alison Kickuth[1,2,7], Adam Lamson[1,7], Benjamin Dalton[3], Yitong Xu[4,6], Pavel Mejstřík[2], Stefano Di Talia[4] & Jan Brugués[1,2,5✉]

Early development across vertebrates and insects critically relies on robustly reorganizing the cytoplasm of fertilized eggs into individualized cells[1,2]. This intricate process is orchestrated by large microtubule structures that traverse the embryo, partitioning the cytoplasm into physically distinct and stable compartments[3,4]. Here, despite the robustness of embryonic development, we uncover an intrinsic instability in cytoplasmic partitioning driven by the microtubule cytoskeleton. By combining experiments in cytoplasmic extract and in vivo, we reveal that embryos circumvent this instability through two distinct mechanisms: either by matching the cell-cycle duration to the time needed for the instability to unfold or by limiting microtubule nucleation. These regulatory mechanisms give rise to two possible strategies to fill the cytoplasm, which we experimentally demonstrate in zebrafish and *Drosophila* embryos, respectively. In zebrafish embryos, unstable microtubule waves fill the geometry of the entire embryo from the first division. Conversely, in *Drosophila* embryos, stable microtubule asters resulting from reduced microtubule nucleation gradually fill the cytoplasm throughout multiple divisions. Our results indicate that the temporal control of microtubule dynamics could have driven the evolutionary emergence of species-specific mechanisms for effective cytoplasmic organization. Furthermore, our study unveils a fundamental synergy between physical instabilities and biological clocks, uncovering universal strategies for rapid, robust and efficient spatial ordering in biological systems.

Physical mechanisms have a fundamental role in establishing boundaries within living systems, from the intracellular level to collectives of organisms[5–8]. In early embryos, cell boundaries are established by rapid cleavage divisions that robustly organize the cytoplasm into progressively smaller cellular compartments[1,2]. The compartmentalization of the cytoplasm can occur before[4,9] or without[3,10] the formation of a new plasma membrane, raising the question of how boundaries between cytoplasmic compartments can be robustly maintained in the absence of physical barriers. Experiments using reconstituted cytoplasm have revealed that cytoplasmic compartments self-organize spontaneously[3,4]. The formation and division of these compartments rely on microtubule asters that define their boundaries[11–14] and dynein activity that transports organelles towards the compartment centre[15] (Extended Data Fig. 1a). Microtubule asters grow via self-amplifying microtubule growth or autocatalytic nucleation, the nucleation and branching of microtubules from existing microtubules[13,16]. Through branching nucleation, asters can explore a large volume of cytoplasm until they meet other asters. When asters enter in contact, the aster–aster interface is thought to be stabilized by components that provide local inhibition to microtubule nucleation and growth, creating robust boundaries that guide cytokinesis[17–19]. This process leads to a regular tessellation of the cytoplasm. However, it is unclear how local inhibition in combination with autocatalytic growth can lead to stable and robust boundaries[20,21]. To shed light on this problem, we combined theory with experiments in reconstituted cytoplasm and living zebrafish and *Drosophila* embryos. Starting from a theoretical prediction, we show that microtubule autocatalytic nucleation gives rise to aster invasion driving the coarsening of cytoplasmic compartments. By performing cell-cycle perturbations and biophysical measurements of microtubule dynamics, we found that the coarsening of cytoplasmic compartments is prevented either by synchronizing the cell-cycle oscillator to the dynamics of the asters or by reducing autocatalytic nucleation. Finally, we show that these mechanisms yield to divergent cytoplasmic organization strategies in embryos.

## Cytoplasmic partitioning is unstable

We investigated cytoplasmic partitioning in live zebrafish embryos and *Xenopus laevis* egg extracts. In zebrafish embryos, cytoplasmic partitioning occurs before cytokinesis. During the first rounds of cell division, microtubule asters divide the cytoplasm into two distinct cytoplasmic compartments—denser regions of cytoplasm rich in organelles such as mitochondria—before the cell membrane ingresses (Fig. 1a–d, Extended Data Fig. 1b–d and Supplementary Video 1). This

[1]Cluster of Excellence Physics of Life, TU Dresden, Dresden, Germany. [2]Max Planck Institute of Molecular Cell Biology and Genetics, Dresden, Germany. [3]Fachbereich Physik, Freie Universität Berlin, Berlin, Germany. [4]Department of Cell Biology, Duke University Medical Center, Durham, NC, USA. [5]Max Planck Institute for the Physics of Complex Systems, Dresden, Germany. [6]Present address: Program in Developmental Biology, Sloan Kettering Institute, Memorial Sloan Kettering Cancer Center, New York, NY, USA. [7]These authors contributed equally: Alison Kickuth, Adam Lamson. ✉e-mail: melissa.rinaldin@tu-dresden.de; jan.brugues@tu-dresden.de

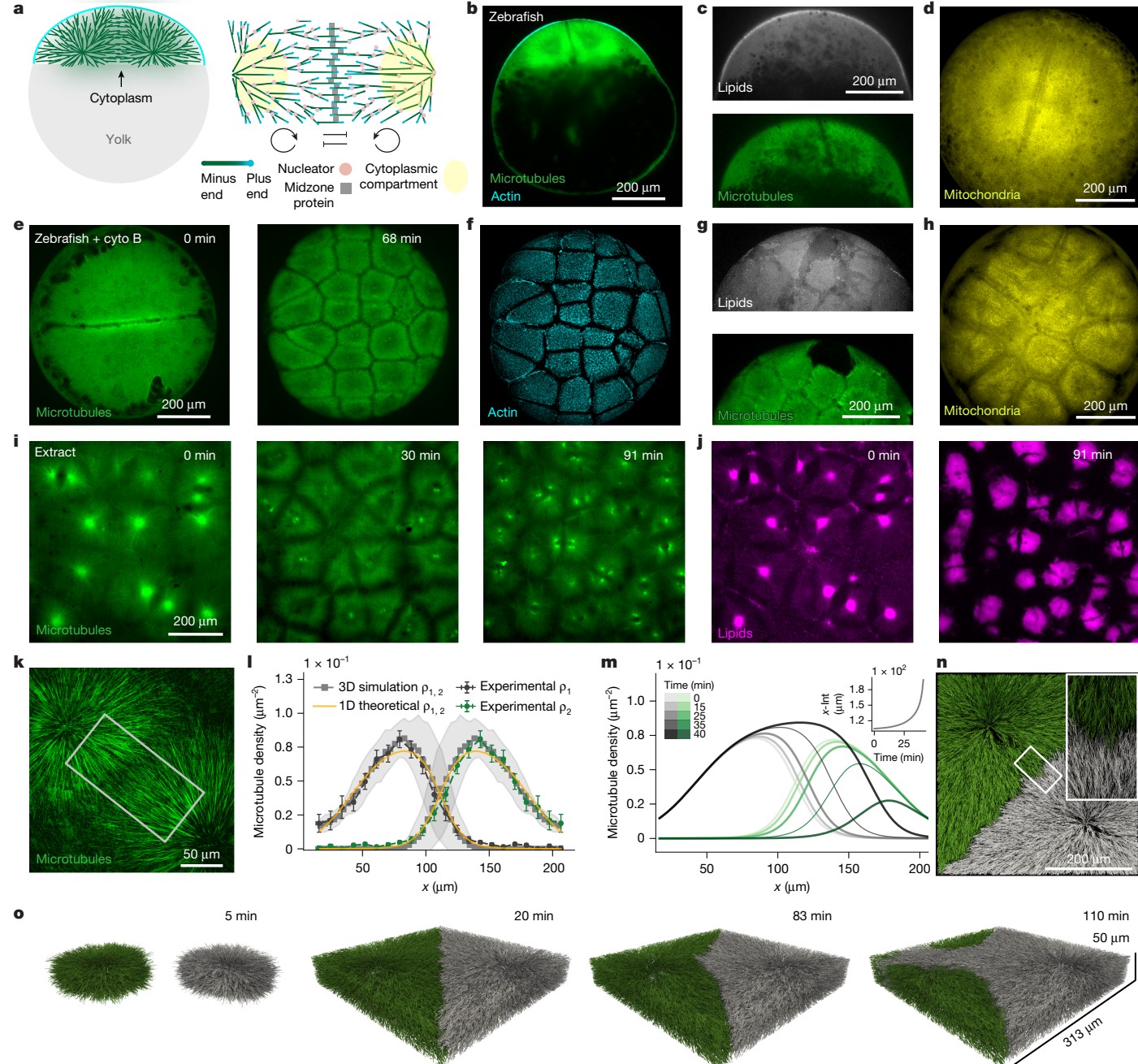

**Fig. 1 | Robust compartmentalization is observed in vitro and in vivo, but theory and simulations predict a physical instability. a**, Schematic of an early zebrafish embryo, with microtubule asters in green and actin cortex in cyan (left), and a schematic of cytoplasmic compartmentalization (right). An aster–aster interaction can be described by a network of two self-amplifying loops interacting via local inhibition. **b**, Light-sheet fluorescence microscopy image of a live zebrafish embryo after the first division. Asters partition the cytoplasm before furrow ingression. Microtubules are shown in green with eGFP–Doublecortin and the actin cortex in cyan with utrophin–mCherry. **c**, Cell membrane (top) and microtubules (bottom) of a zebrafish embryo with PH-Halo and eGFP–Doublecortin. **d**, Mitochondria of a zebrafish embryo with mito-GFP. **e,f**, Live imaging of a syncytial zebrafish embryo. Asters coexist and form boundaries of low microtubule (**e**) and actin density (**f**). Cyto B, cytochalasin B. **g**, Cell membrane (top) and microtubules (bottom) of a syncytial zebrafish embryo. **h**, Mitochondria of a syncytial zebrafish embryo. **i**, Live imaging of

cycling frog egg extract showing cytoplasmic partitioning. Microtubules are shown in green. **j**, Cytoplasmic compartments are visualized in magenta by labelling lipid organelles. **k**, Two asters interacting. **l**, Microtubule density profile of two interacting asters. $x$ indicates the linear coordinates from the centre of one aster (0 µm) to the centre of the adjacent aster (approximately 200 µm). Experimental data are shown in black and green with s.e.m. ($n = 8$ independent samples), agent-based simulations are in grey with 95% confidence interval ($n = 6$ independent simulations) and one-dimensional theory is in orange. **m**, Numerical time evolution of microtubule densities. The inset shows the interface position over time. **n**, Top view of a 3D agent-based simulation of interacting asters in a slab showing boundary formation. Grey and green indicate microtubules of the two asters. The inset shows the interface details. **o**, Side view of temporal evolution of a 3D agent-based simulation showing invasion.

observation led us to test whether the embryo can divide its cytoplasm in the absence of cell membranes between cytoplasmic compartments. To this end, we inhibited the formation of cleavage furrows and cell

membrane ingression by adding cytochalasin B[22], an actin polymerization inhibitor. We observed low-density regions of microtubules and depolymerized actin between compartments over multiple cell

cycles, indicating that the division of the cytoplasm in living zebrafish embryos does not require cytokinesis (Fig. 1e–h, Extended Data Fig. 1e and Supplementary Video 1). In frog extracts, undiluted cytoplasm obtained by crushing frog eggs at high speed self-organizes into distinct compartments that are not separated by cell membranes, similarly to syncytial systems[3]. These compartments form in the absence of cell membranes and divide over multiple cell cycles (Fig. 1i,j, Extended Data Fig. 1f–j and Supplementary Video 2). These results demonstrate that cytoplasmic partitioning is a fundamental process in cell division that precedes and is independent of cytokinesis.

The striking similarities in cytoplasmic partitioning between frog egg extracts and live embryos suggest that extract is a prime system to investigate this process, as it has the advantage that it is easy to manipulate and image. To quantify the formation of compartment boundaries, we measured the microtubule density profile using EB1–mApple, as it labels the growing plus ends of microtubules. We tracked individual EB1–mApple plus ends and reconstructed the density and polarity of microtubules of two compartments from the centre of one aster to the centre of the adjacent aster (Fig. 1l and Extended Data Figs. 1k and 2a–d and Supplementary Note 1). The two profiles corresponding to each compartment have an exponential increase close to their centre (around 0 and +200 μm in the $x$ axis) consistent with autocatalytic growth. Near the interface (around 100 μm in the $x$ axis), the microtubule profiles decay consistent with local inhibition at the antiparallel microtubule overlap[17,18]. These profiles suggest that the interaction between the two asters can be minimally described by a network of two autocatalytic, or self-amplifying, loops interacting via local inhibition (Fig. 1a, bottom). To test whether such a network can explain the robust formation of these compartments, we used a 1D continuum theory of aster–aster interaction (microtubules mostly grow into one direction in the midzone region; Extended Data Fig. 2h–j), incorporating autocatalytic growth, microtubule polymerization, microtubule turnover[16,23] and local inhibition (Supplementary Note 2), resulting in the following equation:

$$\frac{\partial \rho_i}{\partial t} = \mp v_p \frac{\partial \rho_i}{\partial x} + \alpha \frac{\rho_i}{1 + (\rho_1 + \rho_2)/\rho_s} - \theta \rho_i - \lambda \frac{\rho_1 \rho_2}{\rho_1 + \rho_2} \tag{1}$$

The first three terms describe the dynamics of the growth of one single aster[16]. The last term describes phenomenologically the local inhibition between the two asters that could result from crosslinking of antiparallel microtubules, decreasing microtubule polymerization by kinesins and increasing microtubule turnover[17,18]. In this equation, $i = 1,2$ and refers to the two asters, $v_p$ is the polymerization velocity, $\theta$ is the microtubule turnover, $\alpha$ is a parameter related to the autocatalytic growth, $\lambda$ modulates the inhibition between asters, and $\rho_s$ is a density of microtubules that indicates the saturation of microtubule nucleation due to depletion of nucleators as they bind to microtubule. We measured $v_p$ by tracking the plus ends of the microtubules and $\theta$ using single-molecule microscopy of sparsely labelled tubulin dimers. We then globally fit the model to the density profiles to obtain the other parameters. A detailed description of the measurements and values of the parameters is reported in Supplementary Notes 1 and 2 and Supplementary Tables 4 and 6. The theoretical fit to the aster density profiles quantitatively agrees with the experiments (Fig. 1l, orange line). To take into account the geometry and microscopic details of microtubule branches, we validated the continuum theory using 3D agent-based simulations of two interacting asters, which also quantitatively recapitulate the aster density profiles (Fig. 1l, grey lines and Extended Data Fig. 3) and the microtubule organization in 3D (Fig. 1n). We found that both theory and simulations predict that the temporal evolution of these boundaries is unstable (Fig. 1m,o and Supplementary Video 3). Extending the 1D model to two and three dimensions shows that this instability is independent of dimensionality in our system (see Extended Data Fig. 2e–g and Supplementary Note 2.1). This instability is generally expected from local inhibition and self-amplification alone[20]; however,

it disagrees with the robustness of cytoplasmic organization observed in embryos and cycling extracts[3,4].

One possible explanation for this apparent inconsistency between theory and experiments is that the time needed to develop such instability may be larger than the cell-cycle time, which drives the disassembly of the microtubule asters before the assembly of mitotic spindles. Close to the unstable point, the time to develop the instability can become arbitrarily large. Indeed, our numerical solutions suggest that the time to develop this instability can easily be up to 40 min (Fig. 1m), which is comparable with the cell-cycle time in both frog extracts and frog embryos, which are equal to about 40 min and 30 min, respectively[24,25]. To test whether the cell cycle prevents the development of the instability, we arrested cytoplasmic extracts in interphase by blocking translation of cyclin B1 with cycloheximide[3]. Cycloheximide did not affect the speed of microtubule polymerization, turnover (Supplementary Table 4) and overall compartment growth (Supplementary Video 4). We observed that compartments that initially formed with a well-defined boundary as in the control condition, started coarsening by means of the microtubule asters invading each other and fusing, consistent with the aster invasion predicted by our theory (Fig. 2a and Supplementary Video 5). This coarsening can continue for several hours, leading to compartments of few millimetres in size, in contrast to hundreds of microns in the cycling extract. During the coarsening, dynein motors relocated nuclei to the new centre of the larger compartments (Fig. 2a, bottom). However, dynein activity does not hinder the invasion process because invasions occur before active transport of nuclei and organelles, as well as when dynein is inhibited (Extended Data Figs. 4–6 and Supplementary Video 6). Dynein inhibition enhances invasion by leading to splayed asters, presumably due to the lack of proper pole formation[26] (see dynein-inhibited cycling extract in Supplementary Video 6).

To rule out possible artefacts due to global inhibition of protein translation by cycloheximide, we also specifically blocked the translation of cyclins selectively by using morpholinos following existing protocols[27,28] and observed an increase of the cell-cycle duration and aster invasion events (Supplementary Video 7). With a closer examination using higher-resolution imaging (Supplementary Video 8), we observed that an invading aster gained mass, consistent with continuous autocatalytic growth, at the expense of the invaded aster, that eventually disappeared (Fig. 2b). The aster invasion and compartment fusion are accompanied by disassembly of the chromosomal passenger complex at the aster–aster interface. This indicates that inhibition between the asters is present at the beginning of the interaction, consistent with previous experiments[15,17–19], but is disrupted by autocatalytic nucleation over longer timescales (Extended Data Fig. 6e). This invasion dynamics was also reproduced in the 3D agent-based simulations (Fig. 2c). In the simulations, the asters are confined in a rectangular slab, and the invasion occurs laterally, as also seen in the experiments, often with deformations of the interface in a finger-like manner (Supplementary Video 8). Together, our results show that cytoplasmic partitioning is an intrinsically unstable mechanism.

## Cell cycle can prevent the instability

Our results suggest that the cell-cycle duration can determine whether invasion events occur and thus regulate the patterns of cytoplasmic partitioning. To further investigate this dependence, we experimentally quantified the invasion time as a function of the aster mass difference, $\Delta M_i$. We experimentally introduce differences in aster mass by exploiting local variations of sperm nucleus densities in the imaged sample. We calculated the mass of the asters from the area under the curve of 1D profiles of microtubule density (Extended Data Fig. 7a–d). We defined the invasion time $\tau$ as the time for the initial mass difference between the asters $\Delta M_i$ to decrease by a factor $e$ (Fig. 2d). The invasion time decays as the mass difference between the asters increases. This trend was also perfectly captured by a parameter-free prediction of

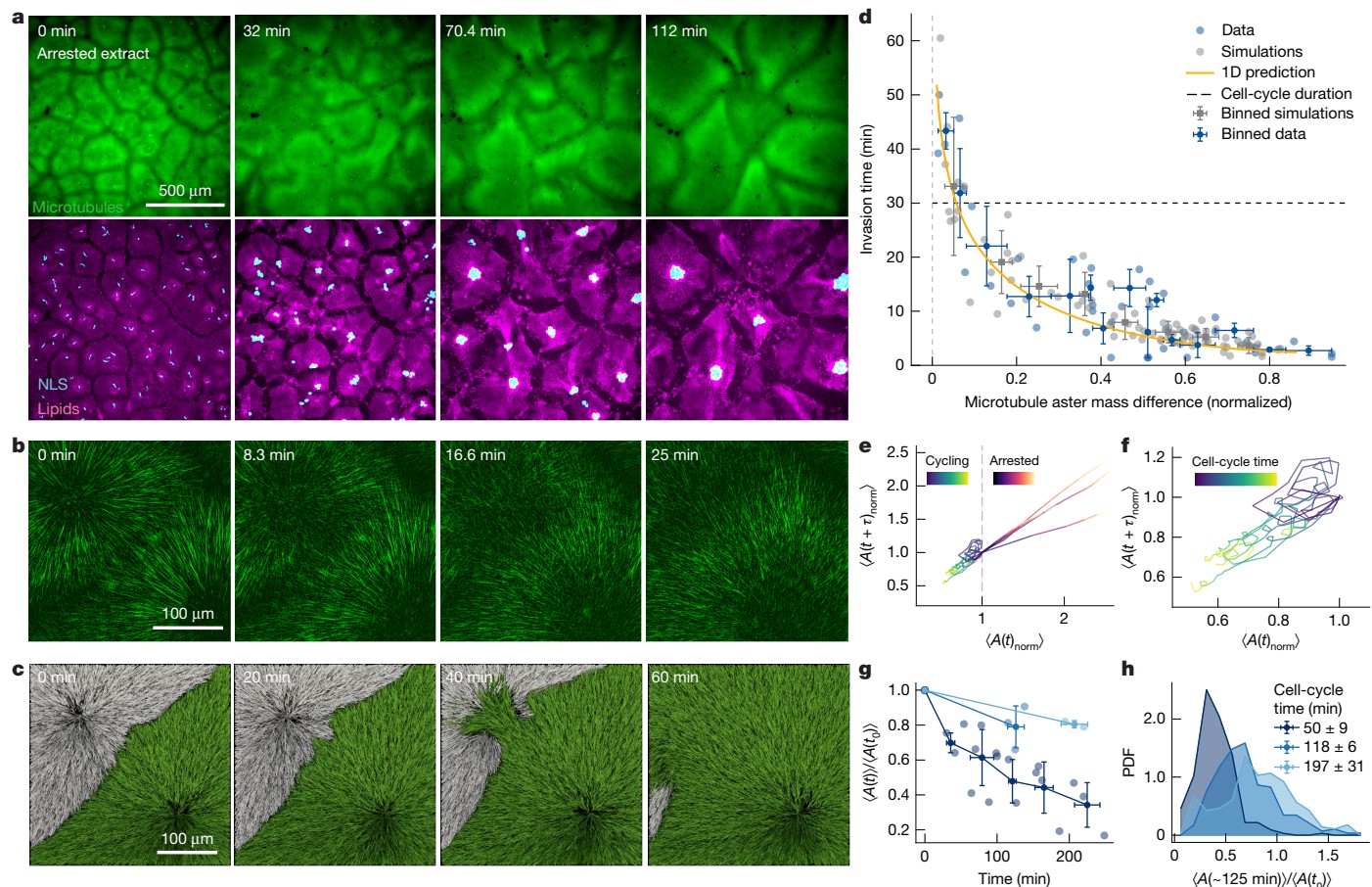

**Fig. 2 | Cytoplasmic partitioning is intrinsically unstable, but the cell-cycle duration can avoid the instability leading to robust compartmentalization. a**, Live imaging of interphase-arrested cytoplasmic extract showing microtubule aster invasion. Microtubules are shown in green. Aster invasion results in the fusion of cytoplasmic compartments and dynein-induced relocation of sperm nuclei. Cytoplasmic compartments are shown in magenta and nuclei in cyan with GFP-nuclear localization signal (NLS). **b**, High-resolution timelapse of an invasion event. **c**, Top view of 3D agent-based simulations of two asters showing the invasion process over time. **d**, Invasion time plotted against initial mass difference between the asters, showing that asters with small mass differences take longer to invade than asters with large mass differences (40 invasion events from $n = 20$ independent samples). Error bars are s.e.m. Cell-cycle time of the frog embryo[25] is plotted for comparison. Simulations are shown in grey ($n = 78$ independent simulations), experimental data in blue and theory in orange.

The error bars are s.d. **e**, Phase portrait of the average area of the compartments for arrested (magma; $n = 6$ independent samples) and cycling extract (viridis; $n = 6$ independent samples) normalized (norm) for the initial area equal to 1 (dashed line). $\tau = 8$ min. Although the area of compartments in arrested extract grows freely, the area of compartments in cycling extract oscillates and maintains a small size, even though some invasion events are present. $A$ is the compartment area. **f**, Zoomed graph of the phase portrait of the cycling extract. The average cell-cycle time varies from 39 to 65 min ($n = 6$ independent samples). **g**, Normalized average compartment area over time. Shades of blue refer to the different cell-cycle times reported in the caption of panel **h**. Experimental data are shown as dots and binned data with error bars (s.d.). $n = 6$ independent samples. **h**, Probability density function (PDF) of normalized average compartment area ($n = 6$ independent samples).

the theory (Fig. 2d, orange line) and simulations (Fig. 2d, grey points and Extended Data Fig. 7f–p). Asters with large mass differences invade in a few minutes. Asters with small mass differences – that represent the situation in living embryos where compartments are highly uniform – have an invasion time comparable with the cell-cycle time.

The time dependence of the invasion events allows the cell-cycle duration to prevent aster invasion, and therefore the runaway growth of asters, if compartments are similar in size. Conversely, for slow cell-cycle times, invasion events may lead to increasing differences between compartments that may amplify the instability, leading to divergent compartment size distributions. This process can be visualized by means of a phase portrait, showing that while in the arrested extract, the compartment size monotonically increases, whereas in the cycling extract, it oscillates around a characteristic compartment size, despite some invasion events (Fig. 2e,f). To further explore the effect of the cell-cycle duration on the compartment size, we systematically delayed the cell-cycle time by titrating cycloheximide amounts in extracts. These experiments showed that the cell-cycle duration directly affects

the average compartment size and therefore patterning of the cytoplasm (Fig. 2g and Supplementary Video 9). Moreover, we observed that although for shorter cell-cycle times the distribution of compartment sizes is peaked, as the cell cycle slows down, the compartment size distribution becomes increasingly broader (Fig. 2h). This monotonously increasing variation in compartment size is in contrast with the heterogeneity of stable compartment sizes (with bounded sizes) that naturally arises in embryos and cytoplasmic extracts as a function of the density of asters[3,29]. These results show that a delicate balance between the cell-cycle time and compartment growth is necessary to achieve a uniform and robust cytoplasmic partitioning.

## Microtubule dynamics regulate the instability

Our data show that changes in cell-cycle timing can have dramatic consequences in the precision of cytoplasmic partitioning, from extremely regular partitioning when matching autocatalytic growth, to system-size coarsening. We wondered whether there were regimes in

the parameter space that could prevent this instability, independently of the cell-cycle timing. To this end, we performed a linear stability analysis of equation (1) (Supplementary Note 2.1), leading to the stability criterion:

$$\theta > \frac{\alpha}{1 + 2\rho_{int}/\rho_s} \tag{2}$$

where $\rho_{int}$ is the density of microtubules where the two asters intersect. We confirmed this stability criterion by numerically solving equation (1) (Fig. 3a). The stability of the compartment boundaries critically depends on a competition between the autocatalytic rate and microtubule turnover, and not on the strength or the shape of the local inhibition term or dimensionality (Extended Data Fig. 2a–f and Supplementary Note 2.1). When the autocatalytic term dominates over turnover, microtubule density profiles feature exponential growth from the centre of the compartment (Fig. 3b, (2)). Although this density will go down as the asters interact, the boundary they form will always be unstable. Conversely, if turnover dominates over autocatalytic growth, the density of microtubules decreases from the centre of the compartment (Fig. 3b, (1)). In this regime, the boundary created as the two asters interact will be stable, but the compartments will be generally smaller with a size defined by the decay length scale of the microtubule density. Consistent with the instability that we measured, extracts fall in the unstable region of the phase diagram (Fig. 3a). These results show that the stability of compartments can be achieved by modulating microtubule nucleation and dynamics, independently of cell-cycle timing.

To investigate the possibility of stabilizing cytoplasmic compartments by changing microtubule dynamics, we fabricated asters with a decreasing microtubule density profile[13]. These asters can be obtained by adding Aurora kinase A antibody-coated (AurkA) beads to extracts[14,15,17] (Fig. 3c) instead of sperm nuclei. The AurkA beads act as artificial centrosomes. AurkA beads trigger the nucleation of microtubules[13,30], but to a lesser extent than chromatin-associated centrosomes. In this condition, we measured that the microtubule density profile decays from the beads (Fig. 3c), consistent with a decrease of microtubule nucleation and a stable system according to the theory. We confirmed this shift to the stable regime by measuring the turnover rates and polymerization velocity of microtubules and fitting the nucleation parameters. As expected, these asters fall into the stable regime of the phase diagram (Fig. 3a, orange area). We then tested whether the system is stable when the cell cycle is arrested. As predicted by the stability criterion, asters formed by addition of AurkA beads in arrested cytoplasm do not invade, and partition the cytoplasm with surprising regularity similar to asters in *Drosophila* extract and embryos[31] (Fig. 3d, Extended Data Fig. 8i–k and Supplementary Video 10). We showed that the stability does not depend on the size of these asters by using a lower bead concentration (Supplementary Video 10). These results are consistent with previous experiments performed in extract with AurkA beads[14,15,17]. Experiments with purified centrosomes from HeLa cells and *Drosophila* embryos in extract also led to stable asters (Extended Data Fig. 8c–g). To confirm that the effect on the stability was solely due to changes in microtubule nucleation, we supplemented extracts in the presence of AurkA beads with constitutively active Ran(Q69L) to increase microtubule nucleation[16] (Fig. 3e). In this situation, nucleators in the cytoplasm are activated and drive the formation of microtubule branches in AurkA bead asters. This branching results in the increase of the density of microtubules from the centre of the compartments similar to the case asters from chromatin-associated centrosomes. Moreover, cell-cycle arrest reveals that AurkA–Ran(Q69L) asters became unstable and invaded as in the sperm aster case (Fig. 3f and Supplementary Video 10), consistent with theory.

To further test the stability criterium, we aimed at perturbing intrinsically unstable asters by modifying microtubule dynamics. To this end, we added purified MCAK-Q710 (ref. 32) to extracts in the presence

of sperm DNA, as in the control situation. Previous work has shown that MCAK-Q710 alters microtubule turnover and dynamics in metaphase[33] and interphase[14]. We measured microtubule growth and turnover as in control, and observed a decrease of microtubule polymerization by approximately 20%, whereas turnover remained unchanged[14] (Supplementary Table 4). These results imply a change in microtubule length, which according to our model directly influences $\rho_s$ and $\alpha$. We then used the predicted change on these parameters while keeping the value of inhibition unchanged, to predict the density profile (up to the absolute value of the density close to the centre, which we fit). The spatial dependence of the density profile in both theory and experiments agree quantitatively (Fig. 3g), and the predicted changes in $\rho_s$ and $\alpha$ match the global fit to the profiles (Extended Data Fig. 8h). The change of these parameters also predicts a shift in the stability of these compartments: from unstable to stable (Fig. 3a). To test this prediction experimentally, we compared the aster organization over time with and without MCAK-Q710. We observed an overall smaller compartment size in the perturbed case due to a decrease in invasion events (Fig. 3i and Supplementary Video 10). In summary, robust compartmentalization of the cytoplasm can be achieved in a parameter regime where microtubule turnover dominates over autocatalytic nucleation rate, independently of the cell-cycle time.

## Divergent partitioning strategies

To investigate the in vivo relevance of the stability prediction, we turned to zebrafish and *Drosophila* embryos. We chose these embryos because of their drastically distinct aster structure despite a comparable embryo size (approximately 700 μm in diameter for zebrafish and approximately 500 μm for the long axis of *Drosophila*). In zebrafish embryos, the density of microtubules in interphase asters increases from the centrosome until microtubules reach the entire cell (Fig. 4a,e). By contrast, in *Drosophila* embryos, microtubule density decreases from the centrosomes (Fig. 4b,k) and microtubule asters do not reach the boundary of the whole syncytium (cortex of the embryo). These asters slowly fill up the embryo volume in subsequent cell divisions. On the basis of the theory and results in extract, we predicted that the cytoplasmic compartments in zebrafish should be unstable and by contrast stable in *Drosophila*. To test this prediction, we first confirmed where these embryos lie in the phase diagram (Fig. 4c). To this end, we quantified microtubule dynamics in embryos by measuring the polymerization velocity as the speed of plus ends, and the microtubule turnover as half-time recovery from photobleaching (for zebrafish) and photoconversion (for *Drosophila*) experiments. We estimated the parameters associated to autocatalytic growth and the local inhibition similarly to the data of microtubule asters in extract. In the stability phase diagram, zebrafish falls into the unstable region, whereas *Drosophila* lies in the stable region, consistent with the shape of the density profiles. The microtubule turnover that we measured in extracts, zebrafish and *Drosophila* is very similar, whereas the shift from the stable to unstable regime is mainly driven by changes in autocatalytic nucleation (Fig. 4d).

We next tested the stability prediction by arresting the cell cycle in interphase in vivo by adding cycloheximide and following the aster dynamics using live imaging. As predicted, cytoplasmic compartments in zebrafish embryos were unstable and invaded each other within 15 min (Fig. 4f, Extended Data Fig. 10a–c and Supplementary Video 12). The invasion leads to the fusion of the compartmentalized unpolymerized actin at the cortex, suggesting that aster invasion can affect organization beyond the cytoplasm (Fig. 4g). Although it is generally hard to observe the invasion process in zebrafish embryos because of the tight control of nucleation and aster size, if such control is perturbed, the competition of autocatalytic waves immediately drives invasion. We could enhance these invasions by nucleating smaller asters triggered by adding an exogenous oil droplet, by depolymerizing

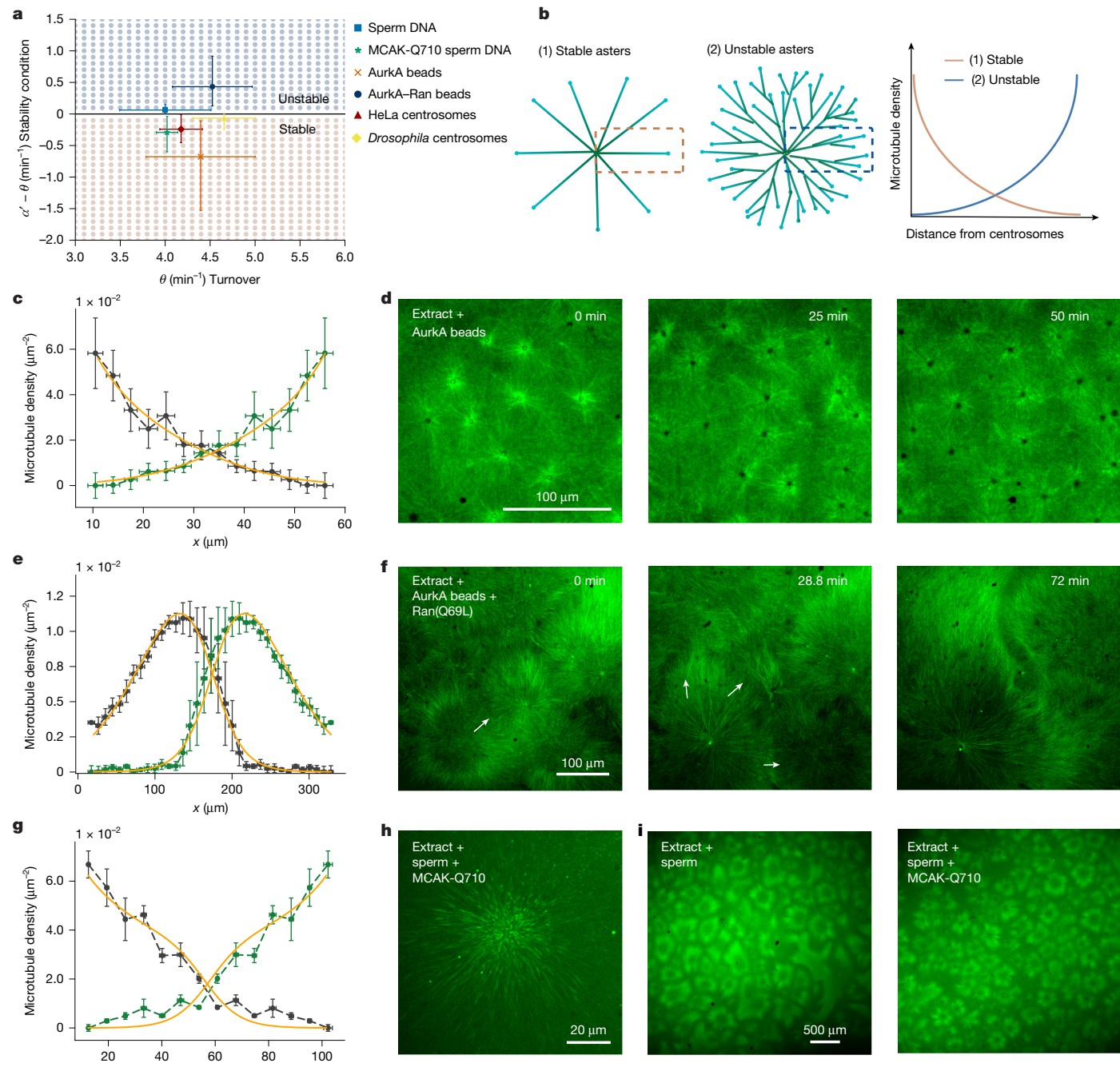

**Fig. 3 | Microtubule dynamics can regulate the stability of cytoplasmic partitioning. a**, Stability phase diagram of $\alpha' - \theta$ versus $\theta$ showing a stable and unstable region. The blue and orange dots correspond to numerical solutions of equation (1). The black line represents the stability criterion $\alpha' = \theta$. The error bars represent the 95% confidence interval of the mean. The numbers of independent tracks, lifetimes and regions of interest analysed are reported in Supplementary Tables 1–3. **b**, Schematics of microtubule asters and 1D density from close to the centre to the boundary. In stable asters, the microtubule density decreases from the centre. In unstable asters, the microtubule density increases from the centre because of the exponential nature of branching. **c**, Microtubule density profile of two AurkA asters measured as the density of plus ends of the microtubule. Experimentally measured profiles are shown in green and dark grey ($n = 7$ independent samples) and 1D global fit is in orange.

**d**, Confocal microscopy time sequence of AurkA asters in interphase-arrested cytoplasmic extract showing that the asters are stable and regularly partition the cytoplasm. **e**, Microtubule density profile of two AurkA–Ran(Q69L) asters. Experimentally measured profiles are shown in green and dark grey ($n = 5$ independent samples) and the 1D global fit is in orange. **f**, Confocal microscopy time sequence of AurkA–Ran(Q69L) asters in interphase-arrested cytoplasmic extract showing that the asters are unstable. **g**, Microtubule density profile of two sperm nuclei asters with MCAK-Q710. Experimentally measured profiles are shown in green and dark grey ($n = 5$ independent samples) and the 1D fit is in orange. **h**, Confocal microscopy image of a MCAK-Q710 sperm nucleus aster. **i**, Comparison of aster organization dynamics with and without MCAK-Q710, showing an overall smaller aster size in the perturbed case due to a decrease of invasion events. All error bars in the profiles are s.e.m. (**c**,**e**,**g**).

locally microtubules and by inhibiting dynein, which leads to asters in closer proximity to each other (Extended Data Fig. 10g–i and Supplementary Video 13).

Similar to extract, the aster invasion in the zebrafish shows relocalization of nuclei by dynein motors to the centre of the fused compartment. However, the confinement of asters within the spherical cap

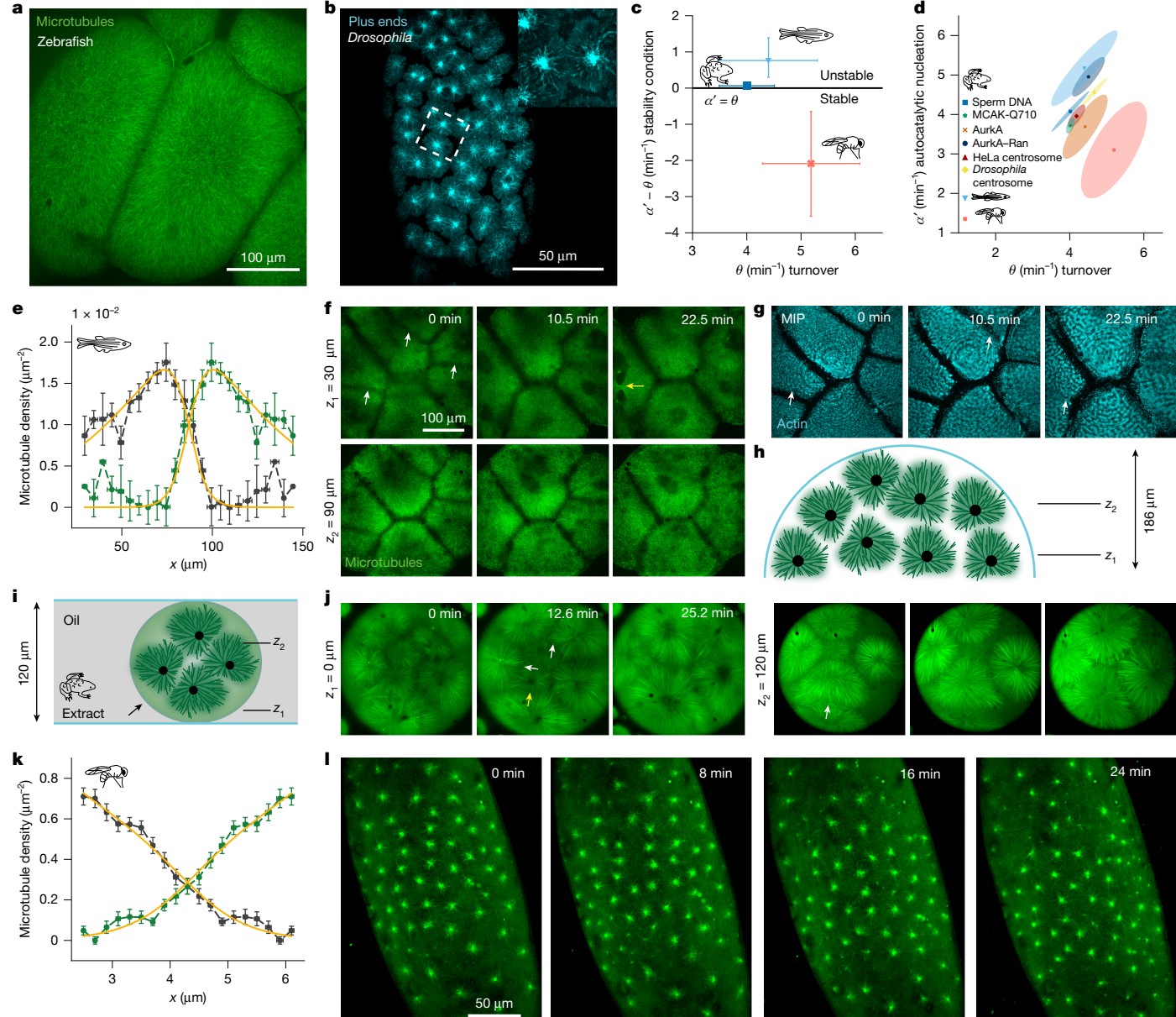

**Fig. 4 | Test of the (in)stability prediction in zebrafish and *Drosophila* embryos. a,b**, Confocal microscopy image of microtubule asters in zebrafish (**a**) and *Drosophila* (**b**) embryos with enlargement in the inset. Asters were visualized by a time projection over 20 frames of growing plus ends shown by EB1 (**b**). **c**, Phase diagram of $\alpha' - \theta$ versus $\theta$ for frog cytoplasm and zebrafish and *Drosophila* embryos. The black line represents the stability criterion $\alpha' = \theta$. Sample numbers are reported in Supplementary Tables 1–3. The error bars represent the 95% confidence interval of the mean obtained with bootstrapping (Extended Data Fig. 9f). **d**, Plot of $\alpha'$ versus $\theta$. The area under the ellipse represents the 95% confidence interval of the mean obtained with bootstrapping (Extended Data Fig. 9f). **e**, Microtubule density profile of asters of the zebrafish embryo. Green and dark grey show the experimental density profiles ($n = 4$) and orange is the global fit. **f**, Live confocal imaging of a zebrafish embryo treated with cycloheximide and cytochalasin B showing invasion events (white arrows).

Microtubules are shown in green at two different planes. At $z_1$, it is possible to observe the relocation of nuclei, indicated by the yellow arrow. **g**, Boundaries between the compartments at the cortex disappear over time (see Supplementary Video 12). MIP refers to maximum-intensity projection. **h**, Schematic of the compartments in the zebrafish embryo. **i**, Schematic of encapsulation of frog egg cytoplasmic extract in droplets. **j**, Timelapse of interphase-arrested frog egg cytoplasmic extract encapsulated in droplets. The droplet diameter is 460 μm. See Supplementary Video 7. **k**, Microtubule density profile of asters of *Drosophila*. Green and dark grey show the experimental density profiles ($n = 8$) and orange is the global fit. **l**, Live imaging of an interphase-arrested *Drosophila* embryo treated with cycloheximide and cytochalasin B showing aster stability (see Supplementary Video 12). The error bars in the microtubule profiles are s.e.m. (**e**,**k**).

of the embryo results in invasion dynamics that visually differ from those observed in the thin layer of extract (Extended Data Fig. 10d for embryo and Extended Data Fig. 7b for extract). To better mimic the embryonic geometry, we confined interphase-arrested extract in droplets and observed invasion patterns similar to those in the embryo (Fig. 4j, Extended Data Fig. 10e and Supplementary Video 7). Further-more, arresting embryos at the 32-cell stage − when aster geometry

resembles that of asters confined in a thin layer − resulted in similar invasion dynamics to the latter case (Extended Data Fig. 10b,f and Sup-plementary Video 12).

In contrast to zebrafish embryos, compartments in *Drosophila* remained stable, reminiscent of the compartments formed in extracts with AurkA beads (Fig. 4l and Supplementary Video 12). Because AurkA beads resemble the compartmentalization of *Drosophila* embryos,

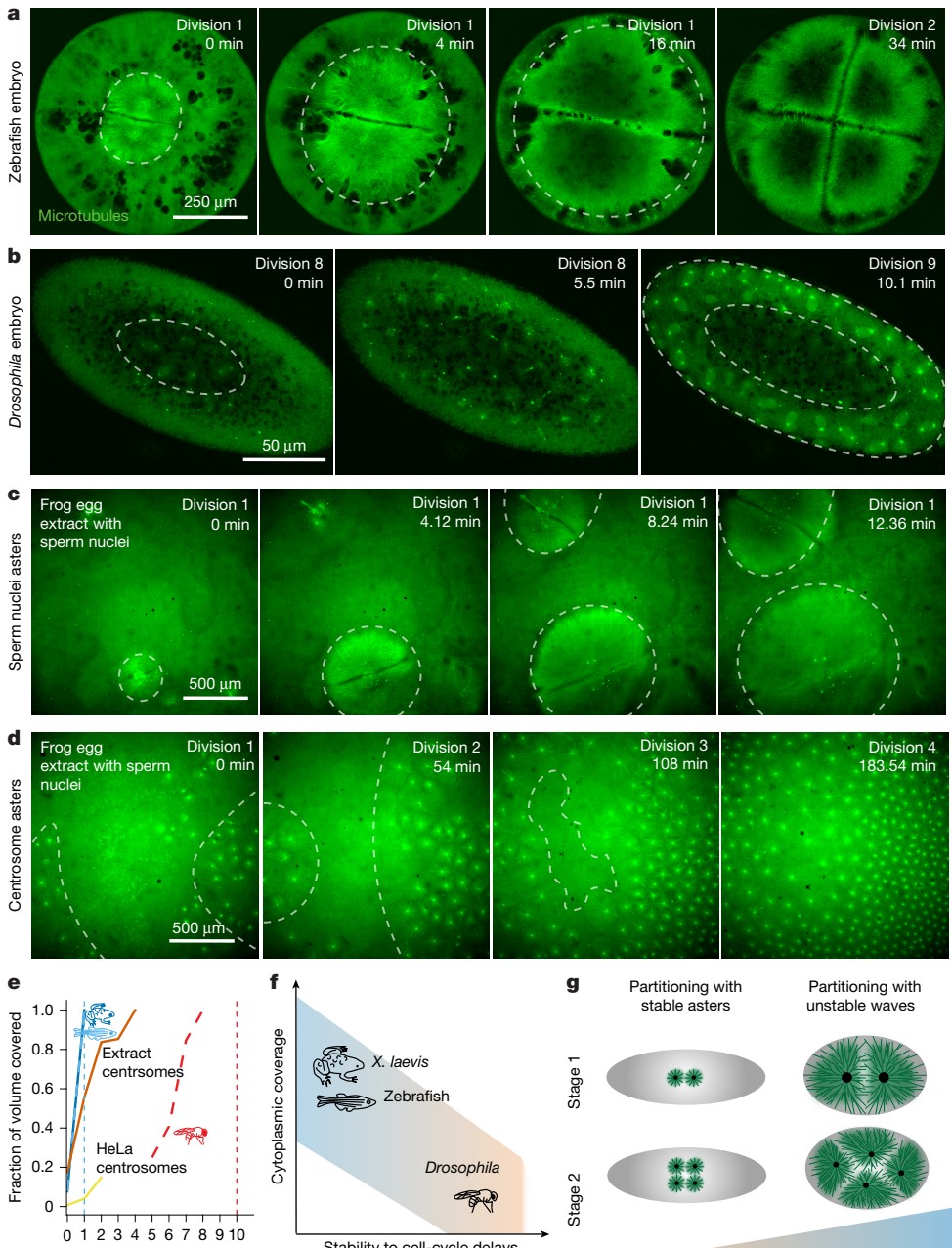

**Fig. 5 | Divergent strategies of cytoplasmic partitioning driven by regulation of microtubule self-amplification. a**, Timelapse of microtubule asters filling the cytoplasm in a zebrafish embryo during the first two divisions. The grey dashed lines indicate the area for the cytoplasm occupied by the asters. **b**, Timelapse of asters filling the cytoplasm in a *Drosophila* embryo during divisions 8 and 9. **c**, Timelapse of asters in cytoplasmic extract supplemented with sparse sperm nuclei. **d**, Microtubule asters originating from centrosomes in extract with slower cell-cycle time. This extract sample was supplemented with sperm nuclei, but the asters originate from centrosomes that are not associated with nuclei. See Supplementary Video 14 for time sequences. **e**, Plot of the cytoplasmic volume occupied by the asters over time. Data for the zebrafish embryo (light blue dashed line), the frog egg extract with sparse sperm nuclei (dark blue line), the egg extract centrosomes (orange line) and purified HeLa centrosomes added to extract

(yellow line) taken from time sequences in Fig. 5a,c,d, Extended Data Fig. 9 and Supplementary Video 14. Data for the *Drosophila* embryo (red dashed line) were taken from Deneke et al[38]. The light blue and red vertical dashed lines indicate when cellularization starts for the zebrafish and *Drosophila* embryos. **f**, Phase diagram of cytoplasmic organization. Embryos during early embryogenesis regulate autocatalytic growth to organize the cytoplasm before cellularization. In frog and zebrafish embryos, cellularization begins at the first division. Asters rapidly fill the cytoplasm, exploiting a high level of autocatalytic nucleation that leads to an instability that requires precise cell-cycle control. In *Drosophila* embryos, cellularization occurs after 13 divisions. Asters are small, possess low autocatalysis and progressively organize the cytoplasm in a stable manner. **g**, Schematic of these divergent strategies to organize the cytoplasm.

we wondered whether changing microtubule nucleation alone not only dictates the stability of the compartments but also the dynamics of organization of the entire cytoplasmic volume as in *Drosophila* embryos. To test the 'drosophilization' of the extract, we looked for

regions in the cytoplasm where there were only centrosomes in the absence of chromatin-associated centrosomes. The centrosome asters had similar profiles to the AurkA beads and *Drosophila* embryos. These asters progressively filled the volume as they divided, similarly

to *Drosophila* embryos (Fig. 5b,d and Supplementary Video 14), and with stark contrast to the complete covering of the whole cytoplasm in control extract and zebrafish embryo during each cell cycle (Fig. 5a,c and Supplementary Video 14).

Together, our data show that our stability criterion can predict the dynamics of divergent compartmentalization strategies in vivo, which can be explained by tuning the amount of autocatalytic microtubule nucleation. In frog and zebrafish, microtubule asters grow with high autocatalytic nucleation, which leads to large asters that can reach the embryo boundary and therefore cover the whole embryo cytoplasm from the first-cell stage, but are unstable. In *Drosophila*, where asters possess low autocatalytic nucleation, compartments are stable but small, and fill the cytoplasm over multiple divisions, leading to lower cytoplasmic coverage (Fig. 5e–g).

## Discussion

How cells establish and maintain their individuality is a fundamental question in biology. Although it is often assumed that multicellular organisms achieve compartmentalization through cytokinesis during development, many organisms undergo extended developmental stages in which nuclear division occurs without cellularization. In these syncytial contexts, compartmentalization depends not on membranes, but on the spatial organization of the cytoplasm itself. This phenomenon was recognized as early as the nineteenth century, when Sachs described discrete cytoplasmic domains around individual nuclei as 'energids'[34]. However, the mechanistic basis of such organization has remained largely unexplored.

Here we addressed how cytoplasmic partitioning is achieved in the absence of membranes by focusing on the role of microtubules emerging from centrosomes. Using a combination of theory describing aster–aster interactions, in vitro reconstitution in *Xenopus* egg extracts and in vivo imaging in zebrafish embryos, we revealed that cytoplasmic partitioning before cytokinesis is an intrinsically unstable process in large vertebrate embryos. This instability stems from a competition between microtubule autocatalytic growth and turnover. Despite this inherent instability, we have demonstrated that precise cell-cycle timing renders this compartmentalization dynamically stable, resulting in remarkably robust partitioning of the cytoplasm. To find the proper geometric centre, cells read the geometrical boundaries of the embryo using unstable microtubule waves that reach the cortex[11,35]. This instability imposes a delicate balance between the waves of autocatalytic growth and the cell-cycle timing. The cell-cycle duration needs to be slow enough for waves to read the geometry of the cell but fast enough that compartments do not fuse. The cell cycle also needs to be synchronous across compartments to avoid invasion events, as seen in zebrafish embryos and *Xenopus* egg extracts. This synchronization is fundamental at the beginning of development when the timescale of completing membrane ingression is longer than a single cell-cycle duration. As development progresses and ingression of the cell membrane is completed, the cell membrane can block invasion events and ensure correct multicellularity. In organisms that do not require early cellularization, as in syncytial *Drosophila* embryos, it is not necessary to immediately read the embryo geometry. Instead, smaller and stable asters progressively divide and compartmentalize the embryo space before cellularization. In this situation, there is no need to rely on unstable autocatalytic processes or to have a perfectly synchronized cell cycle as the compartments remain stable. This mechanism provides greater flexibility over geometry, timing and directionality of cytoplasmic exploration. This strategy is especially advantageous in insect embryos, where the progression of aster-driven filling is universal but adapted across species, with variable timing, speed and paths[36]. For example, in *Drosophila*, correct positioning of asters at the cortex depends on cytoplasmic flows[37–39]. The gradual nature of this partitioning strategy may also allow for localized biochemical patterning.

The stability of asters in *Drosophila* embryos supports a developmental program without the need for strict spatial or temporal synchrony of divisions, a feature shared with other syncytial organisms[36]. Such stability is important because in some cases, asynchronous divisions may even offer a selective advantage: for example, in the malaria parasite *Plasmodium falciparum*, asynchronous nuclear divisions in a shared cytoplasm are thought to enable rapid proliferation before cellularization[40]. Overall, stability of asters observed in *Drosophila* embryos may have allowed the evolutionary maintenance of multinucleated development across diverse species.

Our study underscores that the diverse compartmentalization behaviours observed across species can be explained by the interplay between microtubule turnover and nucleation. As turnover remains conserved among the species examined, our findings suggest that evolutionary changes in microtubule nucleation may contribute to the diverse cytoplasmic partitioning strategies across species. These findings are crucial not only in the context of embryonic development but also for syncytial systems and cytokinesis. In syncytial systems, where cytoplasmic compartments lack cell membrane separation, mechanisms regulated by the cytoskeleton are essential for maintaining distinct borders[41]. Similarly, during cytokinesis, cells briefly become syncytial and must sustain separate cytoplasmic compartments until cytokinesis is completed[4,9].

This work presents a novel integration of general reaction–diffusion mechanisms with biological oscillators, contributing to the understanding of pattern formation dynamics. We explored a network characterized by local self-amplification (autocatalytic growth of the asters) and local inhibition. This network is unstable; however, when properly modulated with the cell-cycle oscillator, it gives rise to dynamically stable and robust states. This combination of unstable networks with oscillators unlocks a realm of previously unexplored unstable regimes, yielding dynamically stable patterns endowed with remarkable traits such as rapid spatial coverage and flexibility.

Overall, our research exemplifies how precise temporal tuning of biological oscillators can govern spatial patterning and cellular organization[41,42], highlighting how physical and geometrical constraints influence the evolution of self-organization mechanisms.

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

# Methods

## Husbandry of experimental animals

Female frog (*X. laevis*) adults and zebrafish adults were maintained and handled according to established protocols[43,44]. Frogs were acquired from Nasco (LM00531) and Xenopus1 (4800). The experiments were approved and licensed by the local animal ethics committee (Landesdirektion Sachsen; license no. DD24-5131/367/9, 25-5131/521/12 and 25-5131/564/25 for frogs II and license no. DD24.1-5131/394/33 for zebrafish) and carried out following the European Communities Council Directive 2010/63/EU on the protection of animals used for scientific purposes, as well as the German Animal Welfare Act. *Drosophila* stocks were maintained at room temperature using standard methods.

## Zebrafish and *Drosophila* transgenic lines used

The following transgenic zebrafish lines were used. Tg(actb2:EGFP-Hsa. DCX) for microtubule visualization, Tg2(actb2:mCherry-Hsa.UTRN) for actin visualization, their double-transgenic Tg(actb2:EGFP-Hsa. DCX; actb2:mCherry-Hsa.UTRN) and Tg(Xla.Eef1a1:mlsEGFP)[45] for mitochondrial compartmentalization. The following fly lines were used. For transgenics: wild-type (w[1118]), Pw[+mC]=PTT-GAJupiter[G00147] (BDSC 6836; FlyBase: FBst0006836) expressing Jupiter–GFP, Pw[+mC]=His2Av-mRFPII.2* (BDSC 23651; FlyBase: FBst0023651) expressing histone–RFP, and w[1118]; Pw[+mC]=osk-GAL4::VP16F/ TM3, Sb[1] (BDSC 44242; FlyBase: FBst0044242), used as a maternal Gal4 driver for UAS lines. A double-transgenic line (yw; His2Av-EGFP/ CyO; TMBD-mCherry/TM6B, Tb) was used to visualize histones and microtubules. A EB1–GFP line w[1118]; Pw[+mC]=ncd-EB1. GFPM1F3 (BDSC 57327; FlyBase: FBst0057327) was used to track EB1 comets. A photoconvertible α-tubulin under UAS line was used for photoconversion: w[*]; Pw[+mC]=UASp-alphaTub84B.tdEOS7M (BDSC 51314; FlyBase: FBst0051314).

## Cytoplasmic extract and cell-cycle manipulations

Cytoplasmic extracts from frog eggs were prepared following standard protocols[46,47]. The following buffers were prepared in advance: Ten times Marc's modified Ringer's (MMR; 1 M NaCl, 20 mM KCl, 10 mM $MgCl_2$, 20 mM $CaCl_2$, 1 mM EDTA and 50 mM HEPES in milliQ water), with pH was adjusted to 7.8 with NaOH and the solution was filter sterilized and stored at room temperature; 20× *Xenopus* buffer (2 M KCl, 20 mM $MgCl_2$ and 2 mM $CaCl_2$ in MilliQ water), with pH adjusted to 7.7 with KOH; 1 M HEPES solution was prepared and pH was adjusted to 7.7 and after filter sterilization, the solution was stored at 4 °C; and a 2 M sucrose solution. The solutions were filter sterilized and stored at 4 °C. Female adult frogs were injected with 0.5 ml of pregnant mare serum gonadotropin (779-675, Covetrus) and 0.5 ml of human chorionic gonadotropin (CG10-10VL, Sigma) 3–8 days and 1 day before the experiment, respectively. After the second injection, frogs were incubated at 16 °C for 18 h in a 1× MMR solution. Of MilliQ water, 3 l was incubated at 16 °C to be used for buffer preparation. On the experiment day, the following buffers were prepared: 1 l of dejelly buffer (20 g of L-cysteine (W326305, Merck), 50 ml of 20× *Xenopus* buffer and MilliQ water), with pH adjusted to 7.8 with NaOH; 1 l of 0.2× MMR, with pH adjusted to 7.8 with NaOH; 1 l of 1× Xenopus buffer (20× *Xenopus* buffer in MilliQ, 50 mM sucrose, 10 mM 1× HEPES), with pH adjusted to 7.7 with KOH; 100 ml of calcium ionophore solution (CaIo; 5 µl of calcium ionophore (A23187, Sigma) in 100 ml of 0.2 MMR buffer); Xenopus buffer+ (100 µl of 10 mg ml⁻¹ solution of leupeptin (15483809, Thermo Scientific), pepstatin (2936.2, Roth) and chymostatin (230790, Calbiochem) in 1× *Xenopus* buffer). Frog eggs were dejellyed by multiple washes with the dejelly buffer. Eggs were then washed multiple times with 1 l of 0.2 MMR buffer and then incubated in the CaIo solution to activate the cell cycle. During the dejelly and activation, eggs were swirled to achieve uniform contact with chemicals and avoid egg aggregation. The activation process lasted 3–5 min, depending on egg number, and was continued until the animal pole became smaller and darker. The eggs were then washed multiple times with 1 l of *Xenopus* buffer and then 3× with 250 ml of *Xenopus* buffer+. Next, eggs were transferred to centrifuge tubes (344057, Beckman) containing 1 ml *Xenopus* buffer+ and 10 µl of cytochalasin B (15466849, Thermo Scientific). Tubes were sequentially centrifuged for 30 s at 500 RPM and for 1.5 min at 2,000 RPM for egg packing. After the excess buffer was removed with an aspirator, eggs were crushed by centrifugation for 15 min at 10,000 RPM at 4 °C. The centrifuge tubes were placed on ice and the cytoplasmic layer was collected by puncturing the tube. Additional LPC (leupeptin–pepstatin–chymostatin) (1/1,000 w/v) and cytochalasin B (1/1,000 w/v) were added to further prevent protein degradation and actin polymerization. The extract was stored in ice and used on the same day. Interphase-arrested extract was obtained as described above but 200 µl of cycloheximide (239763, Merck; 10 mg ml⁻¹) were added to the centrifuge tubes with the *Xenopus* buffer+.

## Extract sample preparation and imaging

The extract was supplemented with de-membraneted sperm to induce nuclei formation and fluorescent labels for visualizing cellular structures. Reactions were set up by mixing 25 µl of undiluted extract, 0.6 µM pig tubulin labelled with 647 Alexa fluorophore[48], 0.12 mg ml⁻¹ GFP-NLS, 0.2 µl of 1:250 diluted stock in water of octadecyl rhodamine B chloride (O246, Invitrogen) and 1 µl of sperm (3,000 sperm per microlitre) in an Eppendorf tube in ice. Reactions were supplemented with the following: anti-INCENP (0.5 µl of antibody (ab12183, abcam) labelled with Alexa Fluor 488 NHS Ester (A20000, Thermo Fisher) instead of GFP-NLS), AurkA beads (1 µl)[30], Ran(Q69L)[16] (30 µM), MCAK-Q710 (1 µl of 1.5 mg ml⁻¹ added). Dynein inhibitor p150-cc1 (concentrations reported in Extended Data Figs. 4 and 5), barasertib (40 µM; S1147, Selleckchem), purified centrosomes from *Droosophila* embryos (HisGFP-TauMCherry line) and HeLa cells were prepared using existing protocols[49,50] (3 µl added to the reaction). The treatment with morpholino to selectively block translation of cyclin B1 and B2 and arrest the cell cycle was performed by mixing 2.5 µl of each the following morpholinos in a solution and then adding 1–5 µl of it to the extract reaction: morpholino anti-*Xenopus* cyclin B1 (*ccnb1a*): ACATTTTCCCAAAACCGACAACTGG; morpholino anti-*Xenopus* cyclin B1 (*ccnb1b*): ACATTTTCTCAAGCGC AAACCTGCA; morpholino anti-*Xenopus* cyclin B2 (*ccnb2l*): AATTG CAGCCCGACGAGTAGCCAT; and morpholino anti-*Xenopus* cyclin B2 (*ccnb2s*): CGACGAGTAGCCATCTCCGGTAAAA. The morpholinos were acquired by custom order to Gene Tools and the sequences were chosen from previous works[27,28]. For slowing the cell cycle, extract reactions were supplemented with 4 µl of cycloheximide in varying concentrations from 2.5 g l⁻¹ to 7.5 g l⁻¹. We could not link a specific concentration to a cell-cycle duration, probably because the amounts of cyclins vary from sample to sample. In all the experiments, the reactions were flicked multiple times and left for 3–5 min in ice to homogeneously distribute the reagents in the extract. Of the reaction, 6 µl was taken from ice, and added either on a 35-mm glass bottom dish (P35G-0.170-14-c, MatTek) and covered with 1 ml of mineral oil (m3516, Sigma) or a 15 µ-Slide eight well (80826, Ibidi) and covered with 300 µl of anti-evaporation oil (50051, Ibidi). The oil was necessary to allow oxygen exchange and viability of the sample for long-term imaging. For the experiments of droplet confinement (Fig. 4j), 10 µl of the reaction with morpholinos was added droplet by droplet in a tube with 0.5 ml mineral oil. The tube was flicked three times to break the droplets into smaller droplets. The reaction droplets in oil were then transferred to a coverslip with a spacer (GBL654004, Grace Bio-Labs) and a second coverslip was used to seal the top. During this process, some air droplets formed in the oil, allowing for oxygen exchange. A well with a different coating that allows for imaging at low magnification with higher resolution was used for Supplementary Videos 4 and 10 in the

MCAK case (80800, Ibidi). Imaging was performed with a spinning disk confocal microscope (IX83 Olympus microscope with a CSU-W1 Yokogawa disk) connected with two Hamanatsu ORCA-Fusion BT Digital CMOS camera (SD1). *Z*-stacks were acquired with a stage-top Z piezo and 10–20-μm *z*-spacing.

## FRAP, EB1 and speckle imaging in extract

Of cytoplasmic extract, 25 μl was supplemented with 1 μl of sperm (3,000 sperm per microlitre), 0.6 μM of pig tubulin labelled with 647 Alexa fluorophore and 0.16 μM EB1–mApple[22] to image the plus ends of the microtubules. The sample was imaged with SD1 and Olympus ×40 Air (0.65 NA) objective, and extract supplemented with beads was imaged with Olympus ×100 (1.35 NA) silicon oil. Every 2 min, we sequentially acquired five images of EB1 comets 3 s apart followed by one image of tubulin, following existing protocols[14]. This choice for the framerate allowed us to minimize bleaching of EB1 and follow the EB1 tracks over time. These reactions were also used for FRAP experiments in the case of extract supplemented with purified centrosomes. The FRAP experiments were performed with SDType1 equipped with a photoactivation module, an Olympus ×40 Air (0.65 NA) objective and a 405-nm laser. For speckle imaging, cycling extract was supplemented with 10 nM of Atto 567-labelled tubulin. The extract was encapsulated between two coverslips separated by a spacer (GBL654004, Grace Bio-Labs) to lower the background noise and prevent aster movement. Because the encapsulation restricts oxygen exchange, the reaction was imaged for 15 min. The speckles were imaged with SDType1 and an Olympus ×60 (1.3 NA) silicon oil objective with low laser intensity and at least 2-s exposure time.

## Zebrafish embryo sample preparation

Embryos were collected in E3 medium (5 mM NaCl, 0.17 mM KCl, 0.33 mM $CaCl_2$ and 0.33 mM $MgSO_4$) within 15 min after spawning and kept at 24–28 °C. Embryo clutch quality was inspected on a dissection stereomicroscope and staged according to morphological criteria[51]. Embryos were mechanically dechorionated and mounted in 1% low melting-point agarose (A9414, Sigma-Aldrich) in E3 medium supplemented with 25% w/v iodixanol (OptiPrep, 07820, STEMCELL Technologies) for refractive index matching in a CELLVIEW cell culture glass bottom dish (627860, Greiner bio-one). Embryos were brought closer to the coverslip surface by keeping the dish upside down until agarose solidified.

## Syncytial embryo and cell-cycle arrest

Dechorionated embryos at the one-cell stage were treated with 10 μg ml$^{-1}$ cytochalasin B, which prevented cytokinesis. The stage of the embryo could be determined by minutes post-fertilization and by comparison with control embryos. When cytochalasin B (15466849, Thermo Scientific) treated embryos had reached the four-cell stage, they were mounted in 1% low-melting-point agarose containing 25% OptiPrep density gradient medium and supplemented with 200–400 g ml$^{-1}$ cycloheximide and 10 μg ml$^{-1}$ cytochalasin B in a glass-bottom dish (627860, Greiner bio-one). For the microtubule depolymerizing drug SbTubA4P[52], when cytochalasin B-treated embryos had reached the 4–8-cell stage, they were mounted in 1 ml of 0.5% low-melting-point agarose supplemented with 50 μl cycloheximide (10 mg ml$^{-1}$ stock), 1 μl cytochalasin B (10 mg ml$^{-1}$ stock) and 5 μl SbTub (10 mM stock activated with blue light).

## Embryo injections

Droplets were injected into the cytoplasm of one-cell-stage transgenic zebrafish embryos[53]. Injection volumes were calibrated to 0.5 nl. Depending on the experiment, 0.5–1 nl was injected per embryo. Ferrofluid droplets[54] were injected without magnetic activation. Embryos were injected with PH-Halo (plextrin homology domain of

PIP2: 4.64 mg ml$^{-1}$ tagged with JF646 (2 μl PH protein and 2 μl JF dye). Embryos were incubated for a few minutes to allow the injection wound to heal and then manually dechorionated and mounted in cytochalsin B containing LMP. Dynein inhibitor p150-cc1 1 nl of 35 mg ml$^{-1}$ inhibitor was injected into each embryo.

## Imaging of zebrafish embryos

For Fig. 1a,b, a zebrafish embryo with chorion was positioned in an agarose column and imaged with a Zeiss LightSheet Z.1, following existing protocols[55]. For Fig. 4f,g and Extended data Fig. 10i, embryos were imaged with a spinning disk confocal microscope (IX83 Olympus microscope with CSU-W1 Yokogawa disk) connected with two iXon Ultra 888 monochrome EMCCD cameras (Andor; SD2). A ×30 NA 1.08 silicone oil objective (Olympus) was used. *Z* stacks were acquired with a stage piezo and 1–2 μm *z*-spacing. In all the other figures, imaging was performed with SD1 and an Olympus ×20 (0.80 NA) air objective. In all experiments, the embryos were kept at 28 °C with a temperature incubator.

## FRAP and EB1 in zebrafish embryo

For EB1 experiments, cbg5Tg embryos were prepared and mounted as described above. The embryos were injected at the one-cell stage with 1.5 nl of a solution comprising 1.5 mg ml$^{-1}$ EB1–mApple in *Xenopus* buffer. Embryos were imaged with SD1 equipped with an Olympus SApo ×60 (1.3 NA) silicon oil objective. Every 2 min, we acquired five consequent images separated by a time delay of 3 s, to avoid bleaching. For FRAP experiments, double-transgenic embryos were prepared and mounted as described above. The embryos were injected at the one-cell stage with 1.5 nl of a solution comprising of 2 μl of pig tubulin labelled with Alexa 647 and 8 μl of *Xenopus* buffer. The embryos were imaged with an Andor IX 81 microscope through a ×10 0.4 NA Air Olympus objective and a Yokogawa CSUX1 spinning disk (SD3). The FRAP experiments were performed with the FRAP module FRAPPA and a 640-nm solid-state laser.

## *Drosophila* embryo preparation and imaging

Before imaging, w1118 males and females of the genotype of interest were housed in a cage covering an apple juice plate at 25 °C, supplemented with yeast paste. Embryos were collected over 2 h on a fresh plate, dechorionated with 50% bleach for 1 min and mounted in Halocarbon oil 27 (9002-83-9, Sigma) on a gas-permeable membrane for imaging. For the cycloheximide treatment, embryos were soaked in 10% concentrated cleaner and degreaser (Citrasolv) for 1 min after dechorionation and mounted in 1 mM cycloheximide (97064-724, VWR) for imaging. For the last two movies in Supplementary Video 12, on the left, the embryos were imaged on a Leica SP8 laser scanning confocal microscope equipped with a Leica ×20 oil-immersion objective (0.75 NA); on the right, the embryos were imaged with a Leica SP8 microscope using a Leica ×63 oil objective (1.40 NA).

## EB1 and photoconversion in *Drosophila*

For EB1 comet tracking, embryos of the EB1–GFP line were mounted as described above and imaged with SD1 and an Olympus ×100 silicon oil objective (1.35 NA). Embryos were imaged for minutes with a frame rate of 1 s. For the photoconversion experiments, embryos of the photoconvertible tubulin were mounted as described above. Photoconversion experiments were performed on the Leica SP8 microscope with the FRAP module in the Leica Application Suite X (LAS X). Experiments were acquired using a Leica ×63 oil objective (1.40 NA) and a 405-nm bleaching laser.

## Injection experiments in *Drosophila*

The embryo preparation follows the description above with the changes: males and females of the same genotype (HisGFP-TauM-Cherry line) were crossed and collection was performed for 30 min

to obtain embryos as early as possible. After collection and dechorionation, 10–30 embryos were aligned on an agarose plate with a brush. They were then positioned on a glass coverslip covered with a thin layer of heptane glue and equipped with two spacers on top of each other (SecureSeal, Grace Bio-Labs). Embryos were desiccated in a box with dehydrating beads (Drierite desiccants, w.a. Hammond Drierite) for 7 min and then covered in Halocarbon oil. After desiccation, embryos were injected under a stereo-microscope with a glass needle and 5–10% of the embryo volume. Concentrations at the needle were 0.1 mg ml$^{-1}$ for cycloheximide and 0.004 mg ml$^{-1}$ for cytochalasin B. The sample was then covered with a glass coverslip and the embryos imaged with SD1 and an Olympus ×60 (1.3 NA) silicon oil objective.

## Statistics and reproducibility

We chose sample sizes based on similar datasets used in the field, consistency of phenotypes and experimental challenges. Experiments were replicated over about 3 years with different microscopes and for some conditions different experimenters. The number of biological replicates is indicated as the number of independent samples and it refers to independent experiments for the in vitro extract studies and embryo number for in vivo studies. We have reported these numbers in the legend for plots with error bars and histograms. For the microtubule density profiles (Figs. 1l, 3c,e,g and 4e,k and Extended Data Fig. 8b,e,f,h), the number of independent samples are reported on Supplementary Table 1. For measurements of microtubule dynamics (Figs. 3a and 4c,d and Extended Data Fig. 9c), they are reported on Supplementary Tables 1–3. Technical replicates are reported in the adjacent columns. Here we have provided the number of biological replicates for images representative of phenotypes: for Fig. 1b, imaging of the microtubules and actin first in the cell cycle in the zebrafish embryo was repeated $n > 20$ times with confocal spinning disk and $n = 1$ with light sheet for visualization purposes; for Fig. 1c, $n = 4$; for Fig. 1d, $n = 8$; for Fig. 1e,f, $n > 20$; for Fig. 1g, $n = 4$; for Fig. 1h, $n = 8$; and for Fig. 1i,j,k, $n > 20$. For Fig. 2a,b, invasion events were $n > 20$. For Fig. 3d,f, $n > 5$; For Fig. 3h, $n = 8$; and for Fig. 3i, $n = 4$. For Fig. 4a,b,f,l, $n > 20$; and for Fig. 4j, $n = 1$ specifically with confinement in droplets, but $n > 20$ to test morpholinos in extract. For Fig. 5, the specific videos were acquired as $n = 1$ as proof of concept; however, these dynamics were observed $n > 20$. For Extended Data Fig. 1b, $n = 8$; for Extended Data Fig. 1c, $n = 4$; for Extended Data Fig. 1e, $n = 1$; for Extended Data Fig. 1f, $n = 12$; for Extended Data Fig. 1h,i, $n > 20$; and for Extended Data Fig. 1k, $n = 8$. For Extended Data Fig. 2g, $n > 20$; and for Extended Data Fig. 2h, $n = 3$. For Extended Data Fig. 4a,e, $n > 5$; and for Extended Data Fig. 4d, $n = 3$. For Extended Data Fig. 5, $n > 5$. For Extended Data Fig. 6, $n > 20$ invasion events. This is a representative analysis of the event. For Extended Data Fig. 6e, $n > 5$; and for Extended Data Fig. 6f, $n = 3$. For Extended Data Fig. 7a, $n > 20$ invasion events. This is a representative analysis of the event to show the method to find the invasion time. For Extended Data Fig. 8c,d, $n = 6$; for Extended Data Fig. 8g, $n = 2$; and for Extended Data Fig. 8i,j, $n > 5$. For Extended Data Fig. 10a, $n = 11$; for Extended Data Fig. 10b, $n = 12$; for Extended Data Fig. 10c, $n = 9$; for Extended Data Fig. 10g, $n = 8$; for Extended Data Fig. 10h, $n = 14$; and for Extended Data Fig. 10i, $n > 5$. Plots showing single lines without errors are relative to specific images or Supplementary Videos (Fig. 5e in the main text and Extended Data Figs. 1b,d,g,j, 2c, 4d,g, 6b–e, 7b–d and 10d,e). These plots are based on the quantification of a single example of a phenotype that was replicated with the $n$ value related for the figures and reported above. In the analysis of the role of cell-cycle times (Fig. 2g,h) and invasion times (Fig. 2d), experiments were analysed in a random order as there was no previous knowledge on the possible outcome. For the other experiments, randomization was not performed and each condition was analysed separately. We did not perform blinding. Data were excluded if embryos or extract underwent early apoptosis.

## Reporting summary

Further information on research design is available in the Nature Portfolio Reporting Summary linked to this article.

## Data availability

There is no restriction on data availability. Raw data supporting the findings of this study and high-resolution images and videos have been deposited on the online repository (https://doi.org/10.25532/OPARA-971). Source data are provided with this paper.

## Code availability

There is no restriction on code availability. The code for performing the simulations has been deposited on the online repository (https://github.com/lamsoa729/AsterInvasion). Scripts for the data analysis have been provided together with the raw data on the online repository (https://doi.org/10.25532/OPARA-971).

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

**Acknowledgements** M.R. acknowledges funding from the Human Frontier of Science (postdoctoral cross-disciplinary fellowship LT000920/2020-C) and the European Molecular Biology Organization (postdoctoral fellowship EMBO ALTF 597-2021). J.B. acknowledges funding from the European Research Council (ERC consolidator grant SynthNuc, 101045468). J.B., M.R., A.K. and A.L. acknowledge support from the Deutsche Forschungsgemeinschaft (DFG; German Research Foundation) under Germany's Excellence Strategy — EXC-2068-390729961 — Cluster of Excellence Physics of Life of TU Dresden. B.D. acknowledges the European Research Council (ERC) advanced grant 835117 NoMaMemo and HPC Service of ZEDAT, Freie Universität Berlin, for providing computing time. Y.X. and S.D.T. acknowledge funding from the NIH to S.D.T. (R01-GM122936 and R35-GM153490). We thank M. Elsner for culturing the HeLa cells, purifying HeLa and *Drosophila* centrosomes, and labelling the incenp antibody; K. Ishihara for providing the Ran(Q69L) protein, the MCAK-Q710 protein, the p150-cc1 protein and the AurkA antibody; O. Thorn-Seshold for providing SbTubA4P; O. Campas for providing ferrofluid; J. Rodenfels for sharing the mitochondria zebrafish line; H. Andreas (MPI-CBG, frog facility) for maintaining the frogs, the fish facility (MPI-CBG) and the light microscopy facility (MPI-CBG) for support with the microscopy imaging; U. Ursic for coating the AurkA beads; the Scientific Computing Core facilities at the Flatiron Institute, a division of the Simons Foundation, for running the computations for the 3D model; J. Eichhorn for the illustrations of the model organisms; A. Mukherjee for initial discussions; D. Needleman, M. Loose, B. Vellutini, M. Ebisuya and M. Riedl for input on the manuscript; and M. Marass for providing editorial advice during the publication process.

**Author contributions** M.R. conducted the experiments in the frog egg extract. A.K. performed zebrafish embryo preparation in all experiments and imaging for Figs. 1b and 4f and Extended Data Fig. 10i, related videos, FRAP and part of the EB1 imaging in zebrafish embryos. Y.X. performed *Drosophila* embryo preparation, photoactivation and imaging for Supplementary Video 12. P.M. performed embryo preparation for the rest of the *Drosophila* experiments. M.R. performed imaging of zebrafish embryos, and injections and imaging of *Drosophila* embryos for all remaining figures expect those done by A.K. and Y.X. M.R. analysed all the experimental data. A.L. conducted 3D agent-based simulations, performed theoretical global fits to experimental data with bootstrapping and provided the 3D continuum model. B.D. conducted 2D agent-based

simulations for the invasion times (Fig. 2d). J.B. developed the 1D continuum model and the extension to 2D and 3D, performed bootstrapping, fitted the experimental data and supervised the work. S.D.T. supervised the *Drosophila* work of Y.X. M.R. and J.B. conceived the work and wrote the manuscript. All authors contributed ideas and reviewed the manuscript.

**Funding** Open access funding provided by Max Planck Society.

**Competing interests** The authors declare no competing interests.

**Additional information**
**Correspondence and requests for materials** should be addressed to Melissa Rinaldin or Jan Brugués.

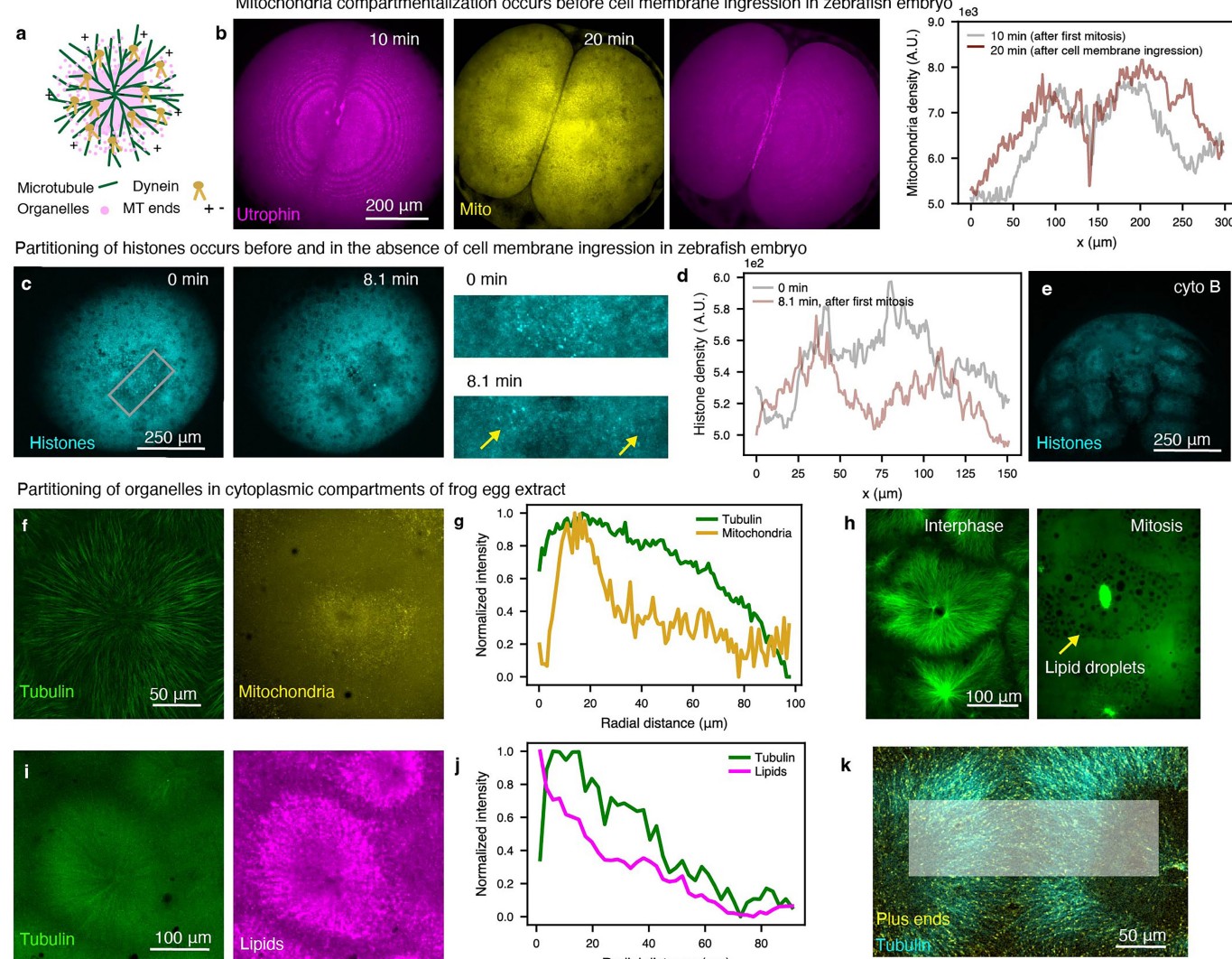

**Extended Data Fig. 1 | Compartmentalization in embryos and extract.**
**a** Live imaging of mitochondria of a zebrafish embryo with mito-GFP (see also Supplementary Video 1). **b** Quantification of the signal of mitochondria shows that the boundary is formed prior furrow formation. **c-d** Live imaging and quantification of histones of zebrafish embryo showing division of the histones prior furrow formation (see also Supplementary Video 1). **e** Live imaging of histones of cytochalasin B-treated zebrafish embryo, showing compartmentalization of the histone signal. (see also Supplementary Video 1).

**f-g** Imaging of mitochondria in a cytoplasmic compartment in extract and quantification of the radial density, together with the tubulin signal. **h** Imaging of lipid droplets as dark circles in a cytoplasmic compartment in extract. **i** Maximum intensity projection of microtubules (green) and membrane-bound organelles/lipids (magenta) of a sperm nuclei aster. **j** Radial density of microtubules and lipids of the sperm nuclei aster on the right. **k** Fluorescence microscopy image of microtubules (cyan) and plus ends visualized as EB1 comets (yellow).

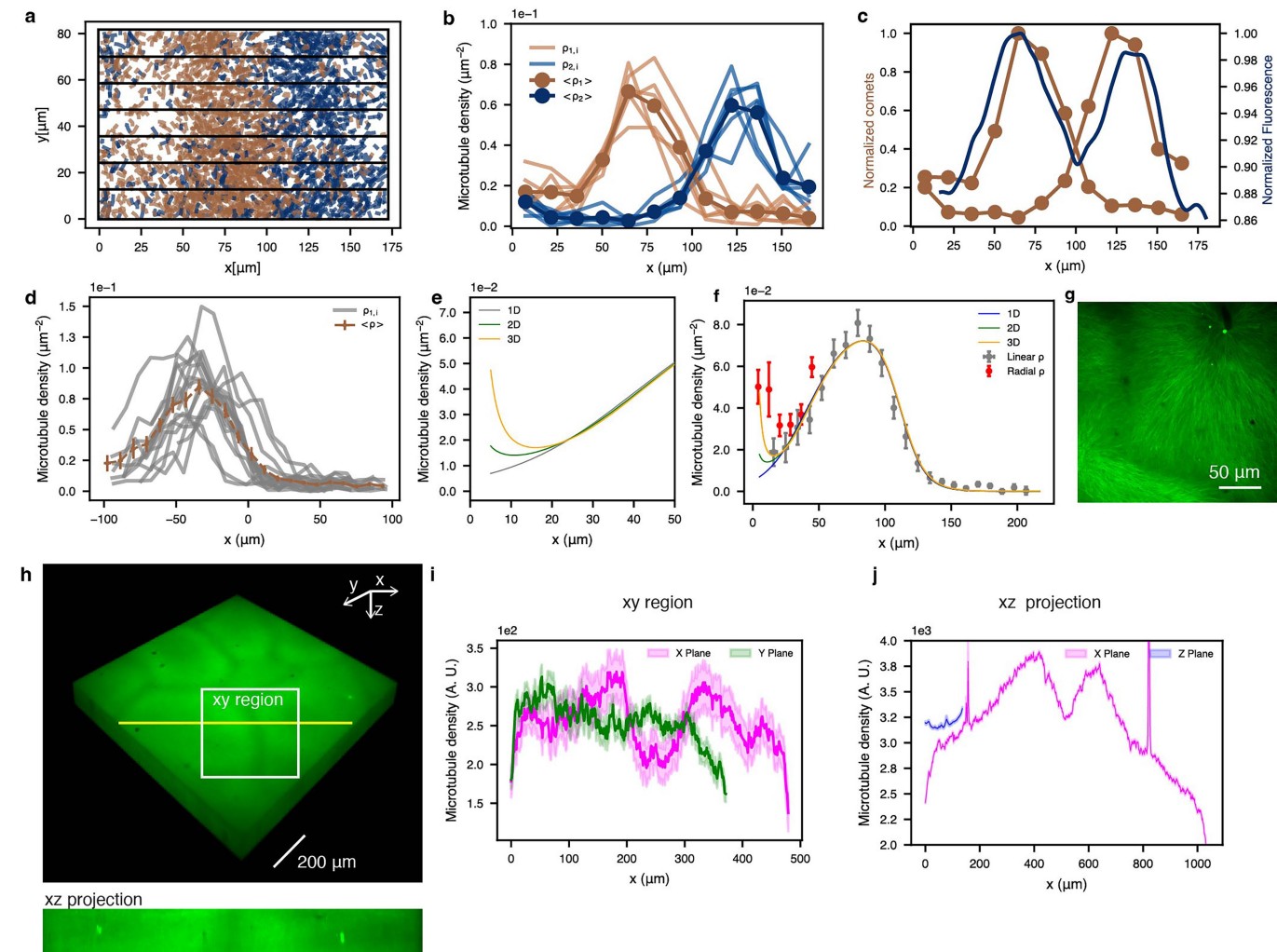

**Extended Data Fig. 2 | Quantification of microtubule density profiles and role of dimensionality. a** Tracks of plus ends. The black lines indicate the regions where the densities were measured and correspond to the different lines in plot b. The colors indicate the tracks moving to the left (blue) and the right (orange). **b** Microtubule density profiles of the tracks in the boxes in black of plot a. **c** The normalized number of EB1 comets (orange) is plotted with the normalized tubulin intensity measured in the same region. The two measurements overlap. This plot is representative of the example shown in (a) and (b). **d** Averaged density profiles in gray are plotted together with the average profile in orange. Error bars refer to s.e.m. and n = 8 independent samples. **e** Theoretical prediction of the microtubule density in 1,2, and 3D close to the center of the aster showing an initial decrease for 2D and 3D. **f** Theoretical prediction of the microtubule density in 1,2, and 3D from the center of one aster to the center of the adjacent aster.

Experimentally measured linear density of microtubules using a rectangular grid in gray, as in Fig. 1n of the main text (measurement of n = 8 independent samples and 40 frames). Error bars are s.e.m. on the different n. Experimentally measured radial density of microtubules of aster in **c** is shown in red (n = 1 sample shown as an example and 5 frames, errors are s.e.m. on the radial variation). **g** Fluorescence image of a microtubule aster with microtubules shown in green. **h** Top: Three-dimensional reconstruction of microtubule asters showing the tubulin in green. Bottom: xz projection of the yellow plane. **i** Average density profile along x (magenta) and y (green). **j** Average density profile along x (magenta) and z (blue). Shaded areas in (i) and (j) show the s.e.m. on the variation of density in the examined region of n = 1 sample as an example, repeated for n = 3 independent experiments.

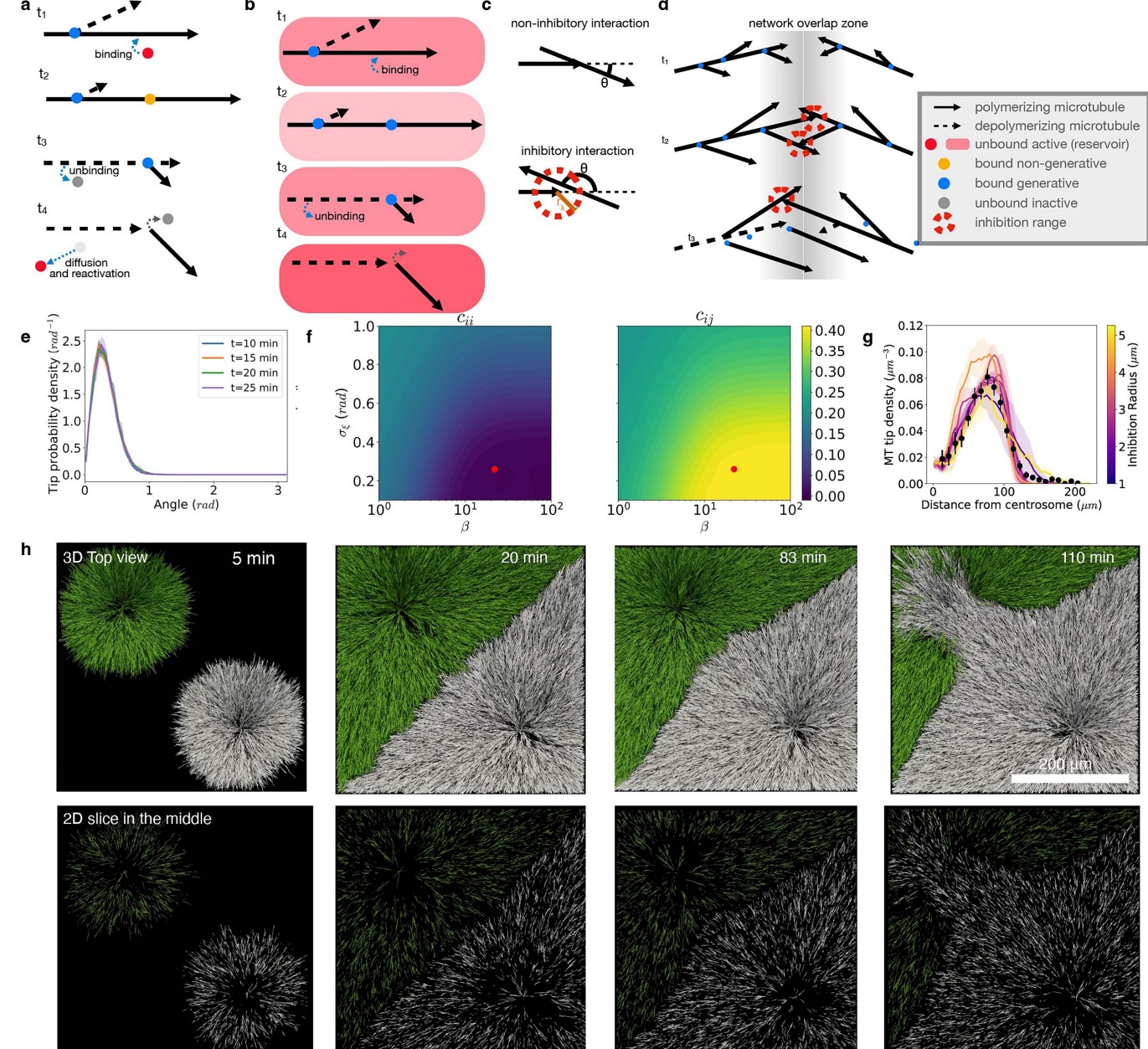

**Extended Data Fig. 3 | Agent-based simulations in three dimensions.**
**a** Schematic for filament dynamics and nucleator binding, unbinding, and nucleation kinetics for the 2D explicit unbound nucleator model. **b** Schematic for filament dynamics and nucleator binding, unbinding, and nucleation kinetics for 3D implicit nucleator model. **c** Schematic showing the inhibitory interactions between filaments. An inhibition event depends on the filament incidence angle $\theta$. When a filament's growing-end is within a radius $r/\lambda$ (orange line) of another filament, the approaching filament switches to a depolymerizing state at a rate dependent on the angle $\theta$. **d** Schematic showing two interacting filament networks. Multiple inhibitory interactions (red dashed circles) prevent the networks from passing through each other. **e** Angular probability distribution of simulated MT tips at boundary between 3D asters. Angle is relative to the direction vector from one centrosome to the other. Probability density was

calculated by binning all tips from one aster inside a cube of side length 10 µm at the center of the system (see Fig. 1) and averaging over 4 min. Data was fitted to the function $g(\theta) = \sin\theta \exp(-\frac{\theta^2}{2\sigma_\xi^2})$. The value of $\sigma$ is 0.260 rad $\pm$ 0.001 rad. **f** Self-interaction and inhibition renormalization parameters $c_{ii}$ and $c_{ij}$ plotted as a function of standard deviation $\sigma_\xi$ and inhibition angle sensitivity $\beta$ calculated from equations (25) and (27) in Supplementary Information. The red dot indicates $\sigma_\xi$ found in (e) and $\beta$ used in simulations. **g** Microtubule density profiles for different values of inhibition radius and scaling of inhibition strength as $\lambda = Cr_\lambda^{-3}$. The data in black show the averaged experimental density profile from Fig. 1l (n = 8 independent samples with s.e.m.). **h** 3D simulations of unstable asters with nucleating centers in the upper left and lower right corner of a 313 µm x 313 µm x 50 µm slab. Top row shows the top view of the slab. Bottom row shows slice close to the center of the slab.

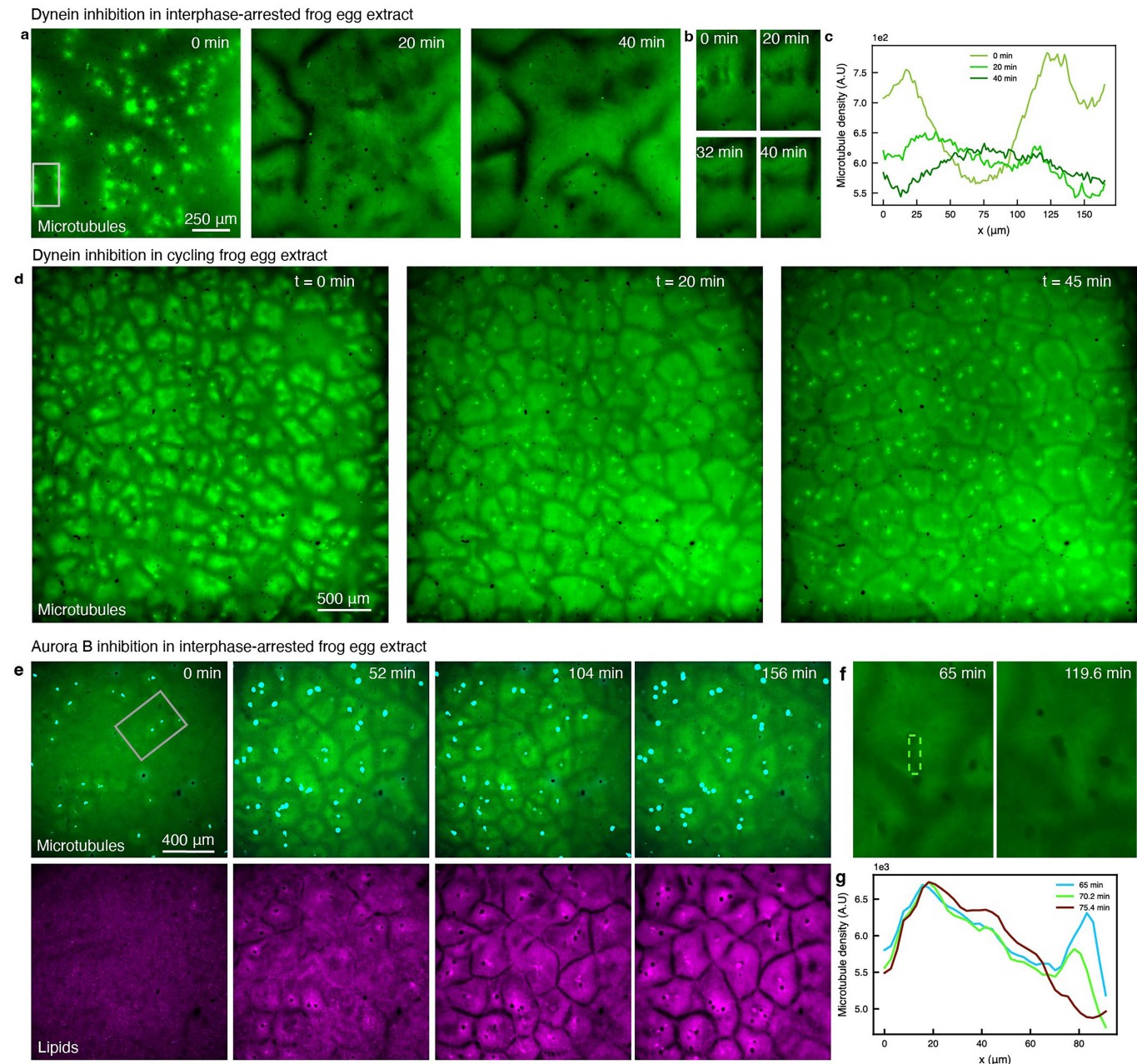

**Extended Data Fig. 4 | Motor inhibition does not prevent invasion. a** Time-lapse of the interphase-arrested extract with complete inhibition of dynein as shown by the condition of 0.08 mg/ml of p150-cc1 in Extended Data Fig. 5. **b** Invasion event in the area shown in a in gray in (a). **c** Plot of the microtubule density along the interface between asters shown in dashed green in (b) over time. **d** Time-lapse of cycling extract with complete inhibition of dynein. **e** Time-lapse of the interphase-arrested extract treated with Aurora kinase B inhibitor Barasertib. Top: microtubules. Bottom: lipids. **f** Invasion event in the area shown in a in gray in (e). **g** Plot of the microtubule density along the interface between asters shown in dashed green in (f) over time.

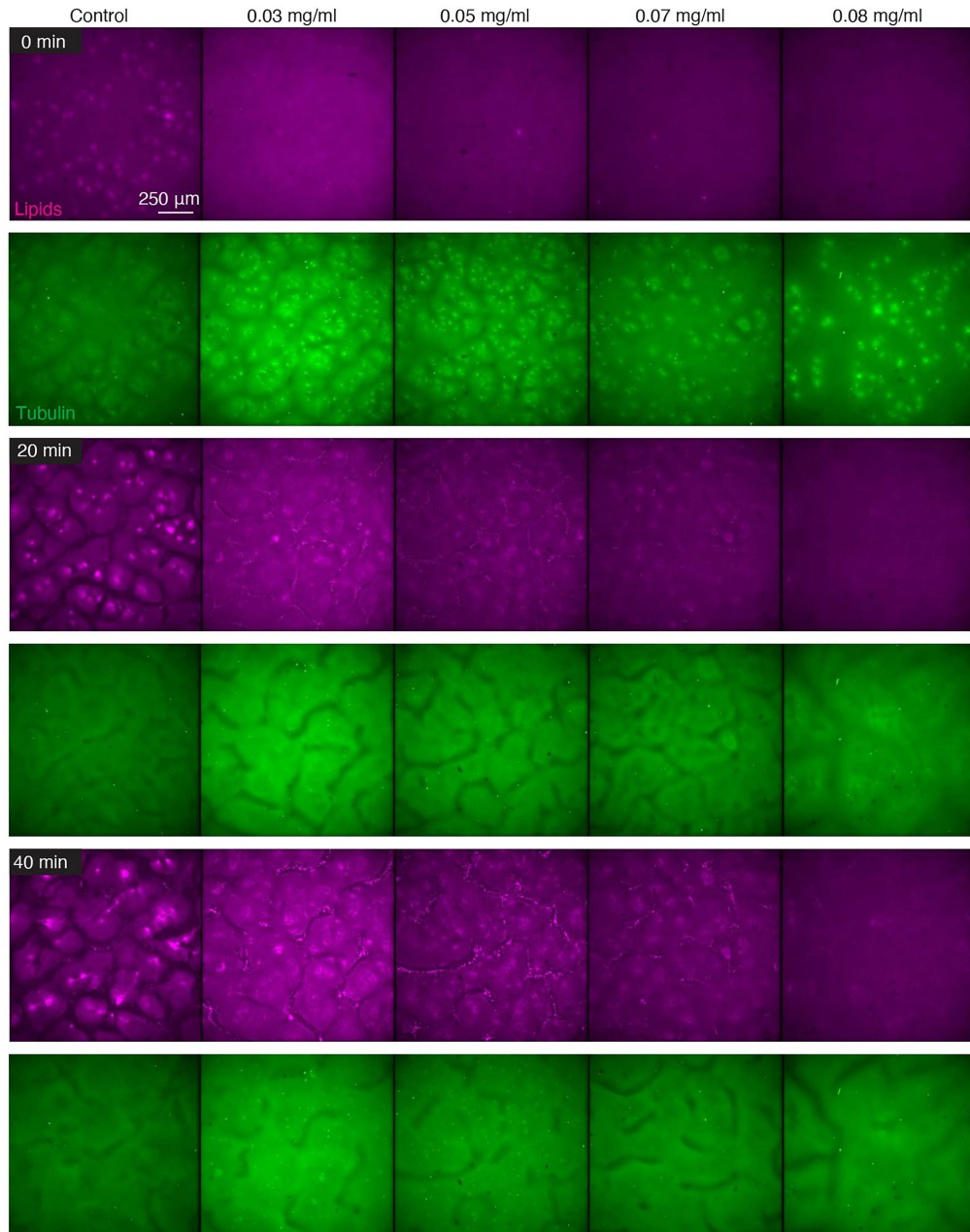

**Extended Data Fig. 5 | Partial and full inhibition of transport with titrations of dynein inhibition.** Dynein titration of p150-cc1 in interphase-arrested extract. Columns show different conditions while rows show time. Lipid signal shows partial to full inhibition of transport of organelles and therefore also compartmentalization. Tubulin signal shows that invasion occurs for all concentrations.

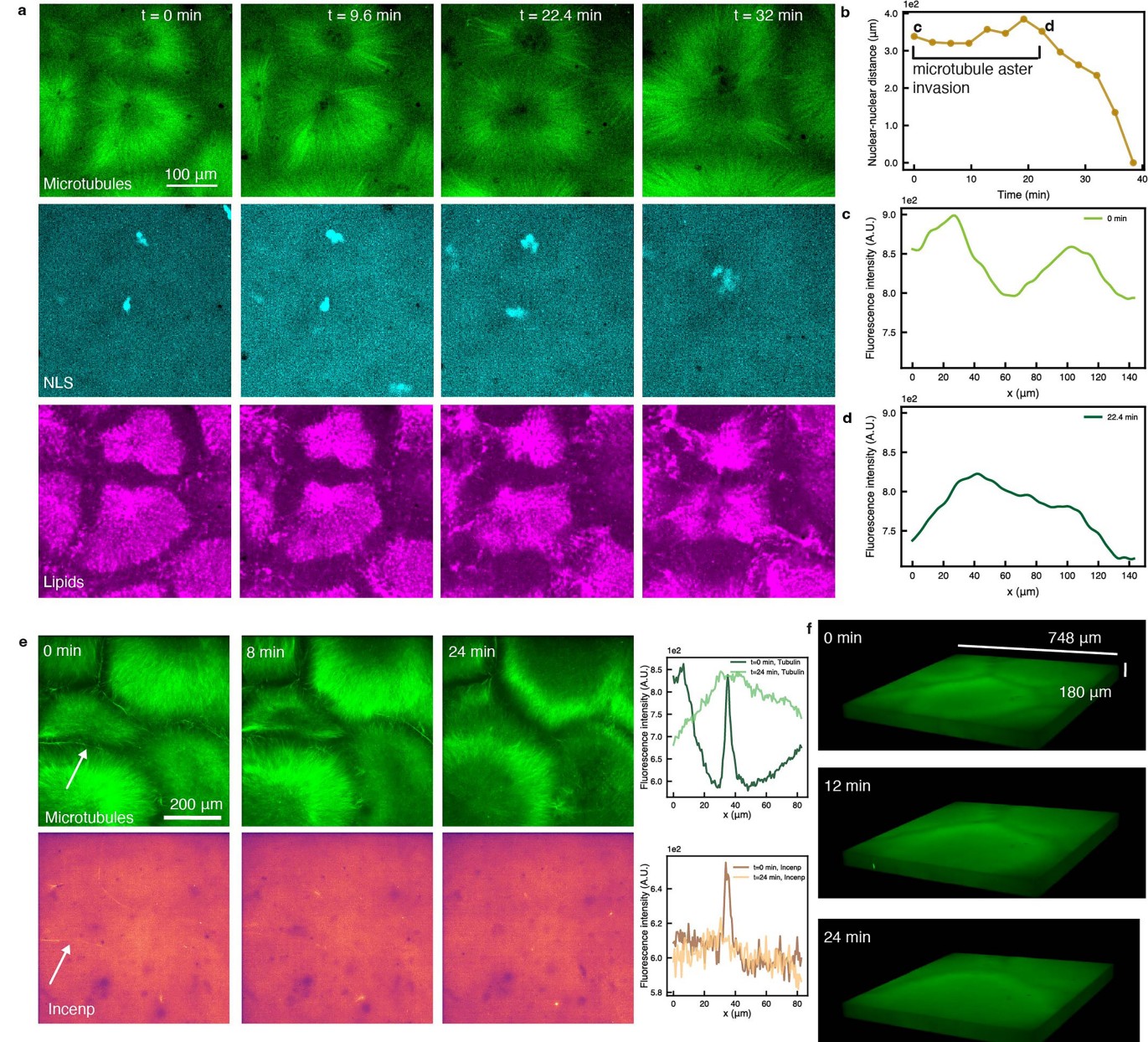

**Extended Data Fig. 6 | Nuclei, lipid organelles, and NLS during an invasion event and 3D imaging. a** Time-lapse of the microtubules, nuclei, and compartments (lipid-bound organelles) during an invasion event. **b** Distance between the nuclei of the compartments. While the microtubule aster invasion process unfolds (0-22.4 min), the distance between the nuclei remains constant. **c** Microtubule density profile at time 0 when the asters interact. **d** Microtubule density profile after the invasion event. **e** Left: Fluorescence microscopy time sequences of microtubules (green) and incenp (magma) during an invasion event. incenp is a factor that localises in the chromosome passenger complex, which is also visible in the microtubule channel as a fine line between the asters. During invasion events, incenp localization shows that the chromosome passenger complex disassembles. Right: Quantification of the intensity profile of tubulin and incenp before and after CPC disassembly. **f** 3D confocal imaging of an invasion event.

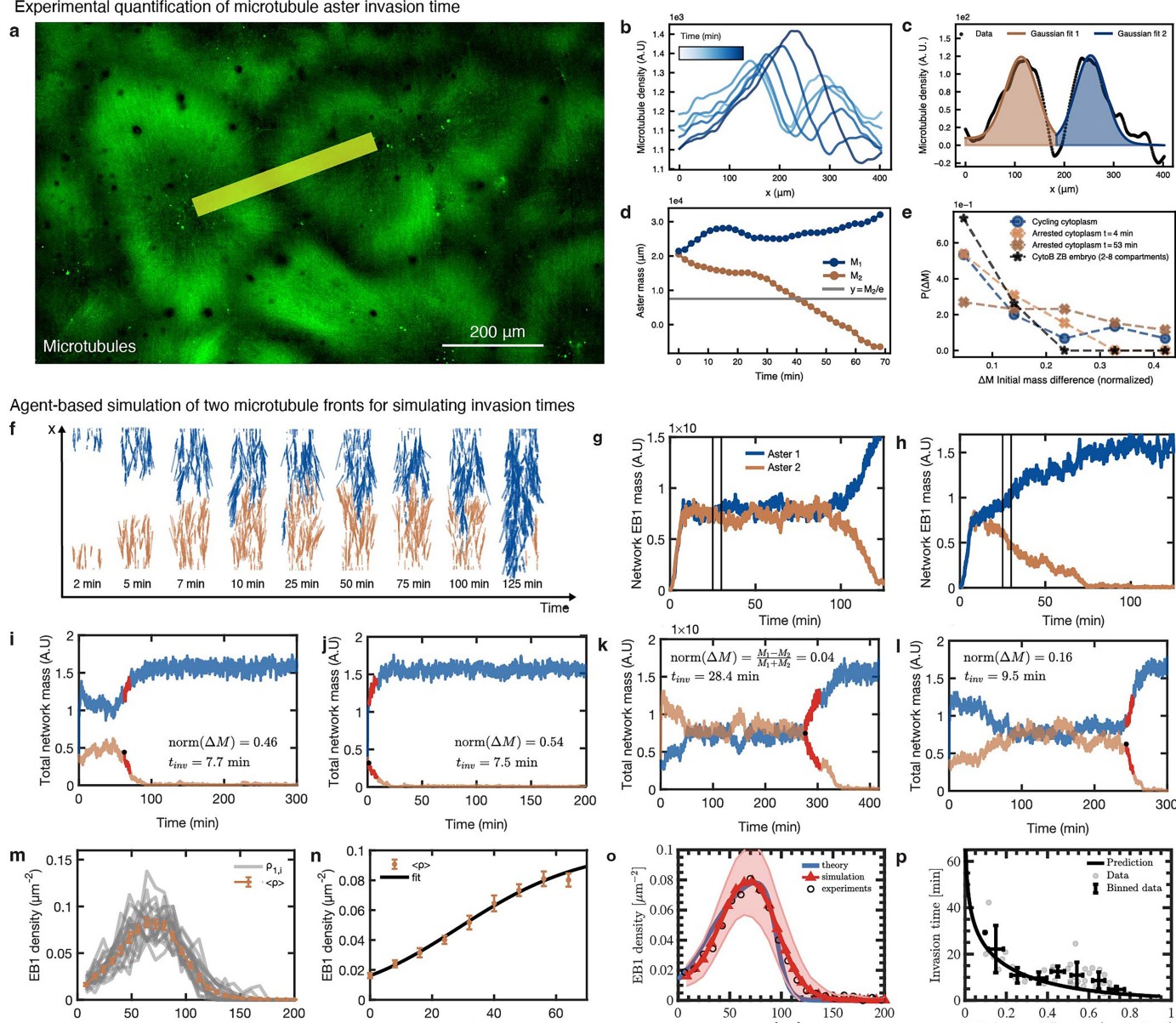

Experimental quantification of microtubule aster invasion time

Agent-based simulation of two microtubule fronts for simulating invasion times

**Extended Data Fig. 7 | Quantification of invasion times in experiments and agent-based simulations. a-b** Microtubule density profiles measured along the yellow line in (a) at different time points. **c** Fit of the microtubule density profile. Orange and blue areas indicate the mass of the asters. **d** Temporal evolution of aster mass. The black line indicates the value of mass equal to initial mass of the invaded aster divided by the base of the natural logarithm. **e** Probability distribution of the mass differences between asters. Arrested extract has a distribution that is less peaked than cycling. During invasion, the distribution becomes less peaked over time. Zebrafish syncytial embryos have a peaked distribution as it is expected from the tight aster size regulation observed in vivo. Samples numbers are: cycling cytoplasm (n = 15), arrested cytoplasm t = 0 min (n = 13), arrested cytoplasm t = 53 min (n = 15), and zebrafish (n = 18). **f** Simulation snapshots showing the time evolution of an unstable two-aster system in 2D planar geometry. Length and width of the

channel are 200 μm and 50 μm, respectively. **g** Total network mass over time for the two-aster system in f. **h** Example of a filtered trajectory. **i-l** Two asters are simulated in the planar-channel geometry. The beginning of each invasion event is indicated with the black point, taken to be $M_{invaded(t0)}$. The red overlaid curves display the invasion events. **m** Microtubule density profiles at t = 25 min (n = 20 independent simulations) are show in gray. The average curve is in orange and error bars are s.e. **n** Fitting results for the average curve in m (n = 20 independent simulations with s.e.). **o-p** Agent-based simulations with filaments that stop growing once they interact with the other aster. This condition does not change the microtubule density profile or timescale of invasion process, suggesting that the details of the inhibition term do not affect the aster profile and invasion timescales (n = 44 independent simulations and errors are s.d.).

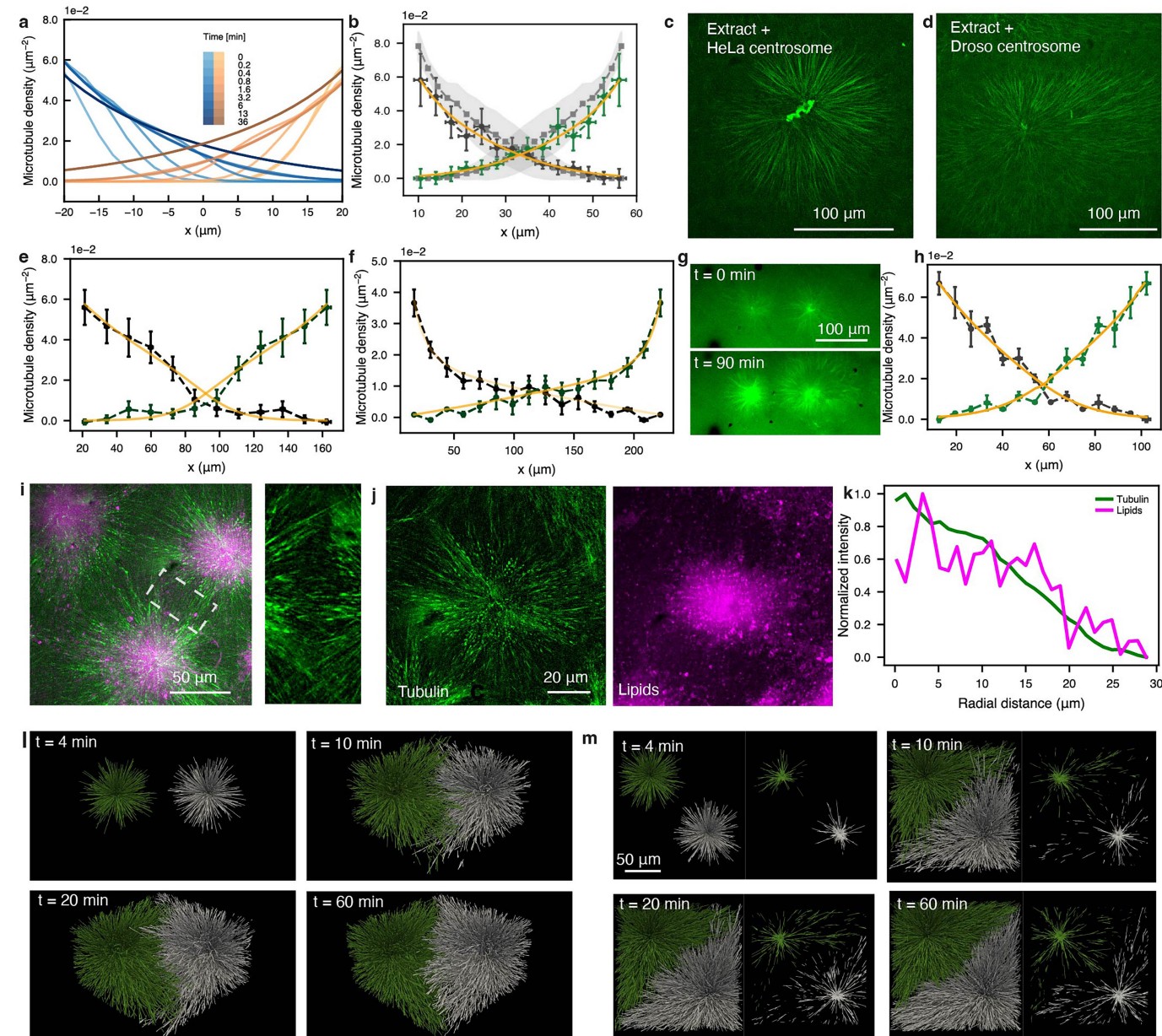

**Extended Data Fig. 8 | The stable case in experiments and agent-based simulations. a** Numerical time evolution of the stable profiles. **b** Experimental profiles (green and black lines, n = 7 independent samples with s.e.m.) and profiles obtained from the agent-based simulations (gray, n = 4 independent samples with 95% confidence interval of the mean) for the stable case. Theoretical prediction in orange. **c** Confocal microscopy image of an aster from purified HeLa centrosome. **d** Confocal microscopy image of an aster from purified *Drosophila* centrosome. **e** Microtubule density profile of two interacting asters from HeLa centrosomes (n = 6 independent samples with s.e.m.). **f** Microtubule density profile of two interacting asters from *Drosophila* centrosomes (n = 6 independent samples with s.e.m). **g** Two interacting Drosophila centrosome asters over time show stability. **h** Global fit of MCAK-Q710 asters (experimental data n = 5 independent samples with s.e.m.). **i** Maximum intensity projection of microtubules (green) and membrane-bound organelles/lipids (magenta) of AurkA bead asters. **j** Enlarged image of the interface in a shown by the dashed rectangle. **k** Radial density of microtubules and lipids of the Aur kA bead aster in d. Implicit 3D simulations of stable asters with nucleating centers placed 65um apart in the 94 μm x 94 μm x 50 μm slab. **l** Filaments from different nucleating centers (green and white) visualized in 3D from the side. **m** Filaments viewed from above (left sides) and sections of filaments that exist within a 5 μm range in the center of the slab (right sides) to compare with a single experimental z-slice. See also Supplementary Video 11.

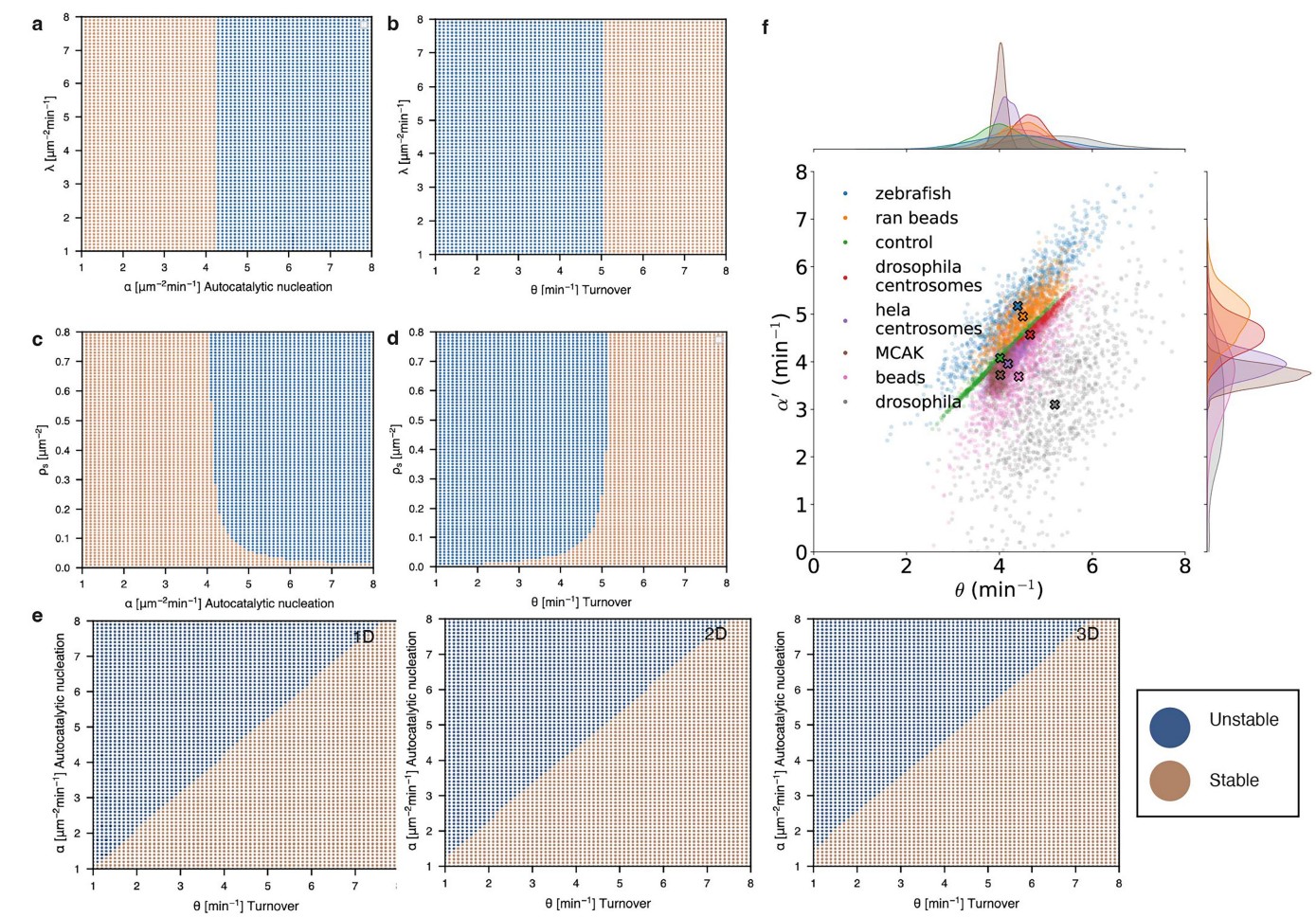

**Extended Data Fig. 9 | Phase diagrams and bootstrapping. a** Phase diagram of $\lambda$ and $\alpha$. **b** Phase diagram of $\lambda$ and $\theta$. **c** Phase diagram of $\rho_s$ and $\alpha$. **d** Phase diagram of $\rho_s$ and $\theta$. **e** Phase diagram of $\alpha$ and $\theta$ in 1, 2 and 3 D. **f** Bootstrapped points for Fig. 4d.

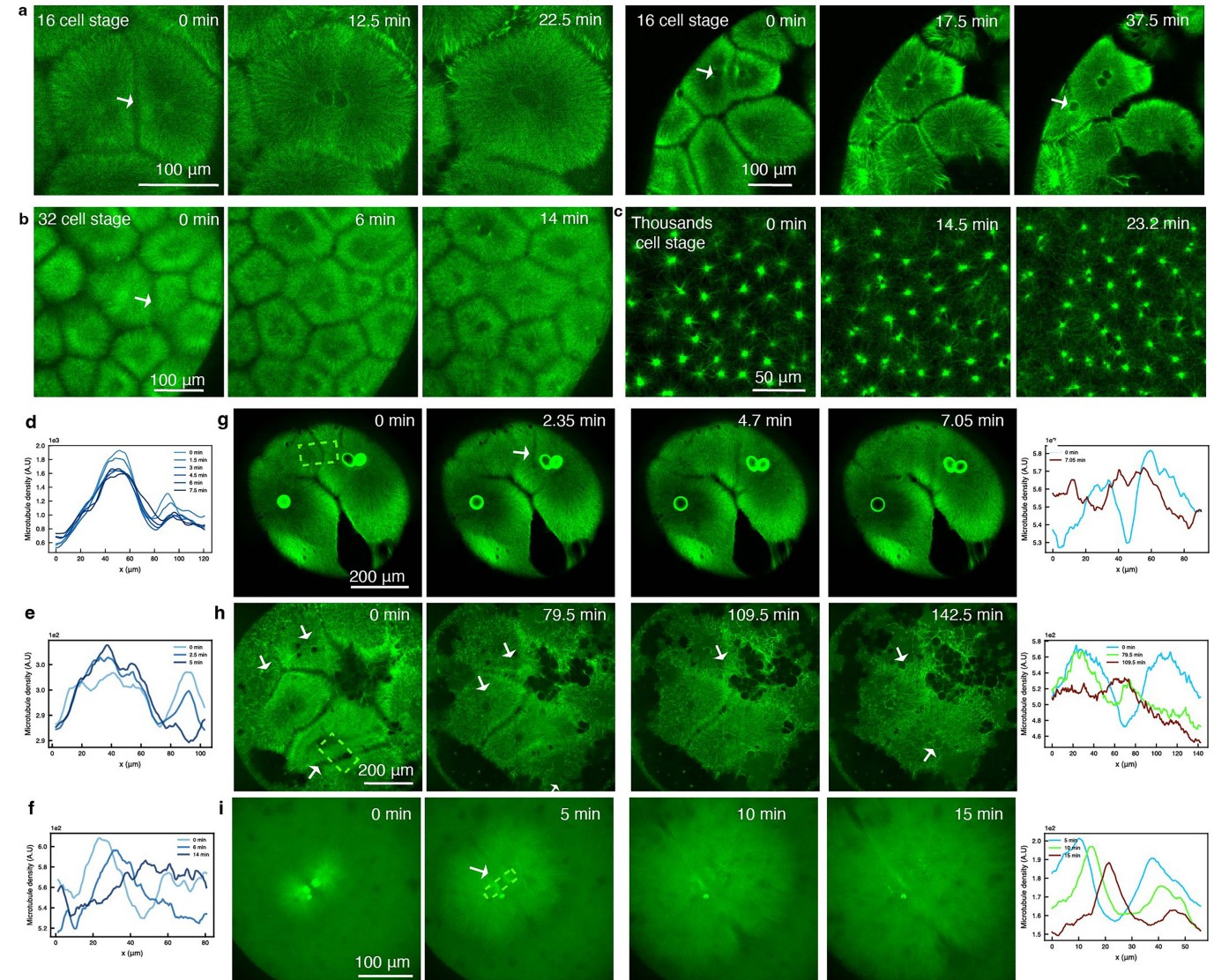

**Extended Data Fig. 10 | Cell cycle arrest at different stages and perturbation of autocatalytic wave competition in zebrafish embryo. a** Cell cycle arrest in syncytial zebrafish embryo at 16 cell stage. **b** Cell cycle arrest in syncytial zebrafish embryo at 32 cell stage. **c** Cell cycle arrest in syncytial zebrafish embryo at thousands cell stage. **d** Microtubule density profiles across the interface during an invasion event in a zebrafish embryo arrested at 8 cell stage (Fig. 4f). **e** Microtubule density profiles across the interface during an invasion event in arrested extract in a droplet showing similar dynamics to the zebrafish embryo (Fig. 4j). **f** Microtubule density profiles across the interface during the invasion event in a zebrafish embryo arrested at 32 cell stage (b). In the case of smaller aster, the invasion event resembles the ones observed in "unconfined" extract. **g** Left: Live imaging of interphase-arrested zebrafish embryo where oil droplets were injected. Right: Plot of the microtubule density interface shown in dashed green over time. **h** Left: Live imaging of interphase-arrested zebrafish embryo treated with microtubule depolymerizing drug. Right: Plot of the microtubule density interface shown in dashed green over time. **i** Left: Live imaging of interphase-arrested zebrafish embryo treated with dynein inhibitor. Right: Plot of the microtubule density interface shown in dashed green over time.

# Reporting Summary

## Statistics

For all statistical analyses, confirm that the following items are present in the figure legend, table legend, main text, or Methods section.

| n/a | Confirmed | |
|---|---|---|
| ☐ | ☒ | The exact sample size (*n*) for each experimental group/condition, given as a discrete number and unit of measurement |
| ☐ | ☒ | A statement on whether measurements were taken from distinct samples or whether the same sample was measured repeatedly |
| ☐ | ☒ | The statistical test(s) used AND whether they are one- or two-sided<br>*Only common tests should be described solely by name; describe more complex techniques in the Methods section.* |
| ☒ | ☐ | A description of all covariates tested |
| ☒ | ☐ | A description of any assumptions or corrections, such as tests of normality and adjustment for multiple comparisons |
| ☐ | ☒ | A full description of the statistical parameters including central tendency (e.g. means) or other basic estimates (e.g. regression coefficient) AND variation (e.g. standard deviation) or associated estimates of uncertainty (e.g. confidence intervals) |
| ☒ | ☐ | For null hypothesis testing, the test statistic (e.g. *F*, *t*, *r*) with confidence intervals, effect sizes, degrees of freedom and *P* value noted<br>*Give P values as exact values whenever suitable.* |
| ☐ | ☒ | For Bayesian analysis, information on the choice of priors and Markov chain Monte Carlo settings |
| ☒ | ☐ | For hierarchical and complex designs, identification of the appropriate level for tests and full reporting of outcomes |
| ☒ | ☐ | Estimates of effect sizes (e.g. Cohen's *d*, Pearson's *r*), indicating how they were calculated |

*Our web collection on statistics for biologists contains articles on many of the points above.*

## Software and code

Policy information about availability of computer code

Data collection
: The Method section specifies how the following microscopes were used for acquiring specific data sets. Confocal data were acquired with:
(1) a spinning disk confocal microscope (IX83 Olympus microscope with a CSU-W1 Yokogawa disk) connected with two Hamanatsu ORCA-Fusion BT Digital CMOS camera), equipped with photoactivation module, and Olympus cellSens 4.3.1 software.
(2) a spinning disk confocal microscope Olympus IXplore SpinSR with cellSens 4.3.1 and Andor Revolution Spinning disk with Andor iQ3.6 software.
(3) a laser scanning confocal microscope Leica SP8 microscope with a Leica Application Suite X (LAS X). The FRAP 4 Module was used for FRAP experiments.
(4) a spinning confocal microscope Andor IX 81 microscope with a Yokogawa CSUX1 spinning disk and Andor iQ 3.6 software. The FRAP module FRAPPA was used for FRAP experiments.
Light sheet data were acquired with a Zeiss LightSheet Zeiss Z.1 and ZEN 2014 SP1 v9.2.10.54 software.

Data analysis
: Data sets were analyzed with Fiji ImageJ2 Version 2.14.0/1.54f and custom made scripts in Python (version 3.9) runnig on PyCharm 2021.1.3. In Fiji, the following plugins were used: Substract Background to remove background, StackReg for image registration for invasion time estimation, Trackmate (PMID: 35654950) run on a custom-made Jython script for tracking of EB1 comets and speckles, 3D Viewer for 3D image reconstruction. All maximum intensity projections were performed in Fiji. Linear density profiles were measured in Fiji with PlotProfile function. Mean values of intensities, lengths and areas of objects of interest were measured with SetMeasurement and Measure function in Fiji. Cellpose 2.0 (PMID: 36344832) was used to segment compartments and it was run on a Python script.

For manuscripts utilizing custom algorithms or software that are central to the research but not yet described in published literature, software must be made available to editors and reviewers. We strongly encourage code deposition in a community repository (e.g. GitHub). See the Nature Portfolio guidelines for submitting code & software for further information.

## Data

Policy information about availability of data

All manuscripts must include a data availability statement. This statement should provide the following information, where applicable:

- Accession codes, unique identifiers, or web links for publicly available datasets
- A description of any restrictions on data availability
- For clinical datasets or third party data, please ensure that the statement adheres to our policy

There is no restriction on data availability. Source data are provided with the online version of this paper and raw data are deposited on the online repository:

## Research involving human participants, their data, or biological material

Policy information about studies with human participants or human data. See also policy information about sex, gender (identity/presentation), and sexual orientation and race, ethnicity and racism.

| | |
|---|---|
| Reporting on sex and gender | *Use the terms sex (biological attribute) and gender (shaped by social and cultural circumstances) carefully in order to avoid confusing both terms. Indicate if findings apply to only one sex or gender; describe whether sex and gender were considered in study design; whether sex and/or gender was determined based on self-reporting or assigned and methods used. Provide in the source data disaggregated sex and gender data, where this information has been collected, and if consent has been obtained for sharing of individual-level data; provide overall numbers in this Reporting Summary. Please state if this information has not been collected. Report sex- and gender-based analyses where performed, justify reasons for lack of sex- and gender-based analysis.* |
| Reporting on race, ethnicity, or other socially relevant groupings | *Please specify the socially constructed or socially relevant categorization variable(s) used in your manuscript and explain why they were used. Please note that such variables should not be used as proxies for other socially constructed/relevant variables (for example, race or ethnicity should not be used as a proxy for socioeconomic status). Provide clear definitions of the relevant terms used, how they were provided (by the participants/respondents, the researchers, or third parties), and the method(s) used to classify people into the different categories (e.g. self-report, census or administrative data, social media data, etc.) Please provide details about how you controlled for confounding variables in your analyses.* |
| Population characteristics | *Describe the covariate-relevant population characteristics of the human research participants (e.g. age, genotypic information, past and current diagnosis and treatment categories). If you filled out the behavioural & social sciences study design questions and have nothing to add here, write "See above."* |
| Recruitment | *Describe how participants were recruited. Outline any potential self-selection bias or other biases that may be present and how these are likely to impact results.* |
| Ethics oversight | *Identify the organization(s) that approved the study protocol.* |

Note that full information on the approval of the study protocol must also be provided in the manuscript.

# Field-specific reporting

Please select the one below that is the best fit for your research. If you are not sure, read the appropriate sections before making your selection.

☒ Life sciences ☐ Behavioural & social sciences ☐ Ecological, evolutionary & environmental sciences

For a reference copy of the document with all sections, see nature.com/documents/nr-reporting-summary-flat.pdf

# Life sciences study design

All studies must disclose on these points even when the disclosure is negative.

| | |
|---|---|
| Sample size | We did no perform statistical analyses to calculate sample size. We chose the sample size based on similar data sets used in the field, reproducibility of the experiments (i.e. experiments was repeated enough time to observe consistent phenotypes), and experimental challenges. We report the sample sizes whenever there are error bars in the legend. We report here below and in the "Statistics and Reproducibility" in the Methods the sample sizes associated to the representative images chosen in the paper. |
| Data exclusions | Data were excluded in the case embryos or extract underwent early apoptosis. |
| Replication | Experiments were replicated over the course of about three years with different microscopes and for some conditions by different experimenters (cell cycle arrest in extract by M.R. and other people in Brugués group, cell cycle arrest in Drosophila by M.R and Y. X., cell cycle arrest zebrafish by M. R. and A. K, EB1 comets for zebrafish by M. R. and A. K.). TThe number of biological replicates is indicated as number of independent samples and it refers to independent experiments for the in vitro extract studies and embryo number for in vivo studies. We report these numbers in the legend for plots with error bars and histograms. For the microtubule density profiles (Figure1l, 3c,e,g, 4e,k, and Extended Data Fig. 8 b,e,f,h), number of independent samples are reported on Supplementary Table 1. For measurements of microtubule dynamics (Figure 3a, 4c-d, and Extended Data Fig. 9c), they are reported on Supplementary Tables 1-3. Technical replicates are reported in the adjacent columns. In the Methods section and here, we provide the number of replicates for images representative of phenotypes in the |

Main Text and Extended Data Figures: Figure1: (b)Imaging of the microtubules and actin first cell cycle in zb embryo was repeated n>20 times with confocal spinning disk and n=1 with light sheet for visualization purposes. (c) n=4. (d) n=8. (e,f) n > 20. (g) n=4. (h) n=8. (i,j,k) n>20. Figure 2: (a,b) Experiments with invasion events were repeated n>20. Figure 3: (d,f) n>5. (h) n=8. (i) n=4. Figure 4: (a,b,f,l) n>20. (j) n=1 specifically with confinement in droplets, but n>20 to test morpholinos in extract. Figure 5: The specific videos were acquired as n=1 as proof of concept, however these dynamics were observed n>20. Extended Data Fig. 1: (b) n=8. (c) n=4. (e) n=1. (f) n=12. (h,i) n>20. (k) n=8. Extended Data Fig. 2: (g) n>20. (h) n=3. Extended Data Fig. 4: (a,e) n>5. (d) n=3. Extended Data Fig. 5: n>5. Extended Data Fig. 6: Experiments with invasion events were repeated n>20. This is a representative analysis of the event. (e) n>5. (f) n=3.  Extended Data Fig. 7: (a) Experiments with invasion events were repeated n>20. This is a representative analysis of the event to show the method to find the invasion time. Extended Data Fig. 8: : (c,d) n=6. (g) n=2. (i,j) n>5. Extended Data Fig. 10: (a) n=11. (b) n= 12. (c) n=9. (g) n=8. (h) n=14. (i) n>5.  Plots showing single lines without errors are relative to specific images or video (Figure 5e in the Main Text and Extended Data Figures 1b,d,g,j, 2c, 4d,g, 6b-e, 7b-d,10 d-e). These plots are based on the quantification of a single example of a phenotype that was replicated with the n value related for the figures and reported above.

| Randomization | In the analysis of the role of cell cycle times (Figure 2g-h) and  invasion times (Figure 2d), experiments were analyzed in a random order as there was no prior knowledge on the possible outcome. For the other experiments, randomization was not performed and each condition was analyzed separately. |
| Blinding | We did not perform blinding. |

# Reporting for specific materials, systems and methods

We require information from authors about some types of materials, experimental systems and methods used in many studies. Here, indicate whether each material, system or method listed is relevant to your study. If you are not sure if a list item applies to your research, read the appropriate section before selecting a response.

## Materials & experimental systems

| n/a | Involved in the study |
|-----|----------------------|
| ☐ | ☒ Antibodies |
| ☐ | ☒ Eukaryotic cell lines |
| ☒ | ☐ Palaeontology and archaeology |
| ☐ | ☒ Animals and other organisms |
| ☒ | ☐ Clinical data |
| ☒ | ☐ Dual use research of concern |
| ☒ | ☐ Plants |

## Methods

| n/a | Involved in the study |
|-----|----------------------|
| ☒ | ☐ ChIP-seq |
| ☒ | ☐ Flow cytometry |
| ☒ | ☐ MRI-based neuroimaging |

## Antibodies

| Antibodies used | Anti-INCENP (abcamp ab12183) and Anti Aurora kA (gift of Keisuke Ishihara (PMID: 27892852) |
| Validation | Anti-INCENP was labelled with Alexa 14 Fluor 488 NHS Ester (Thermofisher, A20000), tested in extract and and zebrafish embryo for localization in the Chromosome Passenger Complex.  Anti Aurora kA was used to coat beads and tested for microtubule nucleation. |

## Eukaryotic cell lines

Policy information about cell lines and Sex and Gender in Research

| Cell line source(s) | Wild type Hela cells were used for centrosome purification and were obtained from the Hyman Lab |
| Authentication | The cells were authenticated in March 2023 with PCR by Eurofins |
| Mycoplasma contamination | The cells tested negative  or Mycoplasma on 16.12.20 and 30.4.21 |
| Commonly misidentified lines (See ICLAC register) | This study did not involve ICLAC lines. |

## Animals and other research organisms

Policy information about studies involving animals; ARRIVE guidelines recommended for reporting animal research, and Sex and Gender in Research

| Laboratory animals | Xenopus Laevis, Drosophila melanogaster, and Danio rerio. |
| Wild animals | This study did not involve wild animals. |
| Reporting on sex | Sex identification cannot be performed for the early embryos in this study (max a few hours aster fertilization). |

| Field-collected samples | We did not collect samples in the field. |
| --- | --- |
| Ethics oversight | Experiments with Xenopus laevis were approved and licensed by the local animal ethics committee (Landesdirektion Sachsen, Germany; license no. DD24-5131/367/9, 25-5131/521/12, and 25-5131/564/25) and carried out following the European Communities Council Directive 2010/63/EU on the protection of animals used for scientific purposes, as well as the German Animal Welfare Act. Experiments with Danio rerio were approved and licensed by the local animal ethics committee (Landesdirektion Sachsen, Germany; license no. DD24.1-5131/ 394/ 33) and carried out following he European Communities Council Directive 2010/63/EU on the protection of animals used for scientific purposes, as well as the German Animal Welfare Act. No ethics oversight was necessary for experiments with Drosophila melanogaster. |

Note that full information on the approval of the study protocol must also be provided in the manuscript.

## Plants

| Seed stocks | *Report on the source of all seed stocks or other plant material used. If applicable, state the seed stock centre and catalogue number. If plant specimens were collected from the field, describe the collection location, date and sampling procedures.* |
| --- | --- |
| Novel plant genotypes | *Describe the methods by which all novel plant genotypes were produced. This includes those generated by transgenic approaches, gene editing, chemical/radiation-based mutagenesis and hybridization. For transgenic lines, describe the transformation method, the number of independent lines analyzed and the generation upon which experiments were performed. For gene-edited lines, describe the editor used, the endogenous sequence targeted for editing, the targeting guide RNA sequence (if applicable) and how the editor was applied.* |
| Authentication | *Describe any authentication procedures for each seed stock used or novel genotype generated. Describe any experiments used to assess the effect of a mutation and, where applicable, how potential secondary effects (e.g. second site T-DNA insertions, mosiacism, off-target gene editing) were examined.* |

