## [Peer Review File · Nature]

Robust cytoplasmic partitioning by solving a cytoskeletal instability

Corresponding Author: Professor Jan Brugués

Version 0:

Reviewer comments:

Referee #1

(Remarks to the Author)

Review comment on Rinaldin et al. (MS#2024-07-14464)

This paper investigates how cytoplasmic compartments are stably formed in space by microtubule asters using cytoplasmic extracts from frog eggs, and zebrafish and fly embryos. By employing a previously established assay that demonstrates the spontaneous formation of cell-like units in egg extracts, the authors performed live fluorescence imaging, molecular perturbations and mathematical modeling to explore how growing microtubule asters touched next to each other can maintain stability without penetrating each other's space. The authors' mathematical analysis suggests that the boundaries between asters are intrinsically unstable, leading to space invasion through autocatalytic amplification of microtubules, which contrasts the naturally occurring stable aster formation observed in the extract. Experimental interventions using cell-cycle inhibitors, kinase-coated microbeads and constitutively active Ran indicate that cell cycle regulation and autocatalytic microtubule growth are crucial factors in making the partitioning events stable. Further *in vivo* experiments in zebrafish and fly embryos support that these factors are essential for the distinct aster positioning observed in these species, proposing a general mechanism underlying cytoplasmic compartmentalization.

My view is that this paper represents a nice incremental extension of previous studies done by Ferrell's group (Cheng and Ferrell, *Science*, 2019) and Mitchison's group (Ngyyen et al., *Science*, 2014), with valuable physical insights. While many of the experimental techniques employed are consistent with those used in earlier studies, the findings presented here, particularly the observation that tuning a small set of parameters can drastically alter aster behavior and coherently explain *in vivo* observations across different species, represent a potential conceptual advance. To fully convince the readership, further refinement in both experimental work and modeling would be needed.

One of my primary concerns is the mathematical model, which currently does not account for the three-dimensional (3D) space over which microtubules explore, nor does it consider the mechanical forces that might influence the boundaries and positioning between asters. In 3D, the probability of physical interaction between microtubules growing from opposite asters might be significantly lower than in 1D or 2D, and so the inhibitory effects might also differ. Forces exerted by microtubule growth and motor proteins could impact the cell cortex and consequently affect aster positioning and invasion. These forces might also push and pull microtubules at the aster-aster interface, creating repulsion or attraction between them. In fact, the movies provided show more complex behaviors, such as migration and fusion, occurring dynamically in the cytoplasmic space. However, in the model, the asters appear to be fixed in position. While starting from a simplified model is a reasonable approach, there is a risk of oversimplification, which should be avoided unless supported by experimental data or robust reasoning. I recall studies in *Drosophila* embryos reporting the role of motor proteins in aster positioning.

Another area that could be improved is the *in vivo* experiments aiming to test the conservation of the partitioning mechanism. The experiments performed in zebrafish show surprisingly small effects on invasion. No perturbation experiments on microtubules were performed to demonstrate the universality of the mechanism. Further exploration regarding this point should strengthen the manuscript's conclusion.

Minor comments:

- 1) The manuscript could benefit from clearer definitions of the cytoplasmic compartmentalization discussed. Is the term referring solely to ordered aster positioning, or does it also involve the creation of a distinct cytoplasmic region by the patterned microtubule network that for example limits protein mobility within the cytoplasm?
- 2) Given the extension from frog egg extracts and fly systems, it is important to rigorously demonstrate that zebrafish embryos exhibit aster-based subcellular partitioning prior to the formation of physical boundaries (e.g., furrow). Comparing movie data of division events with and without actin inhibitors, visualization of membranes, etc. could provide information. Providing images of actin in intact zebrafish embryos would also help confirm the efficacy of the drug treatment. Data indicating the syncytium until the 32-cell stages (line 33, p. 2) are not provided.
- 3) Some of the Supplementary videos appear to be mislabeled in the main text and in the initial frames of the movie files.
- 4) Page 4, line 13. According to the previous lines, it seems that the morpholino "increases" rather than "decreases" cell cycle duration and aster invasion.
- 5) The manuscript could discuss more explicitly how inhibition of cyclin production might affect other factors, such as the downstream targets of CDK. Is there any evidence of entrainment of aster dynamics by cell cycle regulators or vice versa? Or do they just work independently? Since this point is highlighted in several parts of the main text, it is helpful to describe more about the potential interplay between cell cycle and subcellular partitioning.
- 6) Figure 3 and Table S7. It is surprising that different microtubule regulators result in essentially no change in the microtubule turnover rate. Experimentally perturbing the turnover rate could further validate the model used to generate the phase diagram and strengthen the conclusion.
- 7) The authors estimated the value of alpha by measuring the initial growth of asters (line 26, p. 3) or from the steady profile at the center (p. 15 in Supplementary Info). It is likely that this value is assumed to be constant over time and space in their model. Is this assumption reasonable?
- 8) The "Drosophilization" experiments presented in Fig. 5 are somewhat unclear and would benefit from a more controlled setting. This could be done by the addition of purified centrosomes in a chromatin-free condition, for example. The images presented in Fig. 5 also need clarification, especially the white broken lines in panels A and B, which are not explained.
- 9) Some of the experimental data, especially in the later parts of the manuscript, are presented solely as movies without quantification.
- 10) In Figure 5C and D, the authors present a diagram explaining how variations in the mechanism of partitioning observed across different species can be attributed to the tuning of a small set of parameters. While this nicely describes the observed phenomena, it would be worth to elaborate on the potential functional or evolutionary advantages of these different partitioning strategies used in embryos of each species. For example, does the high stability of fly embryos against cell cycle delays with small asters confer any specific advantage over the embryogenesis of other species?

Referee #2

(Remarks to the Author)

This paper by Rinaldin and colleagues looks at the ability of microtubule asters to partition cytoplasm in a syncytium. The work is original. In addition, it is important because, while it is usually assumed that multicellular organisms are subdivided into compartments by cytokinesis during development, in many systems (from malaria to many plants, flies and fish) there are periods of development in which nuclear division occurs without cellularization. In these cases, compartmentalization relies on organization of the cytoplasm itself. While very few have studied the problem in any detail, the phenomenon was observed in the 1800s, when Sacks described these compartments as "energids". This paper is one of very few to attempt to understand how this type of cytoplasmic organization is achieved using microtubules emerging from centrosomes.

In this study Rinaldin and colleagues show, very beautifully, that there are two distinct modes by which this can be achieved by microtubules that stop growing when antiparallel microtubules meet.

1. Microtubule asters can rapidly grow (through explosive nucleation) to fill the entire available cytoplasm (Zebrafish or *Xenopus* like).

2. Microtubule asters can locally partition the cytoplasm, so that it gradually fills with asters over time (*Drosophila* like).

The comparison presented in the paper that studies both regimes is elegant, and the use of a combination of experiments and modelling, enlightening. Remarkably, the authors find that these two regimes differ in their stability over time. While the second regime appears relatively stable, under the first regime, the boundaries between neighboring asters gradually coarsen. Nevertheless, since it takes time for the boundary between compartments to break down, the authors show that the first strategy is a viable one if the cell cycle period is relatively short.

This beautiful observation helps to explain the trade-offs an organism must make when using one of these two regimes to organize the cytoplasm of the developing zygote prior to cellularization:
Fast, global and unstable, versus slow, local and stable.

Overall, this study should interest a very wide audience of biologists interested in cell biology, development, evolutionary biology and biophysics.

Comments:

1. Other factors also likely differ between the two regimes. It would be good if these could be fleshed out in more detail. For

example, the two regimes also seem to differ in the density of microtubules filling the cytoplasm, and the sharpness of the boundary between asters. Does this have consequences on the ability of the two different systems to compartmentalize the cytoplasm in the two cases? Also, does the effective mesh size of the microtubule network needed to partition the cytoplasm depend on the size of the object we need partitioned (e.g. lipid droplets/vesicles/mitochondria/ER?)

2. The self-organising patterning process described by Turing involves the short range diffusion of an autocatalytic signal and a longer range diffusion of an inhibitor. This is inherently STABLE because both diffuse away from the same source. In the case studied here, there is spreading autocatalysis, together a special kind of local inhibition at regions of overlap. Thus, while highlighting the differences, I would strongly advise against calling this Turing patterning. It will just confuse readers and muddies our notions of "Turing patterning".

3. The colors and symbols used in the diagrams are hard to see. As examples of this, the gradients in color that denote time are very hard to discern in 1J and 2F.

4. How exactly was the experimental data shown in 2D generated?
I would expect to see organisms keeping a tight grip on the heterogeneity of nucleation.
Is this the case? Can this be quantified and shown?

5. As mentioned above, in the model the different regimes seem to also differ in the sharpness of the MT boundary and the density of microtubules at the boundaries between asters. Because of this, it would be good to show how MT densities (and density gradients) scale with changes in parameters, and what consequences this has for cytoplasmic partitioning. For example, in Figure 3D there seem to be no clear boundaries between compartments, therefore no precise partitioning. Is this correct?

6. While it is good that the paper includes supplemental material with more precise perturbations. Nevertheless, it's a pity that the mechanisms used to inhibit the cell cycle are so crude. Also, inhibiting protein synthesis and degradation will have numerous side effects. Why not simply add non-degradable cyclin to extracts to stop the clock? Highlighting the more specific perturbations in the body of the paper would really help make the case that what is being measured is a specific consequence of the impact of the perturbations on the cell cycle.

7. What are ranges of cell cycle times during divisions in early developing fish and frog embryos? Do these rates scale with MT growth rate, so that more instability correlates with a faster cell cycle time? I think such measurements are an important test of the theory.

8. Does the MT growth rate affect cell cycle timing to ensure the two systems remain coupled?

9. Specific comments on Figures.

Figure 1C – would be good to show actin as suggested in Figure legend.

Figure 1J – would be useful to show change in boundary over time, to make clear the rate at which the destabilizing of the boundary occurs.

Figure 1K seems to be missing.

Figure 2A – the boundary between cytoplasmic regions seems to become even more prominent over time. Is this predicted from the *in silico* analysis? How do the authors explain this?

Figure 2A – it seems to me important to show what happens when two compartments fuse to show and explain the dynamics of nuclei repositioning, along with lipid and MT reorganization.

Referee #3

(Remarks to the Author)

Rinaldin et al, 2024:

How the cytoplasm is partitioned in the absence of cytokinesis in early developing embryos is an important and long-standing problem. This topic has received renewed attention, thanks to recent work by Cheng and Ferrel (Science, 2019), who showed that the cytoplasm can organize itself into regular, cell-sized compartments through Microtubule (MT) asters growth and dynein activity. In this new study, Rinaldin and colleagues propose that an inherent instability, caused by a competition between self-amplifying MT growth and turnover, affects the robustness of MT aster-based cytoplasm compartment size definition. Interestingly, the impact of this instability in early embryos might be limited by the rapidity of cell cycles (like in fish embryos or xenopus extracts) or by restricting microtubule (MT) nucleation (as seen in drosophila embryos). A mathematical model predicts the origin and limitation of this instability, and this prediction is tested through various experiments and quantitative assays, as well as 2D simulations. This model and its various tests represent the main advancement of this work over the current state of the art. Overall, this work provides an interesting and potentially universal mechanism that could explain how membrane-less syncytia can become organized. My main concerns are listed here:

- Generally speaking, the use and direct comparison between different model systems, although interesting, is a limitation of this work. These organisms are quite divergent from an evolutionary standpoint, which raises concerns about how the

molecular regulation of MT autocatalytic growth, nucleation, or repulsion might be conserved among them. In the paper, fish embryos and *Xenopus* extracts are treated with actin inhibitors, while *Drosophila* embryos have intact actin networks. Moreover, most of the work is done in *Xenopus* extracts using methods and materials similar to the Cheng and Ferrel paper; yet, the abstract only mentions results in fish and fly embryos, which are much less explored in the MS. This is surprising.

- The key novelty of this work is the mathematical model that predicts the existence of an instability where one aster can grow at the expense of another. The model has several shortcomings. First, it is a 1D model, which contrasts with the 2D and 3D experimental situation, where multiple asters interact in all directions. Second, the fitting of the model to experimentally measured MT density relies on circular arguments, as some model parameters are inferred from fitting experimental curves (Fig 1H). Third, the general term used for the inhibition between asters accounts for the density of competing asters but lacks a clear description of the depolymerizing effect of kinesins at aster-aster interfaces.
- Many tests of the effect of cell cycle duration are done using cycloheximide both *in vitro* and *in vivo*, which is likely to also affect MT dynamics and organization. It would be important to better rule out any side effects of this treatment. A control using morpholino to affect cyclins B1 and 2 is provided in video6, but the extracts appear to behave very differently than in cycloheximide, with "invasion" events hard to detect. The same concern applies to experiments with RanQ69L, which may affect other aspects of MT dynamics beyond nucleation.
- A final concern is the relevance of the cytoskeletal instability to *in vivo* situations. Observations from zebrafish embryos (Fig 4E/ Video 9), where the cell cycle is arrested, do not convincingly demonstrate invasion between compartments/asters. There are only a few MTs that cross boundaries, and other parts of aster pairs rather separate from each other (bottom of Fig 4E). This phenotype differs from the *in vitro* situations and simulations. Much more work would be needed to better demonstrate the validity of the model for developing embryos *in vivo*. Could the authors, for instance, manipulate nucleation, repulsion, or other MT properties in *Drosophila*, which is genetically tractable?"

Version 1:

Reviewer comments:

Referee #1

(Remarks to the Author)

The revised manuscript by Rinaldin et al. presents thorough and comprehensive data, both in experimental and computational, that convincingly address most of my earlier comments. The 3D simulations provide strong support for the authors' thesis on boundary robustness and instability, and the new perturbation experiments nicely validate the model's predictions. Overall, this is a beautiful piece of work that proposes the general strategy for embryonic division control and is likely to stimulate broad interest across biology and physics. I have a few minor suggestions that may further improve the clarity of the manuscript before publication:

1. Lines 25-27, p. 8, and Extended Data Fig. 13: The text states that aster invasion is enhanced by inhibiting dynein in zebrafish embryos. Since elsewhere the authors explain that dynein does not play a major role in invasion (e.g., line 4, p. 28; Extended Data Figs. 4 and 5), it would be helpful to clarify this probable discrepancy.
2. In the 3D simulation, the invasion of the aster boundary may critically depend on the Inhibition radius r_{λ} (p. 31, Supplementary Information). If so, how robust are the outcomes to the choice of this parameter? Is the value set to 2 μm reasonable in the context of the actual biological situation? What parameter(s) were tuned to achieve the stable case?
3. Figure 5 caption (panels c-e, lines 5-6, p. 25): In panel c, the caption should explicitly state that the data are from frog cytoplasmic extract. In panel d, the source of the centrosomes should also be specified. In panel e, the labels "egg centrosomes" and "HeLa centrosomes" are not immediately clear. Are these data obtained with frog extracts as in Fig. 5c/d and Extended Data Fig. 9?
4. Figure 1b: The figure shows only one panel, while the caption implies multiple panels (with the description "Left").

Referee #2

(Remarks to the Author)

The authors should be congratulated for their hard work. The paper is much improved by the addition of new data and by verification that the dimensionality of the model has no major impact on the results.

The conclusions of the paper, made by comparing theory and experiment, are important. They help reveal the physical constraints that act to determine how cytoplasmic patterning can evolve in an early embryo. This is precisely where physical modeling makes the greatest contribution to our understanding of biology.

I would still suggest that the references to Turing patterning removed, since in my view this type of language creates more

confusion than it helps to enlighten.

Referee #3

(Remarks to the Author)

"This revised manuscript shows significant improvements over the initial submission. In particular, the new 3D agent-based simulations provide realistic models for comparison with experiments. In addition, the authors have expanded their characterization of aster invasion and the modulation of MT dynamics in vivo, which reinforces the claims of the work.

I have one last point, which arises from the newly added material: In the 3D simulations, aster invasion appears to proceed with finger-like protrusions along the aster center-to-center axis (e.g., Fig. 1q), whereas this is not obvious in the experimental images and movies provided in the manuscript. Could the authors explain why this occurs in simulations but not necessarily in experiments?"

Version 2:

Reviewer comments:

Referee #1

(Remarks to the Author)

The authors' responses and the additional changes in the manuscript have addressed my questions, and I would like to congratulate them on their beautiful work. One minor but important point, which I suggest the authors ensure before publication, concerns the data presented in Fig. 5d (related to my comment #3). The authors explain that the asters are derived from egg-intrinsic centrosomes. Are those not from "high sperm nuclei?" It is generally assumed that *Xenopus* eggs and extracts lack centrosomes and sperm brings them. Although it may be possible that the authors' cycling extracts promote the regeneration of centrosomes, it is important to confirm their assay method (i.e., whether sperm was added or not) and clarify their interpretation accordingly.

Referee #3

(Remarks to the Author)

The authors have addressed my last minor point. Congratulations on a beautiful work!

Point by point response to the referees

Legend:

Blue text: referee comments

Black text: response to referee comments

Green text: additions to the original text

Referee 1

This paper investigates how cytoplasmic compartments are stably formed in space by microtubule asters using cytoplasmic extracts from frog eggs, and zebrafish and fly embryos. By employing a previously established assay that demonstrates the spontaneous formation of cell-like units in egg extracts, the authors performed live fluorescence imaging, molecular perturbations and mathematical modeling to explore how growing microtubule asters touched next to each other can maintain stability without penetrating each other's space. The authors' mathematical analysis suggests that the boundaries between asters are intrinsically unstable, leading to space invasion through autocatalytic amplification of microtubules, which contrasts the naturally occurring stable aster formation observed in the extract. Experimental interventions using cell-cycle inhibitors, kinase-coated microbeads and constitutively active Ran indicate that cell cycle regulation and autocatalytic microtubule growth are crucial factors in making the partitioning events stable. Further in vivo experiments in zebrafish and fly embryos support that these factors are essential for the distinct aster positioning observed in these species, proposing a general mechanism underlying cytoplasmic compartmentalization.

We thank the referee for the kind words and useful suggestions which we address in the following responses.

My view is that this paper represents a nice incremental extension of previous studies done by Ferrell's group (Cheng and Ferrell, Science, 2019) and Mitchison's group (Nguyen et al., Science, 2014), with valuable physical insights.

We appreciate the recognition of the valuable physical insights provided by our work. We would like to clarify the conceptual novelty of our study in relation to the important contributions made by Nguyen et al. (Science, 2014) and Cheng et al. (Science, 2019). In contrast to these two studies, which relied primarily on biochemical perturbations in frog egg extracts, our multidisciplinary approach—combining comparative embryology with quantitative experimental and theoretical methods—allowed us **to uncover a novel coupling between a biological clock (the cell cycle) and a physical instability**. This coupling gives rise to a dynamic and tunable mechanism that

ensures robust yet flexible spatial organization of the cytoplasm during early development. We believe that the discovery of cell cycle control over this intrinsic instability, and its consequences for development, represents the central advance of our work and sets it apart from the two previous Science publications.

Beyond this central finding, our work offers two key contributions that advance the earlier studies:

Role of inhibition in aster-aster interaction. In contrast to what was proposed by Nguyen et al., we show that local inhibition alone is insufficient to ensure stable compartmentalization. Specifically, the Mitchison group proposes in Nguyen et al. (Science, 2014), Nguyen et al. (Mol Biol. Cell 2018) and remarks later in a review in Mitchison and Field (Annu Rev Cell Dev Biol., 2021) that “The PRC1-KIF4A module is likely sufficient to partition MTs between asters” prior to cytokinesis, and they suggest that for longer-lasting syncytia such as *Drosophila* embryos “we predict that a module equivalent to PRC-KIF4A partitions *Drosophila* energids by blocking MT growth at boundaries.” We understand that the authors arrived at this conclusion based on a system limited to compartments formed by interphase-arrested microtubule asters nucleated from Aurora kinase A beads, which—according to our findings—are inherently stable. However, our manuscript now offers a framework that finally explains the physical origin of the boundaries between compartments and how they adapt in different syncytia.

Size regulation of cytoplasmic compartments. Differently from Cheng et al., who reported but did not account for the mechanisms underlying compartment size regulation, our cell cycle perturbation experiments demonstrate that temporal coordination of cytoplasmic instability with the cell cycle oscillator is key to achieving robust compartmentalization.

While many of the experimental techniques employed are consistent with those used in earlier studies, the findings presented here, particularly the observation that tuning a small set of parameters can drastically alter aster behavior and coherently explain *in vivo* observations across different species, represent a potential conceptual advance. To fully convince the readership, further refinement in both experimental work and modeling would be needed.

We hope that the new version convincingly addresses the concerns. We are very enthusiastic about the extension of the model and simulations to 3D and new perturbations which we believe have substantially strengthened the paper, and are grateful to the Reviewer for encouraging us in this direction.

One of my primary concerns is the mathematical model, which currently does not account for the three-dimensional (3D) space over which microtubules explore, nor does it consider the mechanical forces that might influence the boundaries and positioning between asters. In 3D, the probability of physical interaction between microtubules growing from opposite asters might be

significantly lower than in 1D or 2D, and so the inhibitory effects might also differ. Forces exerted by microtubule growth and motor proteins could impact the cell cortex and consequently affect aster positioning and invasion. These forces might also push and pull microtubules at the aster-aster interface, creating repulsion or attraction between them. In fact, the movies provided show more complex behaviors, such as migration and fusion, occurring dynamically in the cytoplasmic space. However, in the model, the asters appear to be fixed in position. While starting from a simplified model is a reasonable approach, there is a risk of oversimplification, which should be avoided unless supported by experimental data or robust reasoning. I recall studies in *Drosophila* embryos reporting the role of motor proteins in aster positioning.

We thank the Reviewer for this comment on the role of dimensionality and mechanics for aster invasion which prompted us to expand theory and simulations to 3D and test the role of microtubule motors. Here we address both aspects.

Dimensionality. In the initial submission, we simplified the problem to 1D because microtubule asters maximally interact at the midzone region (Nguyen et al. *Science*, 2014 and Glotzer et al., *Science* 2005), which is the area where antiparallel microtubules from two asters overlap. While the midzone region is a three-dimensional object, microtubules mostly align along one direction, suggesting that the interaction is symmetric along the interface between the asters. We could see this by measuring the density profiles across the boundary as a function of the position along the perpendicular direction to the boundary, which do not change over 80 μm along the boundary (*Figure R1a-b*). We have now also analyzed the microtubule density in 3D by measuring the average tubulin intensity profile along the xy and xz directions. We find that there is no substantial change of microtubule density across the interface also in the third dimension (*Figure R1c-e*)

Figure R1: **a** [TEXT REDACTED] **b** Microtubule density profile along different i . This profile does not change along the interface, only across it. **c** Top: Three-dimensional reconstruction of microtubule asters showing the tubulin in green. Bottom: xz projection of the yellow

plane. **d** Average density profile along x (magenta) and y (green). **e** Average density profile along x (magenta) and z (blue). The shaded area denotes the standard deviation.

This analysis suggests that the problem of aster-aster interactions is essentially one dimensional and explains why the 1D model can faithfully reproduce the experimental microtubule density profiles in all conditions. We then employed 2D simulations to investigate the role of geometry, which did not reveal any significant effects since we could recapitulate the experimental density profile and invasion times observed in the experiments and theory.

However, we agree with Reviewer 1 that dimensionality can play a role in modelling microtubule density in space, therefore we extended the 1D theoretical model to 2D and 3D and completely remade our agent-based simulations in 3D. By extending the theoretical model to higher dimensions, we discovered that dimensionality mostly plays two roles in our system:

- 1) **Geometric decrease of microtubule density.** Due to radial symmetry, asters expanding in 2D and 3D exhibit a geometric decrease in microtubule density. This leads to a change of the flux term in the form of the following geometrical term:

$$\frac{-(N_d-1)v_p\rho}{r},$$

where N_d is the dimensionality, v_p is the polymerization velocity, and ρ the microtubule density (see new supplementary note). Without nucleation and turnover, this term makes **the density decay in 2D and 3D in comparison to 1D**, which would remain flat. However, because of MT autocatalytic nucleation and turnover this effect quickly becomes negligible. Beyond a radius $N_d-1 \sim \frac{v_p}{\theta} \sim 5-10 \mu\text{m}$, turnover effects are larger than any geometric decay, which is negligible compared to the typical size of asters of around $100 \mu\text{m}$. This effect can be seen for small distances from the center of the aster (*Figure R2a*) where we compare the new fits in 1, 2 and 3D, **but does not affect the shape of the microtubule density profile in the nucleation and interaction region**. The latter overlaps for all dimensionalities (*Figure R2b*). To consider this effect more explicitly, while in the original manuscript we plotted the microtubule density centered at 0 -the point of intersection of the two aster profiles- we now start the x axis at the center of the centrosomes of the left aster. Therefore, the x-coordinate of the first experimental point now corresponds to the distance between the center of the centrosome and where the EB1 comets begin to be measured. We note that the profiles do not start at the centrosome, because it is challenging to count the EB1 comets close to the centrosome due to the high density that prevents to correctly track them over time. However, we have performed an analysis of the radial counts of comets of the aster in *Figure R2c* without tracking their direction to observe the decay close to the centrosome (*Figure R2b, red points*).

Figure R2: **a** Theoretical prediction of the microtubule density in 1,2, and 3D close to the center of the aster showing an initial decrease for 2D and 3D. **b** Theoretical prediction of the microtubule density in 1,2, and 3D from the center of one aster to the center of the adjacent aster. Experimentally measured linear density of microtubules using a rectangular grid in gray, as in Figure 1n of the main text (measurement of 8 ROIs and 40 frames). Experimentally measured radial density of microtubules of aster in **c** is shown in red (measurement of 1 ROI and 5 frames). **c** Fluorescence image of a microtubule aster with microtubules shown in green

2) Probability of interaction for antiparallel microtubules. The second effect of dimensionality is, as the Reviewer suggests, the impact on the probability for antiparallel microtubules to physically interact at the aster-aster boundary. This probability corresponds experimentally to the decrease in density at the midzone region where antiparallel microtubules meet. Our previous analysis of microtubule density across the interface suggests that this decrease in density occurs only along the direction of antiparallel alignment in 2D (Figure R1b,d,e). To evaluate this effect quantitatively, we derived a 3D version of our model and computed the inhibition term from the probability of finding microtubules of opposed polarity (see SI 2.3 Angular and higher dimensionality continuum model). We used this 3D model to re-derive an effective 1D equation near the boundary (because of the axisymmetric symmetry). We found that the effect of dimensionality is captured in renormalized terms, but **the shape of the equation does not change with respect to the original 1D model.**

By extending the theory to **three dimensions, we can recapitulate the same microtubule density profiles that we found in 1D.** We have also tested how the dimensionality affects the compartment stability by plotting the phase diagram of α and θ for the 1D, 2D and 3D case and observe no substantial differences (Extended Data Fig. 11).

To better understand the physical interaction between the microtubules in space, and to confirm the conclusions from the continuum model, we have extended the simulations from 2D to 3D. To computationally support this extension, we implemented several improvements to our simulations: (1) we simulate $\sim 10^5$ - 10^6 filaments, in contrast to the original manuscript where we could only simulate 10^3 - 10^4 filaments. This was essential to maintain the same density of microtubules in the 3D simulations as measured in the experiments.

(2) We include an inhibitory interaction range to account for an effective filament tip search radius and/or protein binding and diffusion effects for more realistic modeling of inhibition.

(3) We increase the simulated real time from 30 minutes to ~ 2 hours allowing us to reach timescales seen in experiments. Modeling more filaments for longer simulation time in significantly less real time ($\sim 1-7$ days) was accomplished thanks to GPU accelerated code, allowing for improved parameter optimization and exploration.

We explain how these processes are taken into account in the SI section 3.2. We show that the microtubule density profile in 3D agrees with the experiments (*Figure R3a*) and the 1D theory recapitulates the aster-aster interaction (*Figure R3b*) and invasion event with a strikingly similar dynamics to the one observed in the experiments (*Figure R3c*).

Figure R3: **a** Microtubule density profile from experiments (black, green), 1D theory (orange), and 3D simulations (gray points, shaded area indicates the standard deviation). **b** Top view of a three-dimensional agent-based simulation of interacting asters in a rectangular slab showing boundary formation. **c** Side view of temporal evolution of a three-dimensional agent-based simulation of interacting asters in a slab, showing invasion.

Altogether, the extension of the theory to higher dimensions, and the new 3D simulations allowed us to **fully explore in detail the role of dimensionality in the stability of the compartments**. We find that the 1D description, complemented with the 3D simulations, is appropriate for studying the interaction between asters. This approach simplifies the calculations and provides conceptual clarity to our manuscript.

Role of mechanics. As Reviewer 1 points out, the current model does not consider the mechanics of the asters. Mechanical forces between asters can emerge from collective microtubule growth (pushing forces) or work from dynein motors (pulling forces). While we are confident that these forces are responsible for aster positioning, we experimentally find that they do not affect the relative stability of aster interfaces and the invasion process itself. To verify that motor forces do

not affect the invasion process we controllably inhibited dynein in extracts by titrating p150-cc1, a recombinant form of dynactin that inhibits dynein transport (Gaetz et al., J. Cell Biol. 2004). We verify that the inhibition works properly by observing partial to absent transport of organelles. In case of complete inhibition of dynein activity, **asters remain fixed in place, as in the model, and we still observe aster invasion events coming from the competition of autocatalytic waves, but without re-localization of nuclei.** We show in *Figure R4a* and Video 6 a time-lapse of a plane of the interphase-arrested extract with complete inhibition of dynein and the analysis of an invasion event (*R4b-c*). Interestingly, we have also inhibited dynein in cycling extract and we observe in this scenario invasion events that we attribute to the incorrect centering of the asters (*Figure R4c-d*). This perturbation is also recapitulated in experiments in embryos, as we will discuss in the next points (*Figure R8c*). In Extended Data Figure 5 and Video 6, we show both the tubulin and lipid fluorescence time lapses of different titrations of the dynein inhibitor where it is possible to observe partial to full inhibition of transport of lipid organelles and NLS. To test the effect if the absence of kinesins can affect the invasion process, we have inhibited Aurora kinase B in interphase-arrested extract with Barasertib. Under this perturbation, we still observe compartmentalization and aster invasions (*Figure R4e-g*).

Consistently, our live imaging in the control situation suggests that mechanics is not relevant during the aster invasion process. As observed in *Figure R5*, the distance between the centers of the aster -where the nuclei are positioned- is constant during the invasion. This data demonstrates that aster mechanics and repositioning of nuclei is not necessary for the invasion process -and act at different timescales than the invasion process- and can thus be neglected to investigate the stability of the compartments. However, active aster positioning is likely to be important for forming and maintaining equally sized compartments, as also suggested by the dynein inhibition experiments in cycling extract, and will be an interesting aspect to study in subsequent work.

Dynein inhibition in interphase-arrested frog egg extract

Dynein inhibition in cycling frog egg extract

Aurora B inhibition in interphase-arrested frog egg extract

Figure R4 *a* Time-lapse of the interphase-arrested extract with complete inhibition of dynein. *b* Invasion event in the area shown in *a* in gray. *c* Plot of the microtubule density along the interface between asters shown in dashed green

in *b* over time. *d* Time-lapse of the interphase-arrested extract with complete inhibition of dynein in cycling extract. *e* Time-lapse of the interphase-arrested extract treated with Aurora kinase B inhibitor. Analysis of invasion events for the gray region in *f* are shown in *g*. Relative videos are shown in video 6.

Figure R5 *a* Time-lapse of the microtubules, nuclei, and compartments (lipid-bound organelles) during an invasion event. *b* Distance between the nuclei of the compartments. While the microtubule aster invasion process unfolds (0–22.4 min), the distance between the nuclei remains constant. *c* Microtubule density profile at time 0 when the asters interact. *d* Microtubule density profile after the invasion event. See also Video 6 for a video of intensity of microtubules, lipids, and NLS during an invasion event.

Another area that could be improved is the *in vivo* experiments aiming to test the conservation of the partitioning mechanism. The experiments performed in zebrafish show surprisingly small effects on invasion. No perturbation experiments on microtubules were performed to demonstrate the universality of the mechanism. Further exploration regarding this point should strengthen the manuscript’s conclusion.

Effect of the invasion in zebrafish. We thank the Reviewer for this comment and agree that the invasion was not easily appreciable in the image presented in the main text, which we attribute to the different confinement of the asters. While asters in the extract are confined to a thin layer (Figure R1c), those in zebrafish embryos are enclosed within a spherical cap (Figure 1b), resulting in distinct visual phenotypes of interface dynamics during aster invasion. We have now mimicked the spherical embryo geometry through the encapsulation of extract in oil into small droplets and successfully recapitulated the dynamics of the interface that we observed *in vivo* (Figure R6a and Video 7). We also realized that the invasion dynamics is much clearer on specific planes than in the MIP. We have now included in the main text images of multiple planes, where it is also possible to see the dramatic relocation of nuclei that is seen in extract (Figure R6b, $z = 30 \mu\text{m}$, yellow arrow). Finally, we now also show in the main text the interesting fusion of “compartments at the cortex” that results from aster invasion (Figure R6b, utrophin signal in cyan).

Figure R6 a Interphase-arrested cytoplasmic extract encapsulated in droplets. Droplet diameter is 460 μm . Left: Schematic of the encapsulation. Right: Time lapses are reported on two different planes. **b** Time lapses of invasion events in zebrafish embryo at 8 cell stage. Left: The microtubules are shown in green at two different planes. At $z1$, it is possible to observe the relocation of nuclei, indicated by the yellow arrow. Right: top, boundaries between the compartments at the cortex, shown by Utrophin, disappear over time. Bottom: Schematic of the compartments in the zebrafish embryo.

Moreover, we provide images and videos of invasions at different cell stages. In the original manuscript, we showed invasion at 8 cell stage. Now, we provide also example at 16 and 32 (Figure R7a-b), where the asters are smaller but still have an increasing density profile from the center and are predicted to invade. In the invasion event at 32 cell stage (Figure R7b) we can see a larger aster invading a smaller one, as predicted. This event more closely resembles the invasions observed in extract, likely because arresting the cell cycle after several rounds of syncytial division results in greater variability in aster mass and size—conditions that more closely mirror those in extract- and the size of the aster respect to the embryo is smaller, therefore the effect of the confinement in the curved geometry becomes negligible.

We show that when we arrest the cell cycle at late stages (thousands of asters) when the asters have a decreasing density profile corresponding to the stable condition, the compartments remain stable (Figure R7 d). We have also improved the imaging and conditions for cell cycle arrest in *Drosophila* embryo. We use embryos that are treated with cycloheximide and cytochalasin, like the zebrafish embryos and we also show invasion in droplets of *Drosophila* extract (Figure R7 d-e).

Cell cycle arrest at different stages in zebrafish embryo

Figure R7 **a** Cell cycle arrest in syncytial zebrafish embryo at 16 cell stage. **b** Cell cycle arrest in syncytial zebrafish embryo at 32 cell stage. **c** Cell cycle arrest in syncytial zebrafish embryo at thousands cell stage. **d** Cell cycle arrest in *Drosophila* in the syncytial stage with asters in the cytoplasm. **e** Cell cycle arrest in a droplet of *Drosophila* extract in oil. **f** Microtubule density profiles across the interface during an invasion event in a zebrafish embryo arrested at 8 cell stage (Figure R6b). **g** Microtubule density profiles across the interface during an invasion event in arrested extract in a droplet showing a dynamics similar to the zebrafish embryo (Figure R6a). **h** Microtubule density profiles across the interface during the invasion event in a zebrafish embryo arrested at 32 cell stage (b). In the case of smaller aster, the invasion event resembles the ones observed in “unconfined” extract.

Perturbations of microtubules and MAPs in zebrafish. We have performed several perturbations in zebrafish embryos where we observed invasions, suggesting that it is a general phenomenon that can happen by competition of autocatalytic microtubule waves.

- (1) Initially we tried to make small asters by injecting Aurora kA beads and purified centrosomes from HeLa and *Drosophila* embryos, but these did not nucleate asters. However, we managed to create small asters by injecting oil droplets that nucleate microtubules on the surface triggering aster formation. These smaller asters were rapidly invaded by larger ones, consistent with our mechanism (Figure R8a).
- (2) By incubating embryos with a microtubule depolymerizing drug, we have created asters of lower mass at the boundary than the center (admittedly without a good control on the strength of the depolymerization, with the drug typically either depolymerizing all microtubules or having no effect). In this condition we consistently observe invasions and relocation of nuclei from the boundary to the center of the aster. (Figure R8b).

- (3) By inhibiting dynein in the embryo, we perturbed the relocalization of the nuclei prior to the formation of boundaries. This generally leads to asters that are not regularly spaced, and specifically centers that are in very close proximity to each other. In the presence of autocatalytic waves, this leads to the disassembly of boundaries (Figure R8c).
- (4) By using Barasertib to inhibit Aurora Kinase B, a kinase involved in midzone formation and inhibition between the asters, we biochemically lowered the inhibition. In this way, we could induce invasion, even at 2 cell stage which is very hard to observe in embryos treated with cycloheximide and cytochalasin since the asters have very similar mass of microtubule and size (Figure R8d). Moreover, because Aurora Kinase B is responsible of cytokinesis and lead to a syncytial embryo, we did not use cytochalasin in these experiments. Interestingly, with this perturbation, we have also observed that while an interface between asters is originally formed also in the absence of Aurora Kinase B, if the cell cycle is slower or arrested, the microtubule asters can invade (white arrow in Figure R8e shows an invasion event). This result supports the point of the paper that the interplay between the cell cycle timing and the timescale associated to the creation and maintenance of an interface between asters is key to maintain robust boundaries (Figure R8e).

Altogether, these experiments demonstrate that perturbations to either the relative microtubule mass or the aster interface can trigger aster invasion in zebrafish embryos. Given that the instability arises intrinsically from autocatalytic aster growth, numerous biochemical or physical imbalances between asters can facilitate the onset of this process. We also supplemented these experiments with perturbations in extracts to test the generality of the mechanism (Figure R4).

Finally, as suggested by the comment of Reviewer 3, we have tried to perturb the nucleation in *Drosophila* embryos, as they are generally more tractable. Indeed, we found that in those embryos it is possible to tune nucleation. Following our results on Aurora kA beads and RANQ69L, we injected RANQ69L in arrested *Drosophila* embryos and cytoplasmic droplets and observed formation of very large asters (Figure R8f). This latter result strengthens the possible role of RanGTP on setting the nucleation and size of asters in interphase, which is an interesting nucleation mechanism for future studies. We note that we performed live imaging of the embryos after the injection to observe invasions, however we could observe these large asters for only a short amount of time before they disassembled and the embryo died.

Figure R8 a Left: Live imaging of interphase-arrested zebrafish embryo where oil droplets were injected. Right: Plot of the microtubule density interface shown in dashed green over time. **b** Left: Live imaging of interphase-arrested zebrafish embryo treated with microtubule depolymerizing drug. Right: Plot of the microtubule density interface shown in dashed green over time. **c** Left: Live imaging of interphase-arrested zebrafish embryo treated with dynein inhibitor. Right: Plot of the microtubule density interface shown in dashed green over time. **d** Left: Live imaging of interphase-arrested zebrafish treated with Aurora kB inhibitor. Right: Plot of the microtubule density interface shown

in dashed green over time. *e* Live imaging of zebrafish treated with Aurora kB inhibitor. *f* Interphase-arrested *Drosophila* embryos and cytoplasmic droplets treated with RANQ69L.

Minor comments:

1) The manuscript could benefit from clearer definitions of the cytoplasmic compartmentalization discussed. Is the term referring solely to ordered aster positioning, or does it also involve the creation of a distinct cytoplasmic region by the patterned microtubule network that for example limits protein mobility within the cytoplasm?

We thank the Reviewer for suggesting a better definition of the compartment. The term indeed involves creation of distinct cytoplasmic regions. These regions are formed by microtubules and dynein motors. Microtubule asters grow and once microtubules from different asters meet, they form a boundary. Since microtubule asters have a polarity, dynein motors transport organelles to the center of the aster, creating a distinct cytoplasmic region dense in organelles. We have now included an illustration in Figure 1A to clarify this process,

[FIGURE REDACTED]

and a textual addition:

Text change: During the first rounds of cell division, microtubule asters divide the cytoplasm into two **distinct** cytoplasmic compartments, **denser regions of cytoplasm rich in organelles**, before the cell membrane ingresses (**Fig. 1A-B**).

While the mobility of membrane-bound organelles is limited to the compartment by dynein, previous work showed that the mobility of proteins in the cytoplasm is not affected by compartmentalization (Huang et al., Nat. Comm. 2022).

2) Given the extension from frog egg extracts and fly systems, it is important to rigorously demonstrate that zebrafish embryos exhibit aster-based subcellular partitioning prior to the formation of physical boundaries (e.g., furrow).

To better visualize compartmentalization of organelles, we have included imaging of mitochondria and lipid droplets in extracts, as well as mitochondria and histones in both unperturbed and cytochalasin B-treated (syncytial) zebrafish embryos. In untreated embryos, we observe clear compartmentalization of mitochondria and coordinated histone segregation prior to furrow formation. Notably, in the syncytial condition, both mitochondria and histones remain compartmentalized despite the absence of physical cell boundaries (Figure R9).

Figure R9 *a* Live imaging of mitochondria of a zebrafish embryo with mito-GFP. *b* Quantification of the signal of mitochondria shows that the boundary is formed prior furrow formation. *c* Live imaging of mitochondria of a cytochalasin B-treated zebrafish embryo with mito-GFP. *d* Live imaging of histones of zebrafish embryo showing division of the histones prior furrow formation (*e*). *f* Live imaging of histones of cytochalasin B-treated zebrafish embryo, showing compartmentalization of the histone signal. See Video 1 for all the relative videos of zebrafish embryo. *g* Imaging of mitochondria in a cytoplasmic compartment in extract and quantification of the radial density, together with the tubulin signal. *h* Imaging of lipid droplets as dark circles in a cytoplasmic compartment in extract

Comparing movie data of division events with and without actin inhibitors, visualization of membranes, etc. could provide information. Providing images of actin in intact zebrafish embryos would also help confirm the efficacy of the drug treatment.

We provide the visualization of the cell membrane and video (Figure R10 and Video1). The visualization of actin in intact zebrafish embryos is reported together with the imaging of the mitochondria in Video 1.

Figure R10 Live imaging of the cell membrane (top) and microtubules (bottom) of a zebrafish embryo and cytochalasin b-treated zebrafish embryo by PH-Halo and EGFP-Doublecortin, respectively.

Data indicating the syncytium until the 32-cell stages (line 33, p. 2) are not provided.

We wrote that the embryo is in a syncytium until the 32-cell stage because the cells on the top of the yolk are not fully closed by a cell membrane, however we realize that using the term syncytium in this context can be confusing. We have removed the sentence.

3) Some of the Supplementary videos appear to be mislabeled in the main text and in the initial frames of the movie files. Text change 4) Page 4, line 13. According to the previous lines, it seems that the morpholino "increases" rather than "decreases" cell cycle duration and aster invasion.

We thank the Reviewer for pointing this out. We have corrected the video labels and clarified and that the morpholino decreases cell cycle duration.

5) The manuscript could discuss more explicitly how inhibition of cyclin production might affect other factors, such as the downstream targets of CDK. Is there any evidence of entrainment of aster dynamics by cell cycle regulators or vice versa? Or do they just work independently? Since this point is highlighted in several parts of the main text, it is helpful to describe more about the potential interplay between cell cycle and subcellular partitioning.

To test if aster dynamics is affected by cycloheximide, we compared the values of polymerization velocity in the initial manuscript and also turnover in the revision in the case of cycling and arrested extract. We found that these values are comparable, suggesting that there is no effect from the cell cycle oscillator to microtubule dynamics (Table S7).

Moreover, in Video 4 (that was in original manuscript) we added cycloheximide locally in the extract and show that the drug does not affect the overall growth of the asters. We wrote in the main text: "Cycloheximide did not affect the speed of microtubule polymerization (**table S7**) and overall compartment growth". In conclusion, we think that the two processes are independent but must have been tuned to each other such that the timing of the cell cycle and formation of

boundaries is of the same scale, to avoid the instability and cover the embryo space at the same time. We have summarized this point in the Discussion in the initial manuscript: “This instability imposes a delicate balance between the waves of autocatalytic growth and the cell cycle timing. The cell cycle duration needs to be slow enough for waves to read the geometry of the cell but fast enough that compartments do not fuse.”

6) Figure 3 and Table S7. It is surprising that different microtubule regulators result in essentially no change in the microtubule turnover rate.

In the initial manuscript, we measured the value of turnover for the sperm nuclei asters and assumed the same value for the AurkA bead and AurkA Ran conditions given that all experiments were conducted in frog egg cytoplasm. Following the Reviewer’s comment, we recognized the importance of directly quantifying aster parameters—specifically turnover and polymerization velocity—for each condition. We have now performed these measurements and updated the phase diagram accordingly (see Table S7 and Figure R11). The data confirm a tight regulation of turnover across conditions, suggesting that microtubule dynamic instability is governed primarily by factors intrinsic to the cytoplasm rather than by the nature of the nucleating center. This conclusion is further supported by additional experiments in which purified centrosomes from HeLa cells and *Drosophila* embryos were introduced into the extract (R13 a-b). These centrosomes produced asters with polymerization and turnover values comparable to those of sperm-derived asters. We report the new phase diagram in Figure R11.

Figure R11 Phase diagram of the inferred difference between autocatalytic nucleation and turnover ($\alpha' - \theta$) and turnover alone resulting in the two stable (orange) and unstable (blue) regions. Dots correspond to the numerical solutions of Eq 1 and the black line represents the stability criteria $\alpha' = \theta$. Error bars represent the 95% confidence interval of the mean calculated through bootstrapping (see SI).

Experimentally perturbing the turnover rate could further validate the model used to generate the phase diagram and strengthen the conclusion.

We initially tested commonly used microtubule-depolymerizing and stabilizing drugs, such as Colchicine and Taxol, but found an all-or-nothing response on microtubules, and either had no effect or depolymerized (stabilized) all microtubules in our experimental systems. To achieve a milder modulation of microtubule dynamics, we introduced purified MCAK-Q710, a kinesin

known to influence microtubule dynamics (Needleman et al., Mol Biol Cell 2009 and Ishihara et al., 2016). Consistent with previous findings (Ishihara et al., 2016), MCAK-Q710 addition reduced microtubule polymerization velocity by approximately 20%.

Using single-molecule speckle microscopy, we measured microtubule turnover in MCAK-Q710-treated asters and found no significant change in microtubule lifetime, in agreement with earlier indirect measurements (Ishihara et al., 2016). As turnover remained constant while polymerization velocity decreased, microtubule length was reduced. Based on our model, we predicted how key parameters would shift: α (that changes linearly with v_p) and ρ_s (that changes inversely with v_p), see SI. Using these predicted values, along with the measured values of v_p and θ , and fitting only ρ_0 (the density near the centrosome), we predicted that MCAK-Q710-treated asters would fall within the stable region of the phase diagram—achieving stability purely through altered microtubule dynamics. We validated this prediction by live imaging, observing that MCAK-Q710 asters exhibited a smaller initial size and a reduction in invasion events compared to the control case (Fig. R12 and Video 10).

Figure R12 **a** Experimental microtubule density profile of asters (green and black curves) and theoretical prediction (orange) for asters in frog egg extract treated with MCAK-Q710 **b** Microscopy image of an aster treated with MCAK. Microtubules are in green. **c-d** Comparison of effect of MCAK-Q710 on aster size in control and MCAK-Q710 treated extract (Video 10).

7) The authors estimated the value of alpha by measuring the initial growth of asters (line 26, p. 3) or from the steady profile at the center (p. 15 in Supplementary Info).

We realized that our original description in the text may not have been sufficiently clear, potentially leading to confusion about how we estimated the model parameters. Specifically, when referring to “initial growth,” we were actually describing the region of the steady-state density profile near the centrosome, where asters are not yet interacting. Our rationale was that in this region, the effects of mutual inhibition are negligible, allowing us to isolate and estimate the nucleation parameters. Once nucleation was determined, the inhibition term could then be inferred from the slope of the density profiles at the interface between asters (without fitting). We initially considered this a more stringent validation of the model than a global fit, as it imposed stronger constraints. However, we recognize that this approach may have been unclear. To address this, we

have now decided to perform a global fit of α , ρ_s , λ and ρ_0 , with no constraints and bootstrap the confidence intervals of the non-measured parameters. This revised approach leads a significantly improved fit to the experimental data (Figure R3a and 1n, 3c,e and 4e,k in the main text).

It is likely that this value is assumed to be constant over time and space in their model. Is this assumption reasonable?

The assumption of constant α over time in our model builds on the current understanding of microtubule nucleation at the microscopic level, which has been validated by previous work (Petry et al, Cell 2013 and Ishihara et al., 2016), including work from our group (Decker et al, Elife 2018). In these studies, microtubule nucleation is described as a process in which nucleators bind to existing microtubules and, once bound, initiate the formation of new microtubules. The kinetics of this process depends on the total microtubule mass and available pool of nucleators, resulting in nucleation rates that change as a function of time. These changes are captured in our effective nucleation term, as it is directly derived from the underlying microscopic kinetic equations. Furthermore, our model quantitatively predicts how the timing of invasion—a purely dynamic feature—depends on differences in aster mass.

8) The “Drosophilization” experiments presented in Fig. 5 are somewhat unclear and would benefit from a more controlled setting. This could be done by the addition of purified centrosomes in a chromatin-free condition, for example. The images presented in Fig. 5 also need clarification, especially the white broken lines in panels A and B, which are not explained.

We incorporated purified centrosomes from HeLa cells and observed a more physiologically relevant *Drosophilization* process (see video 14 and Fig. R13).

Figure R13 a-b Microtubule density profile of interacting asters from HeLa and *Drosophila* centrosomes. Black and green show the density profiles measured experimentally of the asters and magenta shows the theoretical prediction. and Representative asters for these conditions are shown on the right. **c** Imaging of the “Drosophilization” process

with asters from *HeLa* cell centrosomes in extract. Dashed lines indicate the area in the cytoplasm occupied by the asters.

Furthermore, we have clarified the distinct partitioning strategies by directly visualizing the processes in embryos and quantifying cytoplasmic coverage over time (Figure R14a,b,e). Additionally, we have revised the figure caption to include a description of the white broken lines.

[PANELS REDACTED]

Figure R14 *a* Time-lapse of microtubule asters filling the cytoplasm in a zebrafish embryo during division 1 and 2. Grey dashed lines indicate the area for the cytoplasm occupied by the asters. *b* Time-lapse of microtubule asters filling the cytoplasm in a *Drosophila* embryo during division 8 and 9. *c* Time-lapse of microtubule asters in cytoplasmic extract with sparse sperm nuclei where asters grow to mm-size in one cell cycle. *d* Microtubule asters in cytoplasmic extract with centrosomes and slower cell cycle time. See Video 14 for time-sequences. *e*[**TEXT REDACTED**]

9) Some of the experimental data, especially in the later parts of the manuscript, are presented solely as movies without quantification.

Please refer to the answer to the previous comment.

10) In Figure 5C and D, the authors present a diagram explaining how variations in the mechanism of partitioning observed across different species can be attributed to the tuning of a small set of parameters. While this nicely describes the observed phenomena, it would be worth to elaborate on the potential functional or evolutionary advantages of these different partitioning strategies used in embryos of each species.

Like Reviewer 1, we are very excited about the small change in microtubule parameters leading to different patterning strategies and their role in evolution. We believe that this is an important message of the paper and we now elaborate better the functional implications:

Zebrafish strategy: This strategy emerges as a constraint to divide large embryos in half for rapid cellularization and successful development. This strategy is efficient to read the geometry of large cells to find the center and divide the cytoplasmic content in a precise manner. For those large cells, the only viable mechanism to scale asters that reach the boundary with a sufficient MT density is to rely on autocatalytic nucleation. Making microtubules longer by only stabilizing them

would lead to a dramatic decrease of their density when reaching the boundary, impairing robust signaling and mechanical integrity. However, the need to scale asters by nucleation leads to the intrinsic instability of the compartments and imposes a constraint on the timing of the cell cycle and growth of these asters.

Drosophila strategy: The gradual filling mechanism driven by small asters provides greater flexibility over the geometry, timing, and directionality of cytoplasmic exploration. This is extremely important for insect embryos where, while progressive filling is universal, timings, speeds, and paths of filling can be very different (Donoughe et al., Nat. Comm. 2022). In *Drosophila*, for example, this is essential for the correct positioning of asters at the cortex that is driven by cytoplasmic flows (Royou et al, J. Cell Biol., 2002 and Deneke et al., Cell 2019). Biochemically, this strategy could benefit syncytial systems by enabling a patterned, inhomogeneous distribution of cytoplasmic content, for example by creating local concentrations of maternal factors responsible for axis body formation. Additionally, it allows growth without the need for scaling, which may offer flexibility in the spatial and temporal reorganization of intracellular resources.

For example, does the high stability of fly embryos against cell cycle delays with small asters confer any specific advantage over the embryogenesis of other species?

Like the Reviewer suggests, the high stability of the asters in *Drosophila* allows it to have a developmental program without divisions synchronized either in space or time and in a syncytium, for example the cricket embryogenesis (Donoughe et al., Nat. Comm. 2022). In *Drosophila*, this is important because the very first divisions are not synchronized (Deneke et al., Cell 2019). We can speculate that this mechanism allows to maintain multinucleated development throughout evolution and can be important for other syncytial systems. Interestingly, in some syncytia, such as the parasite *Plasmodium falciparum*, asynchronous divisions have the functional advantage of faster nuclei proliferation before cellularization (Klaus et al., Sci. Adv. 2022).

We have now integrated these functional advantages in the discussion as follows:

This mechanism provides greater flexibility over geometry, timing, and directionality of cytoplasmic exploration. This strategy is especially advantageous in insect embryos, where the progression of aster-driven filling is universal but adapted across species, with variable timing, speed, and paths³⁵. For example, in *Drosophila*, correct positioning of asters at the cortex depends on cytoplasmic flows^{36,37}. The gradual nature of this partitioning strategy may also allow for localized biochemical patterning, creating inhomogeneous distributions of maternal factors critical for body axis formation.

The stability of asters in *Drosophila* embryos supports a developmental program without the need for strict spatial or temporal synchrony of divisions, a feature shared with other syncytial organisms³⁵. Such stability is important because in some cases, asynchronous divisions may even offer a selective advantage: for example, in the malaria parasite *Plasmodium falciparum*, asynchronous nuclear divisions in a shared cytoplasm are thought to enable rapid proliferation before cellularization³⁸. Overall, stability of asters observed in *Drosophila* embryos may have allowed the evolutionary maintenance of multinucleated development across diverse species.

Referee 2

This paper by Rinaldin and colleagues looks at the ability of microtubule asters to partition cytoplasm in a syncytium. The work is original. In addition, it is important because, while it is usually assumed that multicellular organisms are subdivided into compartments by cytokinesis during development, in many systems (from malaria to many plants, flies and fish) there are periods of development in which nuclear division occurs without cellularization. In these cases, compartmentalization relies on organization of the cytoplasm itself. While very few have studied the problem in any detail, the phenomenon was observed in the 1800s, when Sacks described these compartments as “energids”. This paper is one of very few to attempt to understand how this type of cytoplasmic organization is achieved using microtubules emerging from centrosomes.

We thank the Reviewer for their encouraging words. Prompted by this comment, we have expanded the discussion to better convey the significance of our findings as follows:

How cells establish and maintain their individuality is a fundamental question in biology. While it is often assumed that multicellular organisms achieve compartmentalization through cytokinesis during development, many organisms undergo extended developmental stages in which nuclear division occurs without cellularization. In these syncytial contexts, compartmentalization depends not on membranes, but on the spatial organization of the cytoplasm itself. This phenomenon was recognized as early as the 19th century, when Sachs described discrete cytoplasmic domains around individual nuclei as “energids”³³. However, the mechanistic basis of such organization has remained largely unexplored.

In this study Rinaldin and colleagues show, very beautifully, that there are two distinct modes by which this can be achieved by microtubules that stop growing when antiparallel microtubules meet. 1. Microtubule asters can rapidly grow (through explosive nucleation) to fill the entire available cytoplasm (Zebrafish or *Xenopus* like). 2. Microtubule asters can locally partition the cytoplasm, so that it gradually fills with asters over time (*Drosophila* like). The comparison presented in the paper that studies both regimes is elegant, and the use of a combination of experiments and modelling, enlightening. Remarkably, the authors find that these two regimes differ in their stability over time. While the second regime appears relatively stable, under the first regime, the boundaries between neighboring asters gradually coarsen. Nevertheless, since it takes time for the boundary between compartments to break down, the authors show that the first strategy is a viable one if the cell cycle period is relatively short. This beautiful observation helps to explain the trade-offs an organism must make when using one of these two regimes to organize the cytoplasm of the developing zygote prior to cellularization: Fast, global and unstable, versus slow,

local and stable. Overall, this study should interest a very wide audience of biologists interested in cell biology, development, evolutionary biology and biophysics.

We thank the referee for the kind words and useful suggestions which we address in the following responses.

Comments:

1. Other factors also likely differ between the two regimes. It would be good if these could be fleshed out in more detail. For example, the two regimes also seem to differ in the density of microtubules filling the cytoplasm, and the sharpness of the boundary between asters. Does this have consequences on the ability of the two different systems to compartmentalize the cytoplasm in the two cases? Also, does the effective mesh size of the microtubule network needed to partition the cytoplasm depend on the size of the object we need partitioned (e.g. lipid droplets/vesicles/mitochondria/ER?)

We thank the Reviewer for this interesting point. Although the stability of the compartments is mostly given by turnover and nucleation, the formation of the boundary requires inhibition, and some structural properties of the microtubule asters and compartments are determined by other factors. For example, as the Reviewer correctly points out, we find a much higher density of microtubules close to the center for stable asters (Bead, *Drosophila*, and centrosome cases) than unstable asters (sperm nuclei, Ran, and zebrafish). This difference occurs because in the stable regimes the density decays from the centrosome, while in the unstable ones the density increases until saturation, leading to a homogenous distribution of microtubules. Moreover, the sharpness of the boundary between asters depends on turnover but also inhibition. However, since cytoplasmic compartments occupy a smaller region than the microtubule asters (Figure 1j, Figure 2a, Figure R15a and d), the inhibition structure between the asters does not affect the boundary of the cytoplasmic compartments themselves. We now show some plots comparing of the radial density profiles of microtubules and organelles for both stable (Figure R15c) and unstable cases (Figure R15g for lipids and Figure R9h for mitochondria). Generally, we observe partitioning of all the membrane-bound organelles that Reviewer 2 mentions (lipid droplets/vesicles/mitochondria/ER, see Figure R9g for mitochondria and R9i for lipid droplets), but not free proteins (as was also shown by Huang et al., Nat. Comm. 2022) or condensates (unpublished). We note that the dye that we use for lipids in magenta in all images labels all the membrane-bound organelles, for example the ER. We also show now compartmentalization of mitochondria in zebrafish embryos prior to and without cytokinesis. Finally, we recognize that it is difficult to appreciate the interfaces between asters of Aurora kA beads in the figures in the original manuscript because of the low imaging resolution that was used to show asters stable over time in a large number. We included some videos at higher resolution in the initial manuscript and have now added some higher

resolution images (Figure R15a,b) where it is possible to appreciate the antiparallel microtubules at the interface.

Figure R15 *a* Maximum intensity projection of microtubules (green) and membrane-bound organelles/lipids (magenta) of *Aur kA* bead asters. *b* Enlarged image of the interface in *a* shown by the dashed rectangle. *c* Radial density of microtubules and lipids of the *Aur kA* bead aster in *d*. *e-f* Maximum intensity projection of microtubules (green) and membrane-bound organelles/lipids (magenta) of a sperm nuclei aster. *g* Radial density of microtubules and lipids of the sperm nuclei aster on the right.

2. The self-organising patterning process described by Turing involves the short range diffusion of an autocatalytic signal and a longer range diffusion of an inhibitor. This is inherently STABLE because both diffuse away from the same source. In the case studied here, there is spreading autocatalysis, together a special kind of local inhibition at regions of overlap. Thus, while highlighting the differences, I would strongly advise against calling this Turing patterning. It will just confuse readers and muddies our notions of "Turing patterning".

To avoid the confusion, we have removed: “similar to Turing patterns” in the introduction and changed “Turing-like” with “**general reaction-diffusion mechanisms**” in the conclusions.

3. The colors and symbols used in the diagrams are hard to see. As examples of this, the gradients in color that denote time are very hard to discern in 1J and 2F.

We have now used better colors in all the figures and used some clearer color maps to show the time for the phase portraits in Figure 2.

4. How exactly was the experimental data shown in 2D generated? I would expect to see organisms keeping a tight grip on the heterogeneity of nucleation. Is this the case? Can this be quantified and shown?

During sample preparation, we naturally introduce variability in microtubule aster mass. Specifically, microtubule asters nucleate around sperm nuclei that are heterogeneously distributed within the imaging field. Because asters grow until they encounter neighboring asters, their final size and microtubule mass depend on the relative distances between nuclei. We explain this now better in the main text by adding: **We experimentally introduce differences in aster mass by exploiting the local variations of sperm nuclei densities in the imaged sample.**

Embryos on the other hand, maintain a tight control on nucleation and microtubule aster mass and size by using active centering mechanisms for the nuclei. We have now measured the differences in mass between the asters in a syncytial zebrafish embryo (cyto-B treated) and added it to the existing curves.

Figure R16 Probability distribution of the mass differences between asters. The arrested compartments show a distribution that is less peaked than the one of the cycling compartments. Moreover, over time as more compartments invade, the distribution of mass in the arrested cytoplasm becomes flatter. The zebrafish syncytial embryos shows a more peaked distribution as is expected from the tight aster size regulation observed in vivo.

5. As mentioned above, in the model the different regimes seem to also differ in the sharpness of the MT boundary and the density of microtubules at the boundaries between asters. Because of this, it would be good to show how MT densities (and density gradients) scale with changes in parameters, and what consequences this has for cytoplasmic partitioning. For example, in Figure 3D there seem to be no clear boundaries between compartments, therefore no precise partitioning. Is this correct?

We realize that it is difficult to appreciate the interfaces between asters of Aurora kA beads in the figures in the original manuscript and indeed we did not include imaging of the cytoplasmic compartments there. Now, we have added higher resolution images (Figure R15a,b) where it is

possible to appreciate the antiparallel microtubules at the interface and imaging of the lipids where it is clear that asters from AurkA beads can also drive partitioning.

6. While it is good that the paper includes supplemental material with more precise perturbations. Nevertheless, its a pity that the mechanisms used to inhibit the cell cycle are so crude. Also, inhibiting protein synthesis and degradation with have numerous side effects. Why not simple add non-degradable cyclin to extracts to stop the clock? Highlighting the more specific perturbations in the body of the paper would really help make the case that what is being measured is a specific consequence of the impact of the perturbations on the cell cycle.

We thank the reviewer for their suggestions and indeed we have tried to add non-degradable cyclin B (Cyclin B1 full protein and Cyclin B1 - Δ 90), however this perturbation arrested the extract in metaphase and not interphase, consistent with previous findings (Chang et al., J. Biol. Chem. 2003). We primarily relied on cycloheximide in most experiments because its strong ability to fully arrest the cell cycle in interphase. In early development, both in extracts and embryos, cyclin levels are extremely high, making it difficult to achieve a complete cell cycle arrest. At lower concentrations of cycloheximide, we observed slower or desynchronized cell cycles (unpublished), and in embryos, delayed cycles or centrosome progression without corresponding nuclear divisions. Similar issues arose when using morpholinos. To arrest the extract in interphase more specifically, we also conducted experiments using morpholinos against Cyclin B1 and Cyclin B2. These are commonly used in extract for targeted cell cycle studies (Tsai et al., Curr. Biol. 2014 and Maryu et al, Cell Rep. 2022) and have been verified to modulate cell cycle durations with cell cycle FRET sensors. However, even by using high concentrations of morpholinos (about 1/6th of the volume in extract of undiluted samples) we could not observe a full cell cycle arrest, only a slower cell cycle as shown in the Video in the initial manuscript. To overcome this, and based on prior work, we encapsulated the extract with morpholinos in droplets. This allowed us to create a heterogeneous distribution of morpholino concentrations, with some droplets containing sufficiently high levels to produce robust and observable invasion events (Figure R6a, R7g, and Video 7).

7. What are ranges of cell cycle times during divisions in early developing fish and frogs embryos? Do these rates scale with MT growth rate, so that more instability correlates with a faster cell cycle time? I think such measurements are an important test of the theory.

The cell cycle times of frog, zebrafish, and *Drosophila* embryos are 30, 15, and 8 min. If we plot these data against the microtubule polymerization velocity, we observe an increase of polymerization speed with cell cycle time, however the data points are not enough to determine if there is any scaling.

We expect more instabilities for a slower cell cycle time, as we suggest from the plots of the size distribution of the compartments over time (Fig. 2G and H) where there is a larger variation in size and larger size of compartments for slower cell cycles.

8. Does the MT growth rate affect cell cycle timing to ensure the two systems remain coupled?

This point was also raised by Reviewer 1 in their minor comment 5. The cell cycle and the microtubule growth rate are independent. We compared the values of polymerization velocity and turnover in the case of cycling and arrested extract. We found that these values are comparable, suggesting that there is no affect from the cell cycle oscillator to microtubule dynamics.

Moreover, in Video 4 -that was in the original manuscript -we added cycloheximide locally in the extract and show that the drug does not affect the overall growth of the asters. We wrote in the main text: “cycloheximide did not affect the speed of microtubule polymerization (**Table S7**) and overall compartment growth”. In conclusion, we think that the two processes are independent but must have been tuned to each other such that the timing of the cell cycle and formation of boundaries is of the same scale, to avoid the instability and cover the embryo space at the same time. We have summarized this point in the Discussion in the initial manuscript: “This instability imposes a delicate balance between the waves of autocatalytic growth and the cell cycle timing. The cell cycle duration needs to be slow enough for waves to read the geometry of the cell but fast enough that compartments do not fuse.”

9. Specific comments on Figures. Figure 1C – would be good to show actin as suggested in Figure legend.

We have now added an image of actin in the main text showing the compartmentalization at the cortex (Figure R17). Moreover, we have added images of the actin during an invasion event, showing the fusion of such “cortical compartments” (Figure R6b).

Figure R17 Live imaging of cytochalasin B-treated zebrafish embryo where cell membrane ingression is inhibited. Asters coexist and form boundaries of low microtubule and actin density.

Figure 1J – would be useful to show change in boundary over time, to make clear the rate at which the destabilizing of the boundary occurs.

We have now added a plot of the interface position over time in Figure 1 (Figure R18).

Figure R18 Numerical time evolution of the theoretical microtubule densities shown by the black line in Fig1n of the main text. The time evolution shows that the system is unstable as the left aster invades the right one in less than one hour. The position of the interface over time is shown in the inset.

Figure 1K seems to be missing.

We have corrected the caption.

Figure 2A – the boundary between cytoplasmic regions seems to become even more prominent over time. Is this predicted from the in silico analysis? How do the authors explain this?

The boundaries between microtubule asters on the top figures become clearer over time because the number of polymerized microtubules increases over time, decreasing the amount of noise in the image given by free tubulin. This is a visual effect that is due to the fluorescence imaging. The boundaries between the cytoplasmic compartments at the bottom become clearer over time because there is more time to partition organelles within the compartments. This effect is also visible in cycling extract (Fig 1J).

Figure 2A – it seems to me important to show what happens when two compartments fuse to show and explain the dynamics of nuclei repositioning, along with lipid and MT reorganization.

Also, in response to the point on mechanics of Reviewer 1, we show now the dynamics of repositioning of the cytoplasmic compartment and nuclei during an invasion event (Figure R5).

Referee 3

How the cytoplasm is partitioned in the absence of cytokinesis in early developing embryos is an important and long-standing problem. This topic has received renewed attention, thanks to recent work by Cheng and Ferrel (Science, 2019), who showed that the cytoplasm can organize itself into regular, cell-sized compartments through Microtubule (MT) asters growth and dynein activity. In this new study, Rinaldin and colleagues propose that an inherent instability, caused by a

competition between self-amplifying MT growth and turnover, affects the robustness of MT aster-based cytoplasm compartment size definition. Interestingly, the impact of this instability in early embryos might be limited by the rapidity of cell cycles (like in fish embryos or xenopus extracts) or by restricting microtubule (MT) nucleation (as seen in drosophila embryos). A mathematical model predicts the origin and limitation of this instability, and this prediction is tested through various experiments and quantitative assays, as well as 2D simulations. This model and its various tests represent the main advancement of this work over the current state of the art. Overall, this work provides an interesting and potentially universal mechanism that could explain how membrane-less syncytia can become organized.

We thank the referee for the kind words and useful suggestions which we address in the following responses.

My main concerns are listed here:

- Generally speaking, the use and direct comparison between different model systems, although interesting, is a limitation of this work. These organisms are quite divergent from an evolutionary standpoint, which raises concerns about how the molecular regulation of MT autocatalytic growth, nucleation, or repulsion might be conserved among them. In the paper, fish embryos and Xenopus extracts are treated with actin inhibitors, while *Drosophila* embryos have intact actin networks. Moreover, most of the work is done in Xenopus extracts using methods and materials similar to the Cheng and Ferrel paper; yet, the abstract only mentions results in fish and fly embryos, which are much less explored in the MS. This is surprising.

We strongly believe that the comparative embryology approach allowed us to understand the roles of the instability and autocatalytic growth in development. Moreover, while we did not examine the molecular mechanisms of the autocatalytic nucleation, which we believe are beyond the scope of this paper, we found that the stability just depends on the effective nucleation itself, regardless of how that is regulated. By quantifying the physical parameters of microtubule dynamics, we found that microtubule turnover is a very conserved quantity while nucleation is modulated across these species.

These discoveries could be very important to restrict the analysis of molecular mechanisms regulating interphase asters in these organisms. For example, we believe that it could be very interesting to examine the phosphorylation state of Augmin in these *Drosophila* asters, since it is required for autocatalytic nucleation in spindles (Tariq et al., Elife 2020). While finding the molecular details across the species is interesting for future work, we still tried to test some factors to show the generality of the two stable regimes.

By injecting non degradable Ran (RanQ69L) in *Drosophila* embryos, we can now show the formation of large asters, pointing out that the Ran pathway might have a more general role in regulating asters in interphase. Moreover, we inserted centrosomes from *Drosophila* embryos in extract and we observe formation of asters with a limited size and decreasing profile, suggesting that the factors in the centrosomes are critical to determine the nucleation profile.

We have also treated embryos with Cytochalasin B for consistency with experiments in extract and zebrafish and we report examples of cell cycle arrest in combination with actin inhibition in the main text and SI. We also note that previous studies already tested that the actin does not play a role during the mitotic divisions in the embryo before the nuclei reach the cortex: “Although actin caps and metaphase furrows have been implicated in maintaining the fidelity of nuclear division and the positions of nuclei within the cortex, our observations indicate that these structures are dispensable during the early syncytial blastoderm cell cycles.” (Postner et al., *J. Cell Biol.* 1992).

Finally, we were so excited about the embryo and explaining the different strategies regulated by nucleation that we inadvertently downplayed the extract part in the abstract. We agree with the Reviewer that we used extract methods similar to Cheng et al., 2019 *Science*, however we want to stress that the extract from the frog eggs was only a tool to make a discovery that help us to better understand embryonic development. We have now added extract in the abstract and we thank the reviewer to point this out. We have also now significantly extended the *in vivo* work.

- The key novelty of this work is the mathematical model that predicts the existence of an instability where one aster can grow at the expense of another. The model has several shortcomings. First, it is a 1D model, which contrasts with the 2D and 3D experimental situation, where multiple asters interact in all directions. Second, the fitting of the model to experimentally measured MT density relies on circular arguments, as some model parameters are inferred from fitting experimental curves (Fig 1H). Third, the general term used for the inhibition between asters accounts for the density of competing asters but lacks a clear description of the depolymerizing effect of kinesins at aster-aster interfaces.

We thank the Reviewer for these important comments on the role of dimensionality, fitting, and details of inhibition in our model. We have address them each of them. We repeat here the same response we had for Reviewer 1 on Dimensionality.

Dimensionality. In the initial submission, we simplified the problem to 1D because microtubule asters maximally interact at the midzone region (Nguyen et al. *Science*, 2014 and Glotzer et al., *Science* 2005), which is the area where antiparallel microtubules from two asters overlap. While the midzone region is a three-dimensional object, microtubules mostly align along one direction, suggesting that the interaction is symmetric along the interface between the asters. We could see

this by measuring the density profiles across the boundary as a function of the position along the perpendicular direction to the boundary, which do not change over 80 μm along the boundary (*Figure R1a-b*). We have now also analyzed the microtubule density in 3D by measuring the average tubulin intensity profile along the xy and xz directions. We find that there is no substantial change of microtubule density across the interface also in the third dimension (*Figure R1c-e*)

This suggests that the problem of aster-aster interactions is essentially one dimensional and explains why the 1D model can faithfully reproduce the experimental microtubule density profiles in all conditions. We then employed 2D simulations to investigate the role of geometry, which did not reveal any significant effects since we could recapitulate the experimental density profile and invasion times observed in the experiments and theory.

However, we agree with Reviewer 3 that dimensionality can play a role in modelling microtubule density in space, therefore we extended the 1D theoretical model to 2D and 3D and completely remade our agent-based simulations in 3D. By extending the theoretical model to higher dimensions, we discovered that dimensionality mostly plays two roles in our system:

- 1) **Geometric decrease of microtubule density.** Due to radial symmetry, asters expanding in 2D and 3D exhibit a geometric decrease in microtubule density. This leads to a change of the flux term in the form of the following geometrical term:

$$\frac{-(N_d-1)v_p\rho}{r},$$

where N_d is the dimensionality, v_p is the polymerization velocity, and ρ the microtubule density (see new supplementary note). Without nucleation and turnover, this term makes **the density decay in 2D and 3D in comparison to 1D**, which would remain flat. However, because of autocatalytic nucleation and turnover this effect quickly becomes negligible. Beyond a radius $N_d-1 \sim \frac{v_p}{\theta} \sim 5-10 \mu\text{m}$, turnover effects are larger than any geometric decay, which is negligible compared to the typical size of asters of around 100 μm . This effect can be seen for small distances from the center of the aster (*Figure R2a*) where we compare the new fits in 1, 2 and 3D, **but does not affect the shape of the microtubule density profile in the nucleation and interaction region**. The latter overlaps for all dimensionalities (*Figure R2b*). To consider this effect more explicitly, while in the original manuscript we plotted the microtubule density centered at 0 -the point of intersection of the two aster profiles- we now start the x axis at the center of the centrosomes of the left aster. Therefore, the x-coordinate of the first experimental point now corresponds to the distance between the center of the centrosome and where the EB1 comets begin to be measured. We note that the profiles do not start at the centrosome, because it is challenging to count the EB1 comets close to the centrosome due to the high density that prevents to

correctly track them over time. However, we have performed an analysis of the radial counts of comets of the aster in *Figure R2c* without tracking their direction to observe the decay close to the centrosome (*Figure R2b, red points*).

- 2) Probability of interaction for antiparallel microtubules.** The second effect of dimensionality is, as the Reviewer suggests, the impact on the probability for antiparallel microtubules to physically interact at the aster-aster boundary. This probability corresponds experimentally to the decrease in density at the midzone region where antiparallel microtubules meet. Our previous analysis of microtubule density across the interface suggested this decrease in density occurs only along the direction of anti-parallel alignment in 2D (*Figure R1b,d,e*). To evaluate this effect quantitatively, we derived a 3D version of our model and computed the inhibition term from the probability of finding microtubules of opposed polarity (see SI 2.3 Angular and higher dimensionality continuum model). We used this 3D model to re-derive an effective 1D equation near the boundary (because of the axisymmetric symmetry). We found that the effect of dimensionality is captured in renormalized terms, but **the shape of the equation does not change with respect to the original 1D model.**

By extending the theory to **three dimensions, we can recapitulate the same microtubule density profiles that we found in 1D.** We have also tested how the dimensionality affects the compartment stability by plotting the phase diagram of α and θ for the 1D, 2D and 3D case and observe no substantial differences (Extended Data Fig. 11).

To better understand the physical interaction between the microtubules in space, and to confirm the conclusions from the continuum model, we have extended the simulations from 2D to 3D. To computationally support this extension, we implemented several improvements to our simulations: (1) we simulate $\sim 10^5$ - 10^6 filaments, in contrast to the original manuscript where we could only simulate 10^3 - 10^4 filaments. This was essential to maintain the same density of microtubules in the 3D simulations as measured in the experiments.

(2) We include an inhibitory interaction range to account for an effective filament tip search radius and/or protein binding and diffusion effects for more realistic modeling of inhibition.

(3) We increase the simulated real time from 30 minutes to ~ 2 hours allowing us to reach timescales seen in experiments. This is accomplished in significantly less time (~ 1 -7 days of computational time) thanks to GPU acceleration which allows for accelerated parameter optimization and explorations.

We explain how these processes are taken into account in the SI 3.2 . We show that the microtubule density profile in 3D agrees with the experiments (*Figure R3a*) and the 1D theory and that we

can recapitulate the aster-aster interaction (*Figure R3b*) and invasion event with a strikingly similar dynamics to the one observed in the experiments (*Figure R3c*).

Altogether, the extension of the theory to higher dimensions, and the new 3D simulations allowed us to **fully explore in detail the role of dimensionality in the stability of the compartments**. We find that the 1D description, complemented with the 3D simulations, is appropriate for studying the interaction between asters. This approach simplifies the calculations and provides conceptual clarity to our manuscript.

Fitting. We realized that our original description in the text may not have been sufficiently clear, potentially leading to confusion about how we estimated the model parameters. Specifically, when referring to “initial growth,” we were actually describing the region of the steady-state density profile near the centrosome, where asters are not yet interacting. Our rationale was that in this region, the effects of mutual inhibition are negligible, allowing us to isolate and estimate the nucleation parameter. Once nucleation was determined, the inhibition term could then be inferred from the slope of the density profiles at the interface between asters (without fitting). We initially considered this a more stringent validation of the model than a global fit, as it imposed stronger constraints. However, we recognize that this approach may have been unclear. To address this, we have now decided to perform a global fit of $\alpha, \rho_s, \rho_0, \lambda$ without any constraint and bootstrap the confidence intervals of the non-measured parameters. This revised approach leads a significantly improved fit to the experimental data. Refer to section 2 of the Supplementary Information.

Inhibition. Our model is a continuum coarse-grained description, which by construction neglects microscopic details. However, we demonstrate in the manuscript that any inhibitory function acting symmetrically on both asters—that is, inhibiting the growth or increasing turnover of both asters equally—will lead to the same instability, according to linear stability analysis (SI 2.1). This symmetry condition includes any type of microscopic mechanisms, including, for example, kinesins acting at antiparallel overlaps or signaling molecules binding directly at these sites and triggering decrease of polymerization rates or depolymerization (Nguyen et al. Mol Biol. Cell 2018). Importantly, any mechanism that enhances microtubule (MT) depolymerization can be effectively captured by an increase in MT turnover. Such effects generally take the form of a function—possibly dependent on MT density—multiplied by MT density, matching the structure of our inhibition term. This is very important since it is unclear how exactly factors at the midzone affect microtubule dynamics (Nguyen et al. Mol Biol. Cell 2018). The reason why it is still not clear is that it is extremely hard to quantify the structure and dynamics at the interface between the asters, as this would require electron microscopy methods and super resolution fluorescence microscopy, respectively. In this context, we believe that the new simulations that include a much higher number of microtubules and more realistic interface can provide a platform to explore more detailed microscopic implementations of inhibition. For example, we have inferred a minimum length scale ($\sim 2 \mu\text{m}$) at which the inhibitory mechanism must function to establish an aster

boundary. In the current manuscript, we have implemented a contact inhibition mechanism, one of the leading models believed to operate in our system. In future work, combining our continuum framework with agent-based simulations will allow us to investigate how specific molecular processes—such as angle-dependent inhibition, branching dynamics, or particular binding interactions—contribute to the observed behavior. Nevertheless, as we show, the general stability of the boundary emerges from a competition between nucleation and turnover and is robust to the specific form of inhibition, as long as the symmetry condition is maintained (Supplementary Information 2.1).

We also highlight that the interfaces that we find with the new set of simulations look much more similar to the experiments. We provide here a direct comparison (Figure R19).

Figure R19 Comparison of agent-based simulations of two interacting asters of the previous manuscript (2D) and current manuscript (top view of 3D). White box of 3d simulation is the zoomed in view on the right.

- Many tests of the effect of cell cycle duration are done using cycloheximide both in vitro and in vivo, which is likely to also affect MT dynamics and organization. It would be important to better rule out any side effects of this treatment. A control using morpholino to affect cyclins B1 and 2 is provided in video6, but the extracts appear to behave very differently than in cycloheximide, with “invasion” events hard to detect. The same concern applies to experiments with RanQ69L, which may affect other aspects of MT dynamics beyond nucleation.

We repeat here the response on this topic that we had to the Reviewers 1 and 2.

To test if aster dynamics is affected by cycloheximide, we compared the values of polymerization velocity in the initial manuscript and also turnover in the revision in the case of cycling and arrested extract. We found that these values are comparable, suggesting that there is no effect from the cell cycle oscillator to microtubule dynamics (Table S7).

Moreover, in Video 4 -that was in original manuscript- we added cycloheximide locally in the extract and show that the drug does not affect the overall growth of the asters. We wrote in the main text: “Cycloheximide did not affect the speed of microtubule polymerization and overall compartment growth”. In conclusion, we think that the two processes are independent but must

have been tuned to each other such that the timing of the cell cycle and formation of boundaries is of the same scale, to avoid the instability and cover the embryo space at the same time. We have summarized this point in the Discussion in the initial manuscript: “This instability imposes a delicate balance between the waves of autocatalytic growth and the cell cycle timing. The cell cycle duration needs to be slow enough for waves to read the geometry of the cell but fast enough that compartments do not fuse.”

We have tried to add non-degradable cyclin B (Cyclin B1 full protein and Cyclin B1 - Δ 90), however this perturbation arrested the extract in metaphase and not interphase, consistent with previous findings (Chang et al., J. Biol. Chem. 2003). We primarily relied on cycloheximide in most experiments because its strong ability to fully arrest the cell cycle in interphase. In early development, both in extracts and embryos, cyclin levels are extremely high, making it difficult to achieve a complete cell cycle arrest. At lower concentrations of cycloheximide, we observed slower or desynchronized cell cycles (unpublished), and in embryos, delayed cycles or centrosome progression without corresponding nuclear divisions. Similar issues arose when using morpholinos. To arrest the extract in interphase more specifically, we also conducted experiments using morpholinos against Cyclin B1 and Cyclin B2. These are commonly used in extract for targeted cell cycle studies (Tsai et al., Curr. Biol. 2014 and Maryu et al, Cell Rep. 2022) and have been verified to modulate cell cycle durations with cell cycle FRET sensors. However, even by using high concentrations of morpholinos (about 1/6th of the volume in extract of undiluted samples) we could not observe a full cell cycle arrest, only a slower cell cycle as shown in the Video in the initial manuscript. To overcome this, and based on prior work, we encapsulated the extract with morpholinos in droplets. This allowed us to create a heterogeneous distribution of morpholino concentrations, with some droplets containing sufficiently high levels to produce robust and observable invasion events (Figure R6a and Video 7).

We agree with the Reviewer that the RanQ69L as well as the use of AurkA beads, exogenous centrosomes, and MCAK could affect the microtubule dynamics and we measured the properties of microtubule dynamics for each condition. Please refer to Table S7 for the new values and Figure R11 for the new phase diagram.

• A final concern is the relevance of the cytoskeletal instability to *in vivo* situations. Observations from zebrafish embryos (Fig 4E/ Video 9), where the cell cycle is arrested, do not convincingly demonstrate invasion between compartments/asters. There are only a few MTs that cross boundaries, and other parts of aster pairs rather separate from each other (bottom of Fig 4E). This phenotype differs from the *in vitro* situations and simulations. Much more work would be needed to better demonstrate the validity of the model for developing embryos *in vivo*. Could the authors, for instance, manipulate nucleation, repulsion, or other MT properties in *Drosophila*, which is genetically tractable?"

We repeat here the response we had on the *in vivo* invasions and perturbations to Reviewer 1.

Effect of the invasion in zebrafish. We thank the Reviewer for this comment and agree that the invasion was not easily appreciable in the image presented in the main text, which we attribute to the different confinement of the asters. While asters in the extract are confined to a thin layer (Figure R1c), those in zebrafish embryos are enclosed within a spherical cap (Figure 1b), resulting in distinct visual phenotypes of interface dynamics during aster invasion. We have now mimicked the spherical embryo geometry through the encapsulation of extract in oil into small droplets and successfully recapitulated the dynamics of the interface that we observed *in vivo* (Figure R6a and Video 7). We also realized that the invasion dynamics is much clearer on specific planes than in the MIP. We have now included in the main text images of multiple planes, where it is also possible to see the dramatic relocation of nuclei that is seen in extract (Figure R6b, $z = 30 \mu\text{m}$, yellow arrow). Finally, we now also show in the main text the interesting fusion of “compartments” at the cortex that results from aster invasion (Figure R6b, utrophin signal in cyan).

Moreover, we provide images and videos of invasions at different cell stages. In the original manuscript, we showed invasion at 8 cell stage. Now, we provide also example at 16 and 32 (Figure R7a-b), where the asters are smaller but still have an increasing density profile from the center and are predicted to invade. In the invasion event at 32 cell stage (Figure R7b) we can see a larger aster invading a smaller one, as predicted. This event more closely resembles the invasions observed in extract, likely because arresting the cell cycle after several rounds of syncytial division results in greater variability in aster mass and size—conditions that more closely mirror those in extract- and the size of the aster respect to the embryo is smaller, therefore the effect of the confinement in the curved geometry becomes negligible.

We show that when we arrest the cell cycle at late stages (thousands of asters) when the asters have a decreasing density profile corresponding to the stable condition, the compartments remain stable (Figure R7 d). We have also improved the imaging and conditions for cell cycle arrest in *Drosophila* embryo. We use embryos that are treated with cycloheximide and cytochalasin, like the zebrafish embryos and we also show invasion in droplets of *Drosophila* extract (Figure R7 d-e).

Perturbations of microtubules and MAPs in zebrafish. We have performed several perturbations in zebrafish embryos where we observed invasions, suggesting that it is a general phenomenon that can happen by competition of autocatalytic microtubule waves.

- (1) Initially we tried to make small asters by injecting Aurora kA beads and purified centrosomes from HeLa and *Drosophila* embryos, but these did not nucleate asters. However, we managed to create small asters by injecting oil droplets that nucleate microtubules on the surface triggering aster formation. These smaller asters were rapidly invaded by larger ones, consistent with our mechanism (Figure R8a).

- (2) By incubating embryos with a microtubule depolymerizing drug, we have created asters of lower mass at the boundary than the center (admittedly without a good control on the strength of the depolymerization, with the drug typically either depolymerizing all microtubules or having no effect). In this condition we consistently observe invasions and relocation of nuclei from the boundary to the center of the aster. (Figure R8b).
- (3) By inhibiting dynein in the embryo, we perturbed the relocalization of the nuclei prior to the formation of boundaries. This generally leads to asters that are not regularly spaced, and specifically centers that are in very closed proximity to each other. In the presence of autocatalytic waves, this leads to the disassembly of boundaries (Figure R8c).
- (4) By using Barasertib to inhibit Aurora Kinase B, a kinase involved in midzone formation and inhibition between the asters, we biochemically lowered the inhibition. In this way, we could induce invasion, even at 2 cell stage which is very hard to observe in embryos treated with cycloheximide and cytochalasin since the asters have very similar mass of microtubule and size (Figure R8d). Moreover, because Aurora Kinase B is responsible of cytokinesis and lead to a syncytial embryo, we did not use cytochalasin in these experiments. Interestingly, with this perturbation, we have also observed that while an interface between asters is originally formed also in the absence of Aurora Kinase B, if the cell cycle is slower or arrested, the microtubule asters can invade (white arrow in Figure R8e shows an invasion event). This result supports the point of the paper that the interplay between the cell cycle timing and the timescale associated to the creation and maintenance of an interface between asters is key to maintain robust boundaries (Figure R8e).

Altogether, these experiments demonstrate that perturbations to either the relative microtubule mass or the aster interface can trigger aster invasion in zebrafish embryos. Given that the instability arises intrinsically from autocatalytic aster growth, numerous biochemical or physical imbalances between asters can facilitate the onset of this process. We also supplemented these experiments with perturbations in extracts to test the generality of the mechanism (Figure R4).

Finally, as suggested by this comment, we have tried to perturb the nucleation in *Drosophila* embryos, as they are generally more tractable. Indeed, we found that in those embryos it is possible to tune nucleation. Following our results on Aurora kA beads and RANQ69L, we injected RANQ69L in arrested *Drosophila* embryos and cytoplasmic droplets and observed formation of very large asters (Figure R8f). This latter result strengthens the possible role of RAN residual from mitosis on setting the nucleation and size of asters in interphase, which is an interesting nucleation mechanism for future studies. We note that we performed live imaging of the embryos after the injection to observe invasions, however we could observe these large asters for only a short amount of time before they disassembled.

Second revision round: point by point response to referees

Legend:

Blue text: referee comments

Black text: response to referee comments

Green text: additions to the manuscript

We thank very much all three reviewers for their comments that helped us to strengthen the manuscript.

Referee 1

The revised manuscript by Rinaldin et al. presents thorough and comprehensive data, both in experimental and computational, that convincingly address most of my earlier comments. The 3D simulations provide strong support for the authors thesis on boundary robustness and instability, and the new perturbation experiments nicely validate the model's predictions. Overall, this is a beautiful piece of work that proposes the general strategy for embryonic division control and is likely to stimulate broad interest across biology and physics.

We thank the Reviewer very much for appreciating our revisions and their kind words about our work. We have implemented all minor suggestions.

I have a few minor suggestions that may further improve the clarity of the manuscript before publication:

1. Lines 25-27, p. 8, and Extended Data Fig. 13: The text states that aster invasion is enhanced by inhibiting dynein in zebrafish embryos. Since elsewhere the authors explain that dynein does not play a major role in invasion (e.g., line 4, p. 28; Extended Data Figs. 4 and 5), it would be helpful to clarify this probable discrepancy.

We agree with the Reviewer that a clarification is helpful. While dynein inhibition does not block invasion, it can actually enhance it by preventing the centering of nuclei. In dynein-inhibited cycling extract, invasion resembles the control situation; however we note that dynein inhibition produces unfocused branches and splayed asters (**Extended Data Fig 4d**, time 0), consistent with the literature (Scrofani et al., Mol. Cell. Biol., 2023). We propose that because of this, dynein inhibition may enhance invasion. In zebrafish embryos, prevention of nuclei centering by dynein

leads to asters in closer proximity, thereby affecting their interaction. This situation in the presence of autocatalytic waves promotes disassembly of aster-aster boundary (**Extended Data Fig 13c**).

We have added the following text on Page 4, line 27-31, where we first mention dynein inhibition in cycling extract:

“However, dynein activity **does not hinder** the invasion process because invasions occur before active transport of nuclei and organelles as well as when dynein is inhibited (**Extended Data Fig.4-6 and Video 6**). **Interestingly, dynein inhibition enhances invasion by leading to splayed asters, presumably due to the lack of proper pole formation²⁵ (See dynein-inhibited cycling extract in Video 6).**”

We have also added the following text when referring to zebrafish (Page 8, line 27-30):

“We could enhance these invasions by nucleating smaller asters triggered by adding an exogenous oil droplet, by depolymerizing locally microtubules, and by inhibiting dynein **which leads to asters in closer proximity to each other. (Extended Data Fig. 13 and Video 13).**”

2. In the 3D simulation, the invasion of the aster boundary may critically depend on the Inhibition radius r_λ (p. 31, Supplementary Information). If so, how robust are the outcomes to the choice of this parameter? Is the value set to 2 micrometer reasonable in the context of the actual biological situation?

We agree with the Reviewer that the invasion process depends on the inhibition radius, r_λ , which is the radius at which microtubules can interact. This parameter is phenomenological, but given that we measure an average density of microtubules at the interface of ~ 0.04 microtubules/ μm^2 (**Figure 1n**), we estimate that the radius of a circle that, on average, contains one microtubule is between 2 and 3 μm . Therefore, we think that 2 μm is a reasonable choice for the radius at which microtubules can interact. Changing this radius, however, effectively changes the strength of inhibition (because it determines the number of interactions). To test the effects of changes in the interaction radius without modifying the overall inhibition, we rescale λ so that the number of interactions, which is proportional to the volume set by r_λ , multiplied by the inhibition strength remains constant: $\lambda = Cr_\lambda^{-3}$, where C is a constant. Under this rescaling we performed simulations for r_λ ranging from 1 μm to 5 μm (which is of the scale of one full microtubule) and found that the density profiles were robust to these changes (**Figure R1**, that is included in **Extended Data Fig.3h**).

Figure R1 Microtubule density profiles for different values of inhibition radius and scaling of inhibition strength as $\lambda = Cr_{\lambda}^{-3}$.

What parameter(s) were tuned to achieve the stable case?

We have changed the box length—as in this case the asters are smaller—, number of nucleators, and centrosome nucleation rate to decrease microtubule nucleation (See Table on Page 39 of Supplementary Information). However, we did not change the inhibition parameters.

3. Figure 5 caption (panels c-e, lines 5-6, p. 25): In panel c, the caption should explicitly state that the data are from frog cytoplasmic extract. In panel d, the source of the centrosomes should also be specified. In panel e, the labels egg centrosomes HeLa centrosomes are not immediately clear. Are these data obtained with frog extracts as in Fig. 5c/d and Extended Data Fig. 9?

We have improved the caption clarifying the type of samples and the data used in the plot. We have changed the title of the panel in the figure to: “Cytoplasmic partitioning strategies are recapitulated in frog egg extract.” We have added on the left side of the time sequences the labels: “+low sperm nuclei” and “+egg centrosomes”.

We have also modified caption 5 c-e for clarification:

“c Time-lapse of microtubule asters in cytoplasmic extract supplemented with sparse(low) sperm nuclei where asters grow to mm-size in one cell cycle. d Microtubule asters in cytoplasmic extract with centrosomes intrinsic of the frog egg extract and slower cell cycle time. See Video 14 for time-sequences. e Plot of the cytoplasmic volume occupied by the asters over time. Data for zebrafish embryo (light blue dashed line), frog egg extract with sparse sperm nuclei (dark blue line), egg extract centrosomes (orange line), and purified HeLa centrosomes (yellow line) taken from time-sequences in Fig. 5a,c-d, Extended Data Fig. 9 and Video 14. Data of *Drosophila* embryo (red dashed line) taken from Deneke et al³⁸. Dotted light blue and red lines indicate when cellularization starts for the zebrafish and *Drosophila* embryos.”

The data for frog egg extract are taken from Fig. 5 c and d and Video 14, that includes all three extract conditions.

4. Figure 1b: The figure shows only one panel, while the caption implies multiple panels (with the description).

We are sorry for the mistake and have changed the caption accordingly.

Referee 2

The authors should be congratulated for their hard work. The paper is much improved by the addition of new data and by verification that the dimensionality of the model has no major impact on the results. The conclusions of the paper, made by comparing theory and experiment, are important. They help reveal the physical constraints that act to determine how cytoplasmic patterning can evolve in an early embryo. This is precisely where physical modeling makes the greatest contribution to our understanding of biology.

We thank the Reviewer for their enthusiasm about our work and appreciating our revisions.

I would still suggest that the references to Turing patterning removed, since in my view this type of language creates more confusion than it helps to enlighten.

We agree with the Reviewer and removed all the references to Turing patterns from the paper.

Referee 3

This revised manuscript shows significant improvements over the initial submission. In particular, the new 3D agent-based simulations provide realistic models for comparison with experiments. In addition, the authors have expanded their characterization of aster invasion and the modulation of MT dynamics *in vivo*, which reinforces the claims of the work.

We thank the reviewer for prompting us to perform these 3D simulations and *in vivo* perturbations, which have improved the conclusions of the paper. We thank them for appreciating our revision work.

I have one last point, which arises from the newly added material: In the 3D simulations, aster invasion appears to proceed with finger-like protrusions along the aster center-to-center axis (e.g., Fig. 1q), whereas this is not obvious in the experimental images and movies provided in the manuscript. Could the authors explain why this occurs in simulations but not necessarily in experiments?

We thank the Reviewer for this last point that can further clarify the invasion dynamics for the reader. We actually observe many finger-like protrusions during invasions in experiments. While some can be observed in Video 5 and related Figure 2a, we realize they might not be obvious to the reader. We now show an enlarged video of an experimental invasion event where it is possible to clearly observe finger-like protrusions during invasion. We have included this movie in **Video 8** (see **Figure R2** below for some snapshots). To explain the interface changes in experiments and 3D models, we note that changes in microtubule density at the interface are inherently stochastic. These stochastic fluctuations give rise to small microtubule protuberances of opposite polarity and result in finger-like protrusions from the combined effects of low angle branching and inhibition. Finally, stochastic fluctuations of microtubule densities have the greatest effect in the region of maximum inhibition hence its observed bias along the center-to-center axis.

We point this out in the text (page 5, line 5-8) as follows:

“In the simulations, the asters are confined in a rectangular slab, and the invasion occurs laterally, as also seen in the experiments, often with deformations of the interface in a finger-like fashion (**Video 8** and **Extended Data Fig. 7d**).”

Figure R2 Time sequence during aster invasion showing a finger-like protrusion at the center-to-center axis.

Third revision round: point by point response to referees

Legend:

Blue text: referee comments

Black text: response to referee comments

Green text: additions to the manuscript

Referee 1

The authors' responses and the additional changes in the manuscript have addressed my questions, and I would like to congratulate them on their beautiful work.

We thank the Reviewer for their supportive feedback.

One minor but important point, which I suggest the authors ensure before publication, concerns the data presented in Fig. 5d (related to my comment #3). The authors explain that the asters are derived from egg-intrinsic centrosomes. Are those not from high sperm nuclei? It is generally assumed that *Xenopus* eggs and extracts lack centrosomes and sperm brings them. Although it may be possible that the authors' cycling extracts promote the regeneration of centrosomes, it is important to confirm their assay method (i.e., whether sperm was added or not) and clarify their interpretation accordingly.

We thank the Reviewer for pointing out this last clarification. The sample in Figure 5d was obtained by adding sperm nuclei. Since we do not observe the sperm nuclei in the region where the centrosomes form and duplicate, we hypothesize that these centrosomes originate from the egg. This hypothesis is supported by experiments where we observed formation of small asters associated with centrosome duplication in cycling extract without added sperm nuclei or any other nucleating material. As the Reviewer correctly writes, the cycling extract assay could promote centrosome duplication. Indeed we observe formation of centrosome asters in these samples after a few cell cycles. We have now clarified the assay method in the legend of Figure 5 as follows:

“d Microtubule asters originating from centrosomes in extract with slower cell cycle time. This extract sample was supplemented with sperm nuclei, but asters originate from centrosomes that are not associated with nuclei.”

We have also written now on the micrographs in Figure 5c-d “Frog egg extract with sperm nuclei” and changed the label in Figure 5c to “sperm nuclei asters” and in Figure 5d to “centrosome asters”.

Referee 3

The authors have addressed my last minor point. Congratulations on a beautiful work!

We thank the Reviewer for their kind words.